# Threshold KNN-Shapley: A Linear-Time and Privacy-Friendly Approach to Data Valuation

**Jiachen T. Wang**[*]
Princeton University
tianhaowang@princeton.edu

**Yuqing Zhu**
UC Santa Barbara
yuqingzhu@ucsb.edu

**Yu-Xiang Wang**
UC Santa Barbara
yuxiangw@cs.ucsb.edu

**Ruoxi Jia**[*]
Virginia Tech
ruoxijia@vt.edu

**Prateek Mittal**
Princeton University
pmittal@princeton.edu

## Abstract

Data valuation aims to quantify the usefulness of individual data sources in training machine learning (ML) models, and is a critical aspect of data-centric ML research. However, data valuation faces significant yet frequently overlooked privacy challenges despite its importance. This paper studies these challenges with a focus on KNN-Shapley, one of the most practical data valuation methods nowadays. We first emphasize the inherent privacy risks of KNN-Shapley, and demonstrate the significant technical difficulties in adapting KNN-Shapley to accommodate differential privacy (DP). To overcome these challenges, we introduce *TKNN-Shapley*, a refined variant of KNN-Shapley that is privacy-friendly, allowing for straightforward modifications to incorporate DP guarantee (*DP*-TKNN-Shapley). We show that DP-TKNN-Shapley has several advantages and offers a superior privacy-utility tradeoff compared to naively privatized KNN-Shapley in discerning data quality. Moreover, even non-private TKNN-Shapley achieves comparable performance as KNN-Shapley. Overall, our findings suggest that TKNN-Shapley is a promising alternative to KNN-Shapley, particularly for real-world applications involving sensitive data.

## 1 Introduction

Data valuation is an emerging research area that seeks to measure the contribution of individual data sources to the training of machine learning (ML) models. Data valuation is essential in data marketplaces for ensuring equitable compensation for data owners, and in explainable ML for identifying influential training data. The importance of data valuation is underscored by the DASHBOARD Act introduced in the U.S. Senate in 2019 [66], which mandates companies to provide users with an assessment of their data's economic value. Moreover, OpenAI's Future Plan explicitly highlights "the fair distribution of AI-generated benefits" as a key question moving forward [51].

**Data Shapley.** Inspired by cooperative game theory, [23, 30] initiated the study of using the Shapley value as a principled approach for data valuation. The Shapley value [56] is a famous solution concept from game theory for fairly assigning the total revenue to each player. In a typical scenario of data valuation, training data points are collected from different sources, and the data owner of each training data point can be regarded as "player" in a cooperative game. "Data Shapley" refers to the method of using the Shapley value as the contribution measure for each data owner. Since its introduction, many

---

[*]Correspondence to **Jiachen T. Wang** and **Ruoxi Jia**.

37th Conference on Neural Information Processing Systems (NeurIPS 2023).

different variants of Data Shapley have been proposed [29, 22, 63, 6, 39, 44, 69, 33, 60], reflecting its effectiveness in quantifying data point contributions to ML model training.

**KNN-Shapley.** While being a principled approach for data valuation, the exact calculation of the Shapley value is computationally prohibitive [30]. Various approximation algorithms for Data Shapley have been proposed [30, 28, 50, 64, 7, 48, 44, 62], but these approaches still require substantial computational resources due to model retraining. Fortunately, a breakthrough by [29] showed that computing the *exact* Data Shapley for K-Nearest Neighbors (KNN), one of the oldest yet still popular ML algorithms, is surprisingly easy and efficient. KNN-Shapley quantifies data value based on KNN's Data Shapley score; it can be applied to large, high-dimensional CV/NLP datasets by calculating the value scores on the last-layer neural network embeddings. Owing to its superior efficiency and effectiveness in discerning data quality, KNN-Shapley is recognized as one of the most practical data valuation techniques nowadays [52]. It has been applied to various ML domains including active learning [24], continual learning [57], NLP [43, 42], and semi-supervised learning [9].

**Motivation: privacy risks in data valuation.** In this work, we study a critical, yet often overlooked concern in the deployment of data valuation: privacy leakage associated with data value scores released to data holders. The value of a single data point is always relative to other data points in the training set. This, however, can potentially reveal sensitive information about the rest of data holders in the dataset. This problem becomes even more complex when considering a strong threat model where multiple data holders collude, sharing their received data values to determine the membership of a particular individual. As data valuation techniques such as KNN-Shapley become increasingly popular and relevant in various applications, understanding and addressing the privacy challenges of data valuation methods is of utmost importance. In this work, we study this critical issue through the lens of differential privacy (DP) [15], a de-facto standard for privacy-preserving applications.

Our technical contributions are listed as follows.

**Privacy Risks & Challenges of Privatization for KNN-Shapley** (Section 3). We demonstrate that data value scores (specifically KNN-Shapley) indeed serve as a new channel for private information leakage, potentially exposing sensitive information about individuals in the dataset. In particular, we explicitly design a privacy attack (in Appendix B.2.2) where an adversary could infer the presence/absence of certain data points based on the variations in the KNN-Shapley scores, analogous to the classic membership inference attack on ML model [58]. Additionally, we highlight the technical challenges in incorporating the current KNN-Shapley technique with differential privacy, such as its large global sensitivity. All these results emphasize the need for a new, privacy-friendly approach to data valuation.

**TKNN-Shapley: an efficient, privacy-friendly data valuation technique** (Section 4.2). To address the privacy concerns, we derive a novel variant of KNN-Shapley. This new method considers the Data Shapley of an alternative form of KNN classifier called *Threshold-KNN* (TKNN) [5], which takes into account all neighbors within a pre-specified threshold of a test example, rather than the exact $K$ nearest neighbors. We derive the closed-form formula of Data Shapley for TKNN (i.e., TKNN-Shapley). We show that it can be computed *exactly* with exact linear-time computational efficiency over the original KNN-Shapley. **DP-TKNN-Shapley** (Section 5). Importantly, we recognize that TKNN-Shapley only depends on three simple counting queries. Hence, TKNN-Shapley can be conveniently transformed into a differentially private version by DP's post-processing property. Moreover, we prove that such a DP variant satisfies several favorable properties, including (1) the high computational efficiency, (2) the capability to withstand collusion among data holders without compromising the privacy guarantee, and (3) the ease of integrating subsampling for privacy amplification.

**Numerical experiments** (Section 6). We evaluate the performance of TKNN-Shapley across 11 commonly used benchmark datasets and 2 NLP datasets. Key observations include: (1) TKNN-Shapley surpasses KNN-Shapley in terms of computational efficiency; (2) DP-TKNN-Shapley significantly outperforms the naively privatized KNN-Shapley in terms of privacy-utility tradeoff in discerning data quality; (3) even non-private TKNN-Shapley achieves comparable performance as KNN-Shapley.

Overall, our work suggests that TKNN-Shapley, being a privacy-friendly, yet more efficient and effective alternative to the original KNN-Shapley, signifies a milestone toward practical data valuation.

## 2 Background of Data Valuation

In this section, we formalize the data valuation problem for ML, and review the method of Data Shapley and KNN-Shapley.

**Setup & Goal.** Consider a dataset $D := \{z_i\}_{i=1}^N$ consisting of $N$ data points where each data point $z_i := (x_i, y_i)$ is collected from a data owner $i$. The objective of data valuation is to attribute a score to each training data point $z_i$, reflecting its importance or quality in ML model training. Formally, we aim to determine a score vector $(\phi_{z_i})_{i=1}^N$, wherein $\phi_{z_i}$ represents the value of data point $z_i$. For any reasonable data valuation method, the value of a data point is always relative to other data points in the dataset. For instance, if a data point has many duplicates in the dataset, its value will likely be lower. Hence, $\phi_{z_i}$ is a function of the leave-one-out dataset $D_{-z_i} := D \setminus \{z_i\}$. We write $\phi_{z_i}(D_{-z_i})$ when we want to stress the dependency of a data value score with the rest of the data points.

**Utility Function.** Most of the existing data valuation techniques are centered on the concept of *utility function*, which maps an input dataset to a score indicating the usefulness of the training set. A common choice for utility function is the *validation accuracy* of a model trained on the input training set. Formally, for a training set $S$, a utility function $v(S) := \mathtt{acc}(\mathcal{A}(S))$, where $\mathcal{A}$ is a learning algorithm that takes a dataset $S$ as input and returns a model; $\mathtt{acc}(\cdot)$ is a metric function that evaluates the performance of a given model, e.g., the classification accuracy on a hold-out validation set.

### 2.1 The Shapley Value

The Shapley value (SV) [56] is a classic concept from game theory to attribute the total gains generated by the coalition of all players. At a high level, it appraises each point based on the (weighted) average utility change caused by adding the point into different subsets of the training set. Formally, given a utility function $v(\cdot)$ and a training set $D$, the Shapley value of a data point $z \in D$ is defined as

$$\phi_z\left(D_{-z}; v\right) := \frac{1}{N} \sum_{k=1}^N \binom{N-1}{k-1}^{-1} \sum_{S \subseteq D_{-z}, |S|=k-1} [v(S \cup \{z\}) - v(S)] \tag{1}$$

For notation simplicity, when the context is clear, we omit the utility function and/or leave-one-out dataset, and write $\phi_z(D_{-z})$, $\phi_z(v)$ or $\phi_z$ depending on the specific dependency we want to stress.

The popularity of the Shapley value is attributable to the fact that it is the *unique* data value notion satisfying four axioms: Dummy player, Symmetry, Linearity, and Efficiency. We refer the readers to [23], [30] and the references therein for a detailed discussion about the interpretation and necessity of the four axioms in the ML context. Here, we introduce the *linearity* axiom which will be used later.

**Theorem 1** (Linearity of the Shapley value [56]). *For any of two utility functions $v_1, v_2$ and any $\alpha_1, \alpha_2 \in \mathbb{R}$, we have $\phi_z\left(\alpha_1 v_1 + \alpha_2 v_2\right) = \alpha_1 \phi_z\left(v_1\right) + \alpha_2 \phi_z\left(v_2\right)$.*

### 2.2 KNN-Shapley

Formula (1) suggests that the exact Shapley value can be computationally prohibitive in general, as it requires evaluating $v(S)$ for all possible subsets $S \subseteq D$. Surprisingly, [29, 61] showed that for $K$-Nearest Neighbor (KNN), the computation of the exact Data Shapley score is highly efficient. Following its introduction, KNN-Shapley has rapidly gained attention and follow-up works across diverse areas of machine learning [24, 43, 42, 9]. In particular, it has been recognized by recent studies as "the most practical data valuation technique capable of handling large-scale data effectively" [52, 33].

Specifically, the performance of an unweighted KNN classifier is typically evaluated by its validation accuracy. For a given validation set $D^{(\mathrm{val})} = \{z_i^{(\mathrm{val})}\}_{i=1}^{N_{\mathrm{val}}}$, we can define KNN's utility function $v_{D^{(\mathrm{val})}}^{\mathtt{KNN}}$ on a non-empty training set $S$ as $v_{D^{(\mathrm{val})}}^{\mathtt{KNN}}(S) := \sum_{z^{(\mathrm{val})} \in D^{(\mathrm{val})}} v_{z^{(\mathrm{val})}}^{\mathtt{KNN}}(S)$, where

$$v_{z^{(\mathrm{val})}}^{\mathtt{KNN}}(S) := \frac{1}{\min(K, |S|)} \sum_{j=1}^{\min(K, |S|)} \mathbb{1}[y_{\pi^{(S)}(j; x^{(\mathrm{val})})} = y^{(\mathrm{val})}] \tag{2}$$

is the probability of a (soft-label) KNN classifier in predicting the correct label for a validation point $z^{(\mathrm{val})} = (x^{(\mathrm{val})}, y^{(\mathrm{val})}) \in D^{(\mathrm{val})}$. In (2), $\pi^{(S)}(i; x^{(\mathrm{val})})$ is the index of the $i$th closest data point in

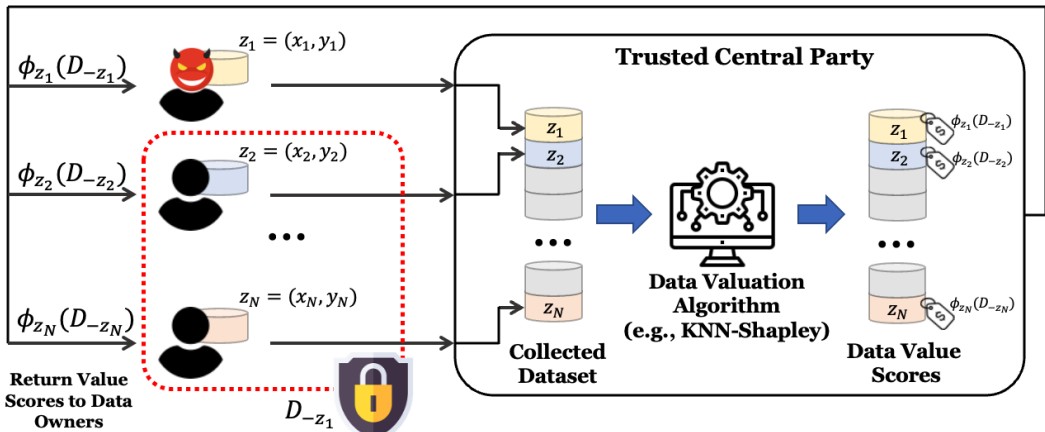

Figure 1: The potential privacy risks in data valuation arise from the dependency of the data value score $\phi_{z_i}$ on the rest of the dataset $D_{-z_i}$. Our goal is to privatize $\phi_{z_i}(D_{-z_i})$ such that it provides strong differential privacy guarantee for the rest of the dataset $D_{-z_i}$.

$S$ to $x^{(\text{val})}$. When $|S| = 0$, $v_{z^{(\text{val})}}^{\text{KNN}}(S)$ is set to the accuracy by random guessing. The distance is measured through suitable metrics such as $\ell_2$ distance. Here is the main result in [29, 61]:

**Theorem 2** (KNN-Shapley [30, 61] (simplified version)). *Consider the utility function in (2). Given a validation data point $z^{(\text{val})} = (x^{(\text{val})}, y^{(\text{val})})$ and a distance metric $d(\cdot, \cdot)$, if we sort the training set $D = \{z_i = (x_i, y_i)\}_{i=1}^{N}$ according to $d(x_i, x^{(\text{val})})$ in ascending order, then the Shapley value of each data point $\phi_{z_i}^{\text{KNN}}$ corresponding to utility function $v_{z^{(\text{val})}}^{\text{KNN}}$ can be computed recursively as follows:*

$$\phi_{z_N}^{\text{KNN}} := f_N(D), \ \text{ and } \ \phi_{z_i}^{\text{KNN}} := \phi_{z_{i+1}}^{\text{KNN}} + f_i(D) \ \text{ for } i = 1, \ldots, N-1$$

*where the exact form of functions $f_i(D)$ can be found in Appendix B.1.*

**Runtime:** *the computation of all Shapley values $(\phi_{z_1}^{\text{KNN}}, \ldots, \phi_{z_N}^{\text{KNN}})$ can be achieved in $O(N \log N)$ runtime in total (dominated by the sorting data points in $D$).*

**The Shapley value corresponding to full validation set:** *for a validation set $D^{(\text{val})}$, recall that $v_{D^{(\text{val})}}^{\text{KNN}}(S) := \sum_{z^{(\text{val})} \in D^{(\text{val})}} v_{z^{(\text{val})}}^{\text{KNN}}(S)$. One can compute the Shapley value corresponding to $v_{D^{(\text{val})}}^{\text{KNN}}$ by summing each $\phi_{z_i}^{\text{KNN}}\left(v_{z^{(\text{val})}}^{\text{KNN}}\right)$, i.e., $\phi_{z_i}^{\text{KNN}}\left(v_{D^{(\text{val})}}^{\text{KNN}}\right) = \sum_{z^{(\text{val})} \in D^{(\text{val})}} \phi_{z_i}^{\text{KNN}}\left(v_{z^{(\text{val})}}^{\text{KNN}}\right)$.*

Theorem 2 tells that for any validation data point $z^{(\text{val})}$, we can compute the *exact* Shapley value $\phi_{z_i}^{\text{KNN}}\left(D_{-z_i}; v_{z^{(\text{val})}}^{\text{KNN}}\right)$ for *all* $z_i \in D$ by using a recursive formula within a total runtime of $O(N \log N)$. After computing the Shapley value $\phi_{z_i}^{\text{KNN}}\left(v_{z^{(\text{val})}}^{\text{KNN}}\right)$ for each $z^{(\text{val})} \in D^{(\text{val})}$, one can compute the Shapley value corresponding to the full validation set by simply taking the sum of each $\phi_{z_i}^{\text{KNN}}\left(v_{z^{(\text{val})}}^{\text{KNN}}\right)$, due to the linearity property (Theorem 1) of the Shapley value.

**Remark 1** (Criteria of Computational Efficiency). *As prior data valuation literature [23], we focus on the total runtime required to release* all *data value scores $(\phi_{z_1}^{\text{KNN}}, \ldots, \phi_{z_N}^{\text{KNN}})$. This is because, in a practical data valuation scenario (e.g., profit allocation), we rarely want to compute the data value score for only a single data point; instead, a typical objective is to compute the data value scores for all data points within the training set.*

## 3 Privacy Risks & Challenges of Privatization for KNN-Shapley

**Scenario.** Figure 1 illustrates the data valuation scenario and potential privacy leakages considered in our paper. Specifically, a centralized, trusted server collects data point $z_i$ from data owner $i$ for each $i \in [N]$. The central server's role is to provide each data owner $i$ with an assessment of the value of their data $z_i$, e.g., the KNN-Shapley value $\phi_{z_i}^{\text{KNN}}$. **A real life example:** Mayo Clinic has created a massive digital health patient data marketplace platform [68], where the patients submit part of their medical records onto the platform, and life science companies/labs pay a certain amount of money to purchase patients' data. The platform's responsibility is to gauge the worth of the data of each patient (i.e., the data owner) to facilitate fair compensation.

The privacy risks associated with KNN-Shapley (as well as other data valuation techniques) arise from the fact that $\phi_{z_i}^{\text{KNN}}(D_{-z_i})$ depends on other data owners' data $D_{-z_i}$. Consequently, the data value score $\phi_{z_i}^{\text{KNN}}$ may inadvertently reveal private information (e.g., membership) about the rest of the dataset. The dependency of a data value score on the rest of the dataset is an unavoidable aspect of data valuation, as the value of a data point is inherently a relative quantity determined by its role within the complete dataset.

**Remark 2** (**Other privacy risks in data valuation**). *It is important to note that in this work, we do not consider the privacy risks of revealing individuals' data to the central server. This is a different type of privacy risk that needs to be addressed using secure multi-party computation (MPC) technique [70], and it should be used together with differential privacy in practice. In addition, to use KNN-Shapley or many other data valuation techniques, the central server needs to maintain a clean, representative validation set, the privacy of which is not considered by this paper.*

**A Simple Membership Inference (MI) Attack on KNN-Shapley (detailed in Appendix B.2.2).** We further illustrate the privacy risks of revealing data value scores with a concrete example. Analogous to the classic membership inference attack on ML model [58], in Appendix B.2.2 we show an example of privacy attack where an adversary could infer the presence/absence of certain data points in the dataset based on the variations in the KNN-Shapley scores. The design is analogous to the membership inference attack against ML models via the *likelihood ratio test* [8]. The AUROC score of the attack results is shown in Table 1. As we can see, our

| | $K=1$ | $K=3$ | $K=5$ | $K=7$ | $K=9$ | $K=11$ | $K=13$ | $K=15$ |
|---|---|---|---|---|---|---|---|---|
| **2DPlanes** | 0.56 | 0.595 | 0.518 | 0.52 | 0.57 | 0.55 | 0.515 | 0.6 |
| **Phoneme** | 0.692 | 0.54 | 0.513 | 0.505 | 0.505 | 0.588 | 0.512 | 0.502 |
| **CPU** | 0.765 | 0.548 | 0.52 | 0.572 | 0.588 | 0.512 | 0.612 | 0.615 |
| **Fraud** | 0.625 | 0.645 | 0.6 | 0.592 | 0.622 | 0.532 | 0.538 | 0.558 |
| **Creditcard** | 0.542 | 0.643 | 0.628 | 0.665 | 0.503 | 0.67 | 0.602 | 0.66 |
| **Apsfail** | 0.532 | 0.595 | 0.625 | 0.6 | 0.53 | 0.645 | 0.532 | 0.52 |
| **Click** | 0.61 | 0.525 | 0.588 | 0.538 | 0.588 | 0.582 | 0.622 | 0.618 |
| **Wind** | 0.595 | 0.51 | 0.518 | 0.528 | 0.558 | 0.562 | 0.505 | 0.577 |
| **Pol** | 0.725 | 0.695 | 0.7 | 0.62 | 0.57 | 0.522 | 0.535 | 0.532 |

Table 1: Results of the AUROC of the MI attack proposed in Appendix B.2.2 on KNN-Shapley. The higher the AUROC score is, the larger the privacy leakage is. The detailed algorithm description and experiment settings can be found in Appendix B.2.2.

MIA attack can achieve a detection performance that is better than the random guess (0.5) for most of the settings. On some datasets, the attack performance can achieve $> 0.7$ AUROC. This demonstrates that privacy leakage in data value scores can indeed lead to non-trivial privacy attacks, and underscores the need for privacy safeguards in data valuation.[1]

### 3.1 Privatizing KNN-Shapley is Difficult

The growing popularity of data valuation techniques, particularly KNN-Shapley, underscores the critical need to mitigate the inherent privacy risks. In the quest for privacy protection, differential privacy (DP) [15] has emerged as the leading framework. DP has gained considerable recognition for providing robust, quantifiable privacy guarantees, thereby becoming the de-facto choice in privacy protection. In this section, we introduce the background of DP and highlight the technical difficulties in constructing a differentially private variant for the current version of KNN-Shapley.

**Background of Differential Privacy.** We use $D, D' \in \mathbb{N}^{\mathcal{X}}$ to denote two datasets with an unspecified size over space $\mathcal{X}$. We call two datasets $D$ and $D'$ *adjacent* (denoted as $D \sim D'$) if we can construct one by adding/removing one data point from the other, e.g., $D = D' \cup \{z\}$ for some $z \in \mathcal{X}$.

**Definition 3** (Differential Privacy [15]). *For $\varepsilon, \delta \geq 0$, a randomized algorithm $\mathcal{M} : \mathbb{N}^{\mathcal{X}} \to \mathcal{Y}$ is $(\varepsilon, \delta)$-differentially private if for every pair of adjacent datasets $D \sim D'$ and for every subset of possible outputs $E \subseteq \mathcal{Y}$, we have $\Pr_{\mathcal{M}}[\mathcal{M}(D) \in E] \leq e^{\varepsilon} \Pr_{\mathcal{M}}[\mathcal{M}(D') \in E] + \delta$.*

That is, $(\varepsilon, \delta)$-DP requires that for all adjacent datasets $D, D'$, the output distribution $\mathcal{M}(D)$ and $\mathcal{M}(D')$ are close, where the closeness is measured by the parameters $\varepsilon$ and $\delta$. In our scenario, we would like to modify data valuation function $\phi_{z_i}(D_{-z_i})$ to satisfy $(\varepsilon, \delta)$-DP in order to protect the privacy of the rest of the dataset $D_{-z_i}$. Gaussian mechanism [14] is a common way for privatizing a function; it introduces Gaussian noise aligned with the function's *global sensitivity*, which is the maximum output change when a single data point is added/removed from any given dataset.

**Definition 4** (Gaussian Mechanism [14]). *Define the global sensitivity of a function $f : \mathbb{N}^{\mathcal{X}} \to \mathbb{R}^d$ as $\Delta_2(f) := \sup_{D \sim D'} \|f(D) - f(D')\|_2$. The Gaussian mechanism $\mathcal{M}$ with noise level $\sigma$ is then given by $\mathcal{M}(D) := f(D) + \mathcal{N}\left(0, \sigma^2 \mathbb{1}_d\right)$ where $\mathcal{M}$ is $(\varepsilon, \delta)$-DP with $\sigma = \Delta_2(f)\sqrt{\log(1.25/\delta)}/\varepsilon$.*

---

[1] We stress that the goal here is to demonstrate that the data value scores can indeed serve as another channel of privacy leakage. We do *not* claim any optimality of the attack we construct here. Improving MI attacks for data valuation is an interesting future work.

**Challenges in making KNN-Shapley being differentially private (overview).** Here, we give an overview of the inherent difficulties in making the KNN-Shapley $\phi_{z_i}^{\text{KNN}}(D_{-z_i})$ (Theorem 2) to be differentially private, and we provide a more detailed discussion in Appendix B.3. **(1) Large global sensitivity:** In Appendix B.3, we show that the global sensitivity of $\phi_{z_i}^{\text{KNN}}(D_{-z_i})$ can significantly exceed the magnitude of $\phi_{z_i}^{\text{KNN}}$. Moreover, we prove that the global sensitivity bound cannot be further improved by constructing a specific pair of neighboring datasets that matches the bound. Hence, if we introduce random noise proportional to the global sensitivity bound, the resulting privatized data value score could substantially deviate from its non-private counterpart, thereby compromising the utility of the privatized data value scores. **(2) Computational challenges in incorporating privacy amplification by subsampling:** "Privacy amplification by subsampling" [4] is a technique where the subsampling of a dataset amplifies the privacy guarantees due to the reduced probability of an individual's data being included. Being able to incorporate such a technique is often important for achieving a decent privacy-utility tradeoff. However, in Appendix B.3 we show that the recursive nature of KNN-Shapley computation causes a significant increase in computational demand compared to non-private KNN-Shapley.

These challenges underscore the pressing need for new data valuation techniques that retain the efficacy and computational efficiency of KNN-Shapley, while being amenable to privatization.

## 4 The Shapley Value for Threshold-based Nearest Neighbor

Considering the privacy concerns and privatization challenges associated with the original KNN-Shapley method, we introduce *TKNN-Shapley*, a privacy-friendly alternative of KNN-Shapley which also achieves improved computational efficiency. At the core of this novel method is Threshold-KNN (TKNN) classifier, a simple variant of the KNN classifier. We will discuss how to incorporate DP for TKNN-Shapley in Section 5.

### 4.1 Threshold-based Nearest Neighbor Classifier (TKNN)

Threshold-KNN (TKNN) [5, 72] is a variant of KNN classifier that considers neighbors within a pre-specified threshold of the query example, rather than exclusively focusing on the exact $K$ nearest neighbors. Formally, for a training set $S$ and a validation data point $z^{(\text{val})} = (x^{(\text{val})}, y^{(\text{val})})$, we denote $\text{NB}_{x^{(\text{val})}, \tau}(S) := \{(x, y) | (x, y) \in S, d(x, x^{(\text{val})}) \leq \tau\}$ the set of neighbors of $x^{(\text{val})}$ in $S$ within a pre-specified threshold $\tau$. Similar to the utility function for KNN, we define the utility function for TKNN classifier when using a validation set $D^{(\text{val})}$ as the aggregated prediction accuracy $v_{D^{(\text{val})}}^{\text{TKNN}}(S) := \sum_{z^{(\text{val})} \in D^{(\text{val})}} v_{z^{(\text{val})}}^{\text{TKNN}}(S)$ where

$$v_{z^{(\text{val})}}^{\text{TKNN}}(S) := \begin{cases} \texttt{Constant} & |\text{NB}_{x^{(\text{val})}, \tau}(S)| = 0 \\ \frac{1}{|\text{NB}_{x^{(\text{val})}, \tau}(S)|} \sum_{(x,y) \in \text{NB}_{x^{(\text{val})}, \tau}(S)} \mathbb{1}[y = y^{(\text{val})}] & |\text{NB}_{x^{(\text{val})}, \tau}(S)| > 0 \end{cases} \quad (3)$$

where $\texttt{Constant}$ can be the trivial accuracy of random guess.

**Comparison with standard KNN. (1) Robustness to outliers.** Compared with KNN, TKNN is better equipped to deal with prediction phase outliers [5]. When predicting an outlier that is far from the entire training set, TKNN prevents the influence of distant, potentially irrelevant neighbors, leading to a more reliable prediction score for outliers. **(2) Inference Efficiency.** TKNN has slightly better computational efficiency compared to KNN, as it has $O(N)$ instead of $O(N \log N)$ inference time. This improvement is achieved because TKNN only requires the computation of neighbors within the threshold $\tau$, rather than searching for the exact $K$ nearest neighbors. **(3) TKNN is also a consistent estimator.** The consistency of standard KNN is a well-known theoretical result [26]. That is, for any target function that satisfies certain regularity conditions, KNN binary classifier/regressor is guaranteed to converge to the target function as the size of the training set grows to infinite. In Appendix C.1, we derived a similar consistency result for TKNN binary classifier/regressor.

**Remark 3** (**Intuition: Why we consider TKNN?**). *The 'recursive form' of KNN-Shapley is due to the sorting operation in the prediction of the standard KNN. The recursive form of the formula causes difficulties in incorporating KNN-Shapley with differential privacy. In contrast, TKNN avoids the recursive formula for its Data Shapley value; the intuition is that for TKNN, the selection of neighbors for prediction solely depends on the queried example and the validation data, and is independent of the other training data points.*

## 4.2 Data Shapley for TKNN (TKNN-Shapley)

With the introduction of TKNN classifier and its utility function, we now present our main result, the closed-form, efficiently computable Data Shapley formula for the TKNN classifier.

**Theorem 5** (TKNN-Shapley (simplified version)). *Consider the utility function $v_{z^{(\text{val})}}^{TKNN}$ in (3). Given a validation data point $z^{(\text{val})} = (x^{(\text{val})}, y^{(\text{val})})$ and a distance metric $d(\cdot, \cdot)$, the Shapley value $\phi_{z_i}^{TKNN}$ of each training point $z_i = (x_i, y_i) \in D$ corresponding to utility function $v_{z^{(\text{val})}}^{TKNN}$ can be calculated as*

$$\phi_{z_i}^{TKNN} = f\left(\boldsymbol{C}_{z^{(\text{val})}}(D_{-z_i})\right) \tag{4}$$

*where $\boldsymbol{C}_{z^{(\text{val})}} := (\mathbf{c}, \mathbf{c}_{x^{(\text{val})},\tau}, \mathbf{c}_{z^{(\text{val})},\tau}^{(+)})$ is a 3-dimensional function/vector s.t.*

$$\mathbf{c} = \mathbf{c}(D_{-z_i}) := |D_{-z_i}| \qquad\qquad (\text{size of } D_{-z_i})$$

$$\mathbf{c}_{x^{(\text{val})},\tau} = \mathbf{c}_{x^{(\text{val})},\tau}(D_{-z_i}) := 1 + \left|NB_{x^{(\text{val})},\tau}(D_{-z_i})\right| \qquad (1 + \text{\# neighbors of } x^{(\text{val})} \text{ in } D_{-z_i})$$

$$\mathbf{c}_{z^{(\text{val})},\tau}^{(+)} = \mathbf{c}_{z^{(\text{val})},\tau}^{(+)}(D_{-z_i}) := \sum_{(x,y) \in NB_{x^{(\text{val})},\tau}(D_{-z_i})} \mathbb{1}[y = y^{(\text{val})}] \qquad (\text{\# same-label neighbors in } D_{-z_i})$$

*and $f(\cdot)$ is a function whose exact form can be found in Appendix C.2.*

**Runtime:** *the computation of all $(\phi_{z_1}^{TKNN}, \ldots, \phi_{z_N}^{TKNN})$ can be achieved in $O(N)$ runtime (see Appendix C.2.1).*

**The Shapley value when using full validation set:** *The Shapley value corresponding to the utility function $v_{D^{(\text{val})}}^{TKNN}$ can be calculated as $\phi_{z_i}^{TKNN}\left(v_{D^{(\text{val})}}^{TKNN}\right) = \sum_{z^{(\text{val})} \in D^{(\text{val})}} \phi_{z_i}^{TKNN}\left(v_{z^{(\text{val})}}^{TKNN}\right).$*

The main technical challenges of proving Theorem 5 lies in showing $\boldsymbol{C}_{z^{(\text{val})}} = (\mathbf{c}, \mathbf{c}_{x^{(\text{val})},\tau}, \mathbf{c}_{z^{(\text{val})},\tau}^{(+)})$, three simple counting queries on $D_{-z_i}$, are the key quantities for computing $\phi_{z_i}^{\text{TKNN}}$. In later part of the paper, we will often view $\phi_{z_i}^{\text{TKNN}}$ as a function of $\boldsymbol{C}_{z^{(\text{val})}}$ and denote $\phi_{z_i}^{\text{TKNN}}[\boldsymbol{C}_{z^{(\text{val})}}] := f(\boldsymbol{C}_{z^{(\text{val})}})$ when we want to stress this dependency. The full version and the proof for TKNN-Shapley can be found in Appendix C.2.1. Here, we briefly discuss why TKNN-Shapley can achieve $O(N)$ runtime *in total*.

**Efficient Computation of TKNN-Shapley.** As we can see, all of the quantities in $\boldsymbol{C}_{z^{(\text{val})}}$ are simply counting queries on $D_{-z_i}$, and hence $\boldsymbol{C}_{z^{(\text{val})}}(D_{-z_i})$ can be computed in $O(N)$ runtime for any $z_i \in D$. A more efficient way to compute $\boldsymbol{C}_{z^{(\text{val})}}(D_{-z_i})$ for *all* $z_i \in D$, however, is to first compute $\boldsymbol{C}_{z^{(\text{val})}}(D)$ on the full dataset $D$, and we can easily show that each of $\boldsymbol{C}_{z^{(\text{val})}}(D_{-z_i})$ can be computed from that in $O(1)$ runtime (see Appendix C.2.2 for details). Hence, we can compute TKNN-Shapley $\phi_{z_i}^{\text{TKNN}}(v_{z^{(\text{val})}}^{\text{TKNN}})$ for *all* $z_i \in D$ within an overall computational cost of $O(N)$.

**Comparison with KNN-Shapley.** TKNN-Shapley offers several advantages over the original KNN-Shapley. **(1) Non-recursive:** In contrast to the KNN-Shapley formula (Theorem 2), which is recursive, TKNN-Shapley has an explicit formula for computing the Shapley value of every point $z_i$. This non-recursive nature not only simplifies the implementation, but also makes it straightforward to incorporate techniques like subsampling. **(2) Computational efficiency:** TKNN-Shapley has $O(N)$ runtime in total, which is better than the $O(N \log N)$ runtime for KNN-Shapley.

## 5 Differentially Private TKNN-Shapley

Having established TKNN-Shapley as an alternative to KNN-Shapley which shares many advantages, our next step is to develop a new version of TKNN-Shapley that incorporates differential privacy. In this section, we introduce our proposed Differentially Private TKNN-Shapley (DP-TKNN-Shapley).

### 5.1 Differentially private release of $\phi_{z_i}^{\text{TKNN}}(v_{z^{(\text{val})}}^{\text{TKNN}})$

As $\boldsymbol{C}_{z^{(\text{val})}} = (\mathbf{c}, \mathbf{c}_{x^{(\text{val})},\tau}, \mathbf{c}_{z^{(\text{val})},\tau}^{(+)})$ are the only quantities that depend on the dataset $D_{-z_i}$ in (4), privatizing these quantities is sufficient for overall privacy preservation, as DP guarantee is preserved under any post-processing [16]. Since all of $\mathbf{c}, \mathbf{c}_{x^{(\text{val})},\tau}$ and $\mathbf{c}_{z^{(\text{val})},\tau}^{(+)}$ are just counting queries, the function of $\boldsymbol{C}_{z^{(\text{val})}}(\cdot)$ has global sensitivity $\sqrt{3}$ in $\ell_2$-norm. This allows us to apply the commonly used Gaussian mechanism (Theorem 4) to release $\boldsymbol{C}_{z^{(\text{val})}}$ in a differentially private way, i.e., $\widehat{\boldsymbol{C}}_{z^{(\text{val})}}(D_{-z_i}) := \text{round}\left(\boldsymbol{C}_{z^{(\text{val})}}(D_{-z_i}) + \mathcal{N}\left(0, \sigma^2 \mathbb{1}_3\right)\right)$ where $\widehat{\boldsymbol{C}}_{z^{(\text{val})}}$ is the privatized version of $\boldsymbol{C}_{z^{(\text{val})}}$, and the function $\text{round}$ rounds each entry of noisy value to the nearest integer. The differentially private version of TKNN-Shapley value $\widehat{\phi}_{z_i}^{\text{TKNN}}(D_{-z_i})$ is simply the value score computed by

(4) but use the privatized quantities $\widehat{\mathbf{C}}_{z^{(\text{val})}}$. The privacy guarantee of releasing DP-TKNN-Shapley $\widehat{\phi}_{z_i}^{\text{TKNN}}(D_{-z_i})$ follows from the guarantee of Gaussian mechanism (see Appendix D.1 for proof).

**Theorem 6.** *For any* $z^{(\text{val})} \in D^{(\text{val})}$, *releasing* $\widehat{\phi}_{z_i}^{\text{TKNN}}(v_{z^{(\text{val})}}^{\text{TKNN}}) := \phi_{z_i}^{\text{TKNN}}\left[\widehat{\mathbf{C}}_{z^{(\text{val})}}(D_{-z_i})\right]$ *to data owner* $i$ *is* $(\varepsilon, \delta)$*-DP with* $\sigma = \sqrt{3} \cdot \sqrt{\log(1.25/\delta)}/\varepsilon$.

While our approach may seem simple, we stress that simplicity is appreciated in DP as complex mechanisms pose challenges for correct implementation and auditing [47, 21].

## 5.2 Advantages of DP-TKNN-Shapley (Overview)

We give an overview (detailed in Appendix D.2) of several advantages of DP-KNN-Shapley here, including the efficiency, collusion resistance, and simplicity in incorporating subsampling.

**By reusing privatized statistics,** $\widehat{\phi}_{z_i}^{\text{TKNN}}$ **can be efficiently computed for all** $z_i \in D$**.** Recall that in practical data valuation scenarios, it is often desirable to compute the data value scores for *all of* $z_i \in D$. As we detailed in Section 4.2, for TKNN-Shapley, such a computation only requires $O(N)$ runtime in total if we first compute $\mathbf{C}_{z^{(\text{val})}}(D)$ and subsequently $\mathbf{C}_{z^{(\text{val})}}(D_{-z_i})$ for each $z_i \in D$. In a similar vein, we can efficiently compute the DP variant $\widehat{\phi}_{z_i}^{\text{TKNN}}$ for *all* $z_i \in D$. To do so, we first calculate $\widehat{\mathbf{C}}_{z^{(\text{val})}}(D)$ and then, for each $z_i \in D$, we compute $\widehat{\mathbf{C}}_{z^{(\text{val})}}(D_{-z_i})$. It is important to note that when releasing $\widehat{\phi}_{z_i}^{\text{TKNN}}$ to individual $i$, the data point they hold, $z_i$, is not private to the individual themselves. Therefore, as long as $\widehat{\mathbf{C}}_{z^{(\text{val})}}(D)$ is privatized, $\widehat{\mathbf{C}}_{z^{(\text{val})}}(D_{-z_i})$ and hence $\widehat{\phi}_{z_i}^{\text{TKNN}}$ also satisfy the same privacy guarantee due to DP's post-processing property.

**By reusing privatized statistics, DP-TKNN-Shapley is collusion resistance.** In Section 5.1, we consider the single Shapley value $\phi_{z_i}^{\text{TKNN}}(D_{-z_i})$ as the function to privatize. However, when we consider the release of all Shapley values $(\widehat{\phi}_{z_1}^{\text{TKNN}}, \ldots, \widehat{\phi}_{z_N}^{\text{TKNN}})$ as a unified mechanism, one can show that such a mechanism satisfies *joint differential privacy* (JDP) [34] if we reuse the privatized statistic $\widehat{\mathbf{C}}_{z^{(\text{val})}}(D)$ for the release of all $\widehat{\phi}_{z_i}^{\text{TKNN}}$. The consequence of satisfying JDP is that our mechanism is resilient against collusion among groups of individuals without any privacy degradation. That is, even if an arbitrary group of individuals in $[N] \setminus i$ colludes (i.e., shares their respective $\widehat{\phi}_{z_j}^{\text{TKNN}}$ values within the group), the privacy of individual $i$ remains uncompromised. Our method also stands resilient in scenarios where a powerful adversary sends multiple data points and receives multiple value scores.

**Incorporating Privacy Amplification by Subsampling.** In contrast to KNN-Shapley, the non-recursive nature of DP-TKNN-Shapley allows for the straightforward incorporation of subsampling. Besides the privacy guarantee, subsampling also significantly boosts the computational efficiency of DP-TKNN-Shapley.

## 5.3 Differentially private release of $\phi_{z_i}^{\text{TKNN}}(v_{D^{(\text{val})}}^{\text{TKNN}})$

Recall that the TKNN-Shapley corresponding to the full validation set $D^{(\text{val})}$ is $\phi_{z_i}^{\text{TKNN}}\left(v_{D^{(\text{val})}}^{\text{TKNN}}\right) = \sum_{z^{(\text{val})} \in D^{(\text{val})}} \phi_{z_i}^{\text{TKNN}}(D_{-z_i}; z^{(\text{val})})$. We can compute privatized $\phi_{z_i}^{\text{TKNN}}\left(v_{D^{(\text{val})}}^{\text{TKNN}}\right)$ by simply releasing $\phi_{z_i}^{\text{TKNN}}(D_{-z_i}; z^{(\text{val})})$ for all $z^{(\text{val})} \in D^{(\text{val})}$. To better keep track of the privacy cost, we use the current state-of-the-art privacy accounting technique based on the notion of the Privacy Loss Random Variable (PRV) [17]. We provide more details about the background of privacy accounting in Appendix D.3. We note that PRV-based privacy accountant computes the privacy loss numerically, and hence the final privacy loss has no closed-form expressions.

# 6 Numerical Experiments

In this section, we systematically evaluate the practical effectiveness of our proposed TKNN-Shapley method. Our evaluation aims to demonstrate the following points: **(1)** TKNN-Shapley offers improved runtime efficiency compared with KNN-Shapley. **(2)** The differentially private version of TKNN-Shapley (DP-TKNN-Shapley) achieves significantly better privacy-utility tradeoff compared to naively privatized KNN-Shapley in discerning data quality. **(3)** Non-private TKNN-Shapley maintains a comparable performance to the original KNN-Shapley. These observations highlight TKNN-Shapley's potential for data valuation in real-life applications. Detailed settings for our experiments are provided in Appendix E.

**Remark 4.** *Given that differential privacy offers a provable guarantee against any potential adversaries, our experiments prioritize evaluating the utility of DP-TKNN-Shapley rather than its privacy properties. However, for readers interested in understanding the efficacy of DP in safeguarding training data, we have included an evaluation of the proposed MI attack on DP-TKNN-Shapley in Appendix B.2.2, where it shows a significant drop in attack performance compared to non-DP version.*

## 6.1 Computational Efficiency

We evaluate the runtime efficiency of TKNN-Shapley in comparison to KNN-Shapley (see Appendix E.2 for details as well as more experiments). We choose a range of training data sizes $N$, and compare the runtime of both methods at each $N$. As demonstrated in Figure 2, TKNN-Shapley achieves better computational efficiency than KNN-Shapley across all training data sizes. In particular, TKNN-Shapley is around 30% faster than KNN-Shapley for large $N$. This shows the computational advantage of TKNN-Shapley mentioned in Section 4.2.

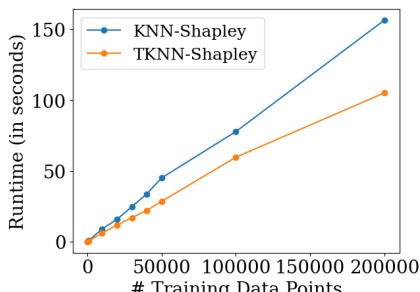

Figure 2: Runtime comparison of TKNN-Shapley and KNN-Shapley across various training data sizes $N$. The plot shows the average runtime based on 5 independent runs. Experiments were conducted with an AMD 64-Core CPU Processor.

## 6.2 Discerning Data Quality

In this section, we evaluate the performance of both DP and non-DP version of TKNN-Shapley in discerning data quality. **Tasks:** We evaluate the performance of TKNN-Shapley on two standard tasks: mislabeled data detection and noisy data detection, which are tasks that are often used for evaluating the performance of data valuation techniques in prior works [39, 60]. Since mislabeled/noisy data often negatively affect the model performance, it is desirable to assign low values to these data points. In the experiment of mislabeled (or noisy) data detection, we randomly choose 10% of the data points and flip their labels (or add strong noise to their features). **Datasets:** We conduct our experiments on a diverse set of 13 datasets, where 11 of them have been used in previous data valuation studies [39, 60]. Additionally, we experiment on 2 NLP datasets (AG News [65] and DBPedia [2]) that have been rarely used in the past due to their high-dimensional nature and the significant computational resources required. **Settings & Hyperparameters of TKNN-/KNN-Shapley:** for both TKNN/KNN-Shapley, we use the popular cosine distance as the distance measure [53], which is always bounded in $[-1, +1]$. Throughout all experiments, we use $\tau = -0.5$ and $K = 5$ for TKNN-/KNN-Shapley, respectively, as we found the two choices consistently work well across all datasets. We conduct ablation studies on the choice of hyperparameters in Appendix E.

### 6.2.1 Experiment for Private Setting

**Baselines:** **(1) DP-KNN-Shapley without subsampling.** Recall from Section 3.1, the original KNN-Shapley has a large global sensitivity. Nevertheless, we can still use Gaussian mechanism to privatize it based on its global sensitivity bound. We call this approach as DP-KNN-Shapley. **(2) DP-KNN-Shapley with subsampling.** Recall from Section 3.1, it is computationally expensive to incorporate subsampling techniques for DP-KNN-Shapley (detailed in Appendix B.3.2). For instance, subsampled DP-KNN-Shapley with subsampling rate $q = 0.01$ generally takes **30×longer time** compared with non-subsampled counterpart. Nevertheless, we still compare with subsampled DP-KNN-Shapley for completeness. These two baselines are detailed in Appendix B.4. We also note that an unpublished manuscript [67] proposed a DP version of Data Shapley. However, their DP guarantee requires a hard-to-verify assumption of uniform stability (see Appendix A). Hence, we do not compare with [67].

**Results:** We evaluate the privacy-utility tradeoff of DP-TKNN-Shapley. Specifically, for a fixed $\delta$, we examine the AUROC of mislabeled/noisy data detection tasks at different values of $\varepsilon$, where $\varepsilon$ is adjusted by changing the magnitude of Gaussian noise. In the experiment, we set the subsampling rate $q = 0.01$ for TKNN-Shapley and subsampled KNN-Shapley. Table 2 shows the results on 3 datasets, and we defer the results for the rest of 10 datasets to Appendix E.3.2. As we can see, **DP-TKNN-Shapley shows a substantially better privacy-utility tradeoff compared to both DP-KNN-Shapley with/without subsampling.** In particular, DP-TKNN-Shapley maintains a high

| Dataset | $\varepsilon$ | Mislabeled Data Detection | | | Noisy Data Detection | | |
|---|---|---|---|---|---|---|---|
| | | DP-TKNN-Shapley (ours) | DP-KNN-Shapley (no subsampling) | DP-KNN-Shapley (with subsampling) | DP-TKNN-Shapley (ours) | DP-KNN-Shapley (no subsampling) | DP-KNN-Shapley (with subsampling) |
| **2dPlanes** | **0.1** | 0.883 (0.017) | 0.49 (0.024) | 0.733 (0.011) | 0.692 (0.014) | 0.494 (0.023) | 0.615 (0.01) |
| | **0.5** | 0.912 (0.009) | 0.488 (0.022) | 0.815 (0.006) | 0.706 (0.004) | 0.494 (0.012) | 0.66 (0.004) |
| | **1** | 0.913 (0.009) | 0.504 (0.019) | 0.821 (0.005) | 0.705 (0.007) | 0.495 (0.011) | 0.665 (0.004) |
| **Phoneme** | **0.1** | 0.816 (0.011) | 0.5 (0.014) | 0.692 (0.011) | 0.648 (0.028) | 0.475 (0.042) | 0.566 (0.01) |
| | **0.5** | 0.826 (0.007) | 0.497 (0.011) | 0.738 (0.003) | 0.683 (0.014) | 0.536 (0.033) | 0.588 (0.004) |
| | **1** | 0.826 (0.005) | 0.486 (0.01) | 0.741 (0.002) | 0.685 (0.016) | 0.494 (0.071) | 0.59 (0.005) |
| **CPU** | **0.1** | 0.932 (0.007) | 0.49 (0.028) | 0.881 (0.005) | 0.805 (0.037) | 0.42 (0.074) | 0.709 (0.011) |
| | **0.5** | 0.946 (0.004) | 0.507 (0.029) | 0.928 (0.002) | 0.838 (0.007) | 0.472 (0.092) | 0.746 (0.003) |
| | **1** | 0.948 (0.002) | 0.512 (0.008) | 0.931 (0.002) | 0.839 (0.003) | 0.455 (0.079) | 0.748 (0.002) |

Table 2: Privacy-utility tradeoff of DP-TKNN-Shapley and DP-KNN-Shapley for mislabeled/noisy data detection task on 3 datasets we use (see Appendix E.3.2 for the results on the rest of 10 datasets). We set $\delta = 10^{-4}$ and show the AUROC at different privacy budgets $\varepsilon$; the higher the AUROC is, the better the method is. We show the standard deviation of AUROC across 5 independent runs in ().

AUROC even when $\varepsilon \approx 0.1$. The poor performance of DP-KNN-Shapley is due to its relatively high global sensitivity.

### 6.2.2 Experiment for Non-private Setting

**Baselines:** Our main baseline for comparison is KNN-Shapley. For completeness, we also compare with other classic, yet much less efficient data valuation techniques, such as Data Shapley [23], Data Banzhaf [60], and leave-one-out error (LOO) [35]. Due to space constraints, we only show the results for the famous Data Shapley here and defer other methods' results to Appendix E.3.3.

**Results:** We use AUROC as the performance metric on mislabeled/noisy data detection tasks. Due to space constraints, we defer the results for the task of noisy data detection to Appendix E. Table 3 shows the results for the task of mislabeled data detection across all 13 datasets we use. As we can see, **TKNN-Shapley shows a comparable performance as KNN-Shapley** across almost all datasets, demonstrating that TKNN-Shapley matches the effectiveness of KNN-Shapley in discerning data quality. Moreover, KNN-Shapley (and TKNN-Shapley) has a significantly better performance compared to Data Shapley, which is consistent with the observations in many existing studies [31, 52]. The poor performance of Data Shapley is attributable to the sample inefficiency and stochasticity during retraining [60].

| Dataset | TKNN-Shapley | KNN-Shapley | Data Shapley |
|---|---|---|---|
| **2dPlanes** | 0.919 (+0.006) | 0.913 | 0.552 |
| **Phoneme** | 0.826 (-0.047) | 0.873 | 0.525 |
| **CPU** | 0.946 (+0.014) | 0.932 | 0.489 |
| **Fraud** | 0.96 (-0.007) | 0.967 | 0.488 |
| **Creditcard** | 0.662 (+0.016) | 0.646 | 0.517 |
| **Apsfail** | 0.958 (+0.01) | 0.948 | 0.496 |
| **Click** | 0.572 (+0.004) | 0.568 | 0.474 |
| **Wind** | 0.889 (-0.007) | 0.896 | 0.469 |
| **Pol** | 0.871 (-0.057) | 0.928 | 0.512 |
| **MNIST** | 0.962 (-0.012) | 0.974 | - |
| **CIFAR10** | 0.957 (-0.034) | 0.991 | - |
| **AG News** | 0.956 (-0.015) | 0.971 | - |
| **DBPedia** | 0.981 (-0.01) | 0.991 | - |

Table 3: AUROC scores of TKNN/KNN-Shapley and Data Shapley for mislabeled data detection tasks on various datasets. The higher the AUROC score is, the better the method is. In the column for TKNN-Shapley, we highlight its performance difference with KNN-Shapley in (). The results for Data Shapley on the last 4 datasets are omitted due to computational constraints.

## 7 Conclusion

In this work, we uncover the inherent privacy risks associated with data value scores and introduce TKNN-Shapley, a privacy-friendly alternative to the widely-used KNN-Shapley. We demonstrate that TKNN-Shapley outperforms KNN-Shapley in terms of computational efficiency, and is as good as discerning data quality. Moreover, the privatized version of TKNN-Shapley significantly surpasses the naively privatized KNN-Shapley on tasks such as mislabeled data detection.

**Future Work. (1) Privacy risks of data revelation to central server:** in this work, we assume the existence of a trusted central server, and we do not consider the privacy risks associated with revealing individuals' data to the central server. Future work should consider integrating secure multi-party computation (MPC) techniques to mitigate this risk [59]. MPC can allow the computation of KNN-Shapley without revealing individual data to the central server, thereby preserving privacy. We envision an end-to-end privacy-preserving data valuation framework that combines both DP and MPC. **(2) Impact of Randomization on Payment Fairness:** the incorporation of differential privacy necessarily adds a degree of randomness to the data value scores. This randomization could potentially impact the fairness of payments to data providers [3]. The influence of this randomness, and its potential implications for payment fairness, remain areas for further investigation.

## Acknowledgments

This work was supported in part by the National Science Foundation under grants CNS-2131938, CNS-1553437, CNS-1704105, CNS-2048091, IIS-2312794, IIS-2313130, OAC-2239622, the ARL's Army Artificial Intelligence Innovation Institute (A2I2), the Office of Naval Research Young Investigator Award, the Army Research Office Young Investigator Prize, Schmidt DataX award, Princeton E-ffiliates Award, Amazon-Virginia Tech Initiative in Efficient and Robust Machine Learning, the Commonwealth Cyber Initiative, a Google PhD Fellowship, and a Princeton's Gordon Y. S. Wu Fellowship. We are grateful to anonymous reviewers at NeurIPS for their valuable feedback.

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

# A  Extended Related Work

**KNN-Shapley & Its Applications.**   The Shapley value is known for being computationally expensive. Fortunately, [29] found that computing the Data Shapley for K-Nearest Neighbors (KNN), one of the most classic yet still popular ML algorithms, is surprisingly easy and efficient. To the best of our knowledge, unweighted KNN is the *only* commonly-used ML model for which the exact Data Shapley can be efficiently computed, referred to as 'KNN-Shapley'. Owing to its superior computational efficiency and effectiveness in distinguishing data quality, KNN-Shapley has become one of the most popular and practical data valuation techniques. In the realm of ML research, for instance, [24] extends KNN-Shapley to active learning, [57] employs it in a continual learning setting, [43, 42] utilize KNN-Shapley for removing confusing samples in NLP applications, and [9] adopts KNN-Shapley for data valuation in semi-supervised learning. Furthermore, KNN-Shapley has also proven to be highly practical in real-life applications: [52] demonstrates that KNN-Shapley is the *only* practical data valuation technique for valuing large amounts of healthcare data, and [33] builds the first data debugging system for end-to-end ML pipelines based on KNN-Shapley.

**Remark 5** (KNN-Shapley vs General Data Shapley). *In comparison to the work of general Data Shapley [23] (including Beta Shapley [39] as well as Data Banzhaf [60]), KNN-Shapley may have the following differences: (**1**) KNN-Shapley focuses on KNN classifiers. As a result, the applicability of KNN-Shapley scores as a proxy of data points' value with respect to other ML models may not be straightforward. However, it is noteworthy that KNN is asymptotically Bayes optimal, implying that KNN-Shapley scores can be justified as a proxy for the data's value relative to the best possible model, i.e., the Bayes classifier, under certain asymptotic conditions. (**2**) For high-dimensional data, such as images, KNN-Shapley requires a public model to first map the original data into data embeddings, and evaluates the value of these embeddings rather than the original data. While this is indeed a constraint in certain scenario, it is important to recognize that utilizing a publicly available foundation model to convert original data into embeddings, followed by the fine-tuning of the model's last layer, has become a common practice. Therefore, in many situations, it might be more desirable to evaluate the value of data embeddings instead of the original data.*

**Privacy and Data Valuation.**   Few studies in the literature consider the privacy risks of data valuation. [59] explores a scenario where a trusted server does not exist, and different data holders collaboratively compute each other's Shapley values without actually examining the data holder's data points, utilizing Multi-party Computation (MPC) techniques. The privacy risks addressed in [59] are orthogonal to those in our paper, and we can combine both techniques for end-to-end privacy protection. Another orthogonal line of works [45, 20, 10, 32] studies the scenario of a central platform collecting private data from privacy-aware agents and offering a differentially private statistic computed from the submitted data as a service in return. Agents consider the privacy costs and benefits of obtaining the statistic when deciding whether to participate and reveal their data truthfully. [46] proposes a privacy attack on the Shapley value for feature attribution, while in this work we study the privacy risks when using the Shapley value for data valuation.

The unpublished manuscript by [67] is the work most closely related to ours. [67] explores how to make the Shapley values of a data point to be differentially private against the rest of the dataset. However, instead of focusing on KNN-Shapley, they study the privatization of the less practical, retrain-based Data Shapley [23, 30]. Furthermore, their algorithm is highly restrictive in that the differential privacy guarantee relies on the "uniform stability" assumption, which is not verifiable for modern learning algorithms such as neural networks. Moreover, while [67] argues the "uniform stability" assumption holds for Logistic regression on bounded data domain, it is unclear whether the uniform stability assumption still holds when the Logistic regression is trained by SGD (which may involve training stochasticity and early stopping). In addition, [67] does not release the implementation. Hence, in our experiment, we do not compare with it.

**Remark 6** (Brief background for the privacy risks in releasing aggregated statistics). *Since 1998, researchers have observed that a lot of seemingly benevolent aggregate statistics of a dataset can be used to reveal sensitive information about individuals [55]. A classic example is Netflix Prize fiasco, where the researchers show that an anonymized dataset can leak many sensitive information about individuals [49]. Dinur and Nissim [12] proved that "revealing too many statistics too accurately leads to data privacy breach". A great amount of discussion and practical realization of these privacy attacks on aggregated statistics can be found in [19]. In 2020, the US Census Bureau used these privacy attacks to justify its use of differential privacy.*

*KNN-Shapley score for an individual is one kind of aggregated statistic that depends on the rest of the dataset. Hence, KNN-Shapley score intrinsically reveals private information about the rest of the dataset (where we use membership inference attack as a concrete example in our paper). In addition, when users collude, their KNN-shapley values can be combined to make joint inferences about the rest of the dataset.*

# B Details about KNN-Shapley, its Privacy Risks & Challenges of Privatization

## B.1 Full version of KNN-Shapley

KNN-Shapley was originally proposed in [29] and was later refined in [61]. Specifically, [61] considers the KNN's utility function formula (2) we present in the main text. Here is the main result of [61]:

**Theorem 7** (KNN-Shapley [61]). *Consider the utility function in (2). Given a validation data point $z^{(\mathrm{val})} = (x^{(\mathrm{val})}, y^{(\mathrm{val})})$ and a distance metric $d(\cdot, \cdot)$, if we sort the training set $D = \{z_i = (x_i, y_i)\}_{i=1}^{N}$ according to $d(x_i, x^{(\mathrm{val})})$ in ascending order, then the Shapley value of each data point $\phi_{z_i}^{KNN}$ corresponding to utility function $v_{z^{(\mathrm{val})}}^{KNN}$ can be computed recursively as follows:*

$$\phi_{z_N}^{KNN} = \frac{\mathbb{1}[N \geq 2]}{N} \left( \mathbb{1}[y_N = y^{(\mathrm{val})}] - \frac{\sum_{i=1}^{N-1} \mathbb{1}[y_i = y^{(\mathrm{val})}]}{N-1} \right) \left( \sum_{j=1}^{\min(K,N)-1} \frac{1}{j+1} \right) + \frac{1}{N} \left( \mathbb{1}[y_N = y^{(\mathrm{val})}] - \frac{1}{C} \right)$$

$$\phi_{z_i}^{KNN} = \phi_{z_{i+1}}^{KNN} + \frac{\mathbb{1}[y_i = y^{(\mathrm{val})}] - \mathbb{1}[y_{i+1} = y^{(\mathrm{val})}]}{N-1} \left[ \sum_{j=1}^{\min(K,N)} \frac{1}{j} + \frac{\mathbb{1}[N \geq K]}{K} \left( \frac{\min(i,K) \cdot (N-1)}{i} - K \right) \right]$$

*where $C$ denotes the number of classes for the classification task.*

### B.1.1 Older version of KNN-Shapley from [29]

Prior to the current formulation of KNN-Shapley introduced by [61], [29] proposed an older version of the algorithm. There is only a small distinction between the updated version by [61] and the older version from [29]. Specifically, the difference between the versions proposed by [61] and [29] lies in the utility function considered when computing the Shapley value. [61] use the utility function (2) as presented in the main text, while the utility function used in [29] is slightly less interpretable:

$$v_{z^{(\mathrm{val})}}^{\texttt{KNN-OLD}}(S) := \frac{1}{K} \sum_{j=1}^{\min(K,|S|)} \mathbb{1}[y_{\pi^{(S)}(j;x^{(\mathrm{val})})} = y^{(\mathrm{val})}] \tag{5}$$

That is, (2) in the maintext divides the number of correct predictions by $\min(K, |S|)$, which can be interpreted as the likelihood of the soft-label KNN classifier predicting the correct label $y^{(\mathrm{val})}$ for $x^{(\mathrm{val})}$. On the other hand, the function above (5) divides the number of correct predictions by $K$, which is less interpretable when $|S| < K$.

Nevertheless, the main result in [29] shows the following:

**Theorem 8** (Older Version of KNN-Shapley from [29][2]). *Consider the utility function in (5). Given a validation data point $z^{(\mathrm{val})} = (x^{(\mathrm{val})}, y^{(\mathrm{val})})$ and a distance metric $d(\cdot, \cdot)$, if we sort the training set $D = \{z_i = (x_i, y_i)\}_{i=1}^{N}$ according to $d(x_i, x^{(\mathrm{val})})$ in ascending order, then the Shapley value of each data point $\phi_{z_i}^{KNN\text{-}OLD}$ corresponding to utility function $v_{z^{(\mathrm{val})}}^{KNN\text{-}OLD}$ can be computed recursively as follows:*

$$\phi_{z_N}^{KNN\text{-}OLD} = \frac{\mathbb{1}[y_N = y_{\mathrm{val}}]}{\max(K,N)}$$

$$\phi_{z_i}^{KNN\text{-}OLD} = \phi_{z_{i+1}}^{KNN\text{-}OLD} + \frac{\mathbb{1}[y_i = y_{\mathrm{val}}] - \mathbb{1}[y_{i+1} = y_{\mathrm{val}}]}{K} \frac{\min(K,i)}{i}$$

As we can see, both versions of KNN-Shapley have recursive forms. Since the utility functions they consider are very similar to each other except for the normalization term for small subsets, the two versions of KNN-Shapley have very close value scores in practice and as shown in the experiments of [61], the two versions of KNN-Shapley perform very similarly.

**Remark 7** (**The use of older version of KNN-Shapley for DP-related experiments**). *In the experiments, we use the more advanced version of the KNN-Shapley from [61] except for the DP-related experiments. This is because the global sensitivity of KNN-Shapley is difficult to derive, and we can only derive the global sensitivity of the older version of KNN-Shapley from [29] (as we will*

---

[2]We state a more generalized version which does not require $N \geq K$.

*show in Appendix B.3). Hence, we can only use the older version of KNN-Shapley from [29] as the baseline in DP-related experiments. It is important to note that for non-DP experiments, the performance of KNN-Shapley and its older variant is very close to each other, and **which version of the KNN-Shapley we use for non-DP experiments does not affect the final conclusion**.*

## B.2 Settings & Additional Experiments for Privacy Risks for KNN-Shapley

### B.2.1 Experiment for the changes of data value after eliminating nearby points

We first investigate the impact of removing a data point on the KNN-Shapley score of another data point that is close to the removed one. Specifically, we calculate the KNN-Shapley score for a chosen data point in the dataset, and then repeat the process after eliminating one of its nearby points from the dataset.

**Settings.** We use commonly used datasets in the past literature, and the details for data preprocessing can be found in Appendix E.1. We calculate the KNN-Shapley score for a randomly selected data point in the dataset, and then repeat the computation of the KNN-Shapley score after we eliminate its nearest neighbor from the dataset.

**Results.** The results for a variety of other datasets are shown in Figure 3. We can see a significant difference in the KNN-Shapley score of the investigated data point depending on whether the nearest data point has been removed. We remark that how the data value change (increases or decreases) when the nearest data point is excluded depends on the label of the nearest data point as well as the validation data being used.

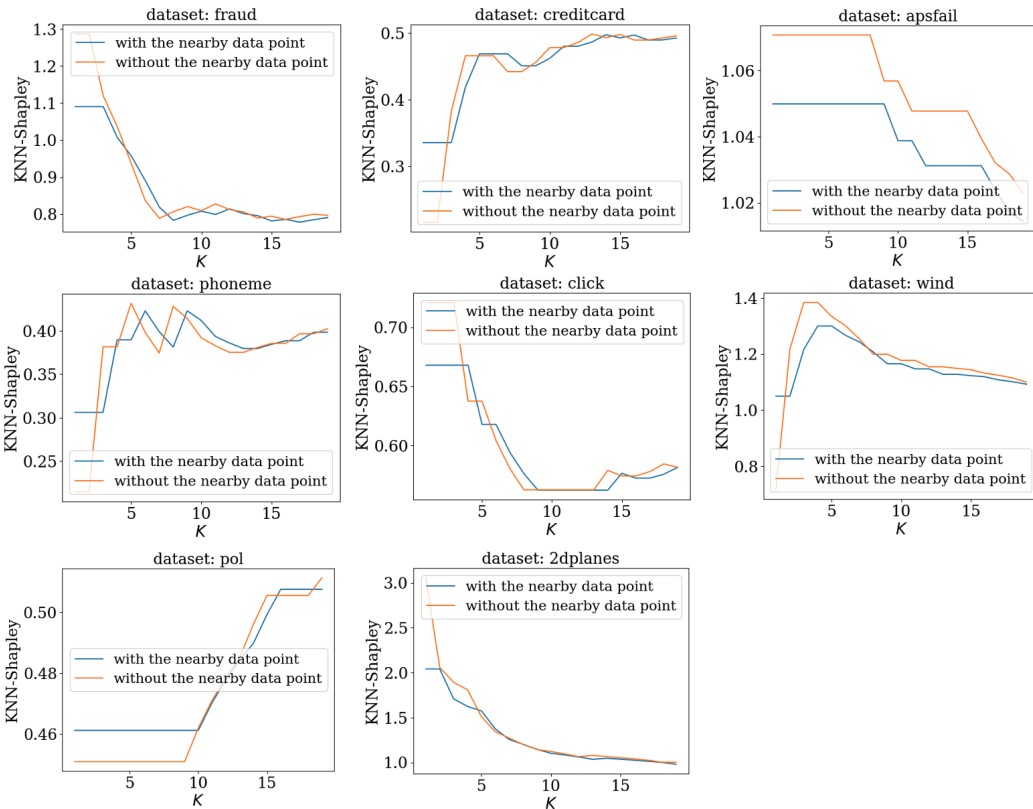

Figure 3: Curves of KNN-Shapley of a fixed data point as a function of $K$ (the hyperparameter for KNN), with and without a particular nearby data point in the training set.

For completeness, we also plot the same figures but for the older version of KNN-Shapley from [29], which shows similar results.

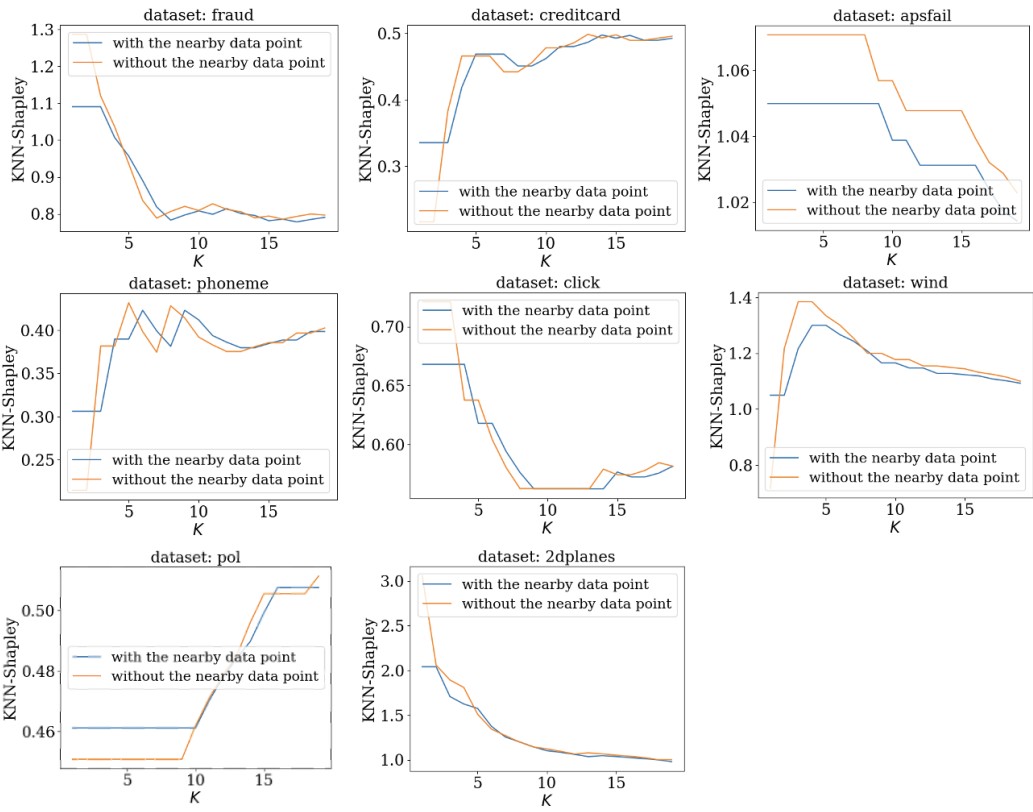

Figure 4: Curves of KNN-Shapley (old version from [29]) of a fixed data point as a function of $K$ (the hyperparameter for KNN), with and without a particular nearby data point in the training set.

### B.2.2 An Instantiation of Membership Inference Attack via Data Value Scores

The observation in Appendix B.2.1, demonstrates the potential for KNN-Shapley scores to leak membership information of other data points in the dataset. To further demonstrate such privacy risk, in this section we show an example of a privacy attack where an adversary could infer the presence/absence of certain data points in the dataset based on the variations in the KNN-Shapley scores, analogous to the classic membership inference attack on ML model [58]. The attack result further highlights the need for privacy protection when applying data valuation techniques.

**Remark 8** (Motivation of Membership Inference Attack). *Membership inference attack, i.e., confirming one's membership in a database, could pose significant privacy risks. For example, if a medical database is known to contain data from patients with specific conditions, this could disclose members' health status. We choose MI attack as our main example of the privacy risk as it has long been recognized as a fundamental privacy challenge across various domains, and such attacks have been extended well into the field of machine learning.*

**Remark 9.** *We stress that our goal here is the proof-of-concept, where we demonstrate that the data value scores can indeed serve as another channel of privacy leakage, and can indeed lead to the design of membership inference attacks. We do* not *claim any optimality of the attack we construct here. Designing the best membership inference attack on data value scores can be the topic of a single paper on its own, and is an interesting future work.*

**Membership Inference Attack via KNN-Shapley Scores (Algorithm 1).**    Our membership inference attack technique leverages KNN-Shapley scores to detect the presence or absence of specific data points in a dataset. The design is analogous to the membership inference attack against ML models via the *likelihood ratio test* [8]. **Threat model:** The threat model we consider here is that the attacker can compute the value of any data point among any datasets (analogue to the setting of MIA against ML models where the attacker can train models on any datasets he/she constructs). Moreover,

**Algorithm 1 - Membership Inference Attack via KNN-Shapley Scores.** We collect data value scores of a copy of the target example on datasets with and without the target example, estimate mean and variance of the loss distributions, and compute a likelihood ratio test.

---

**Require:** dataset $D$, query example $z := (x, y)$, data distribution $\mathbb{D}$.

1: $\text{KNNSV}_{\text{in}} = \{\}$
2: $\text{KNNSV}_{\text{out}} = \{\}$
3: **for** $T$ times **do**
4:      $D_{\text{shadow}} \xleftarrow{\$} \mathbb{D}$ // Sample a shadow dataset
5:      $\text{KNNSV}_{\text{in}} \leftarrow \text{KNNSV}_{\text{in}} \cup \{\phi_z(D_{\text{shadow}} \cup z)\}$ // collect IN data value
6:      $\text{KNNSV}_{\text{in}} \leftarrow \text{KNNSV}_{\text{in}} \cup \{\phi_z(D_{\text{shadow}}\}$ // collect OUT data value
7: **end for**
8: $\mu_{\text{in}} \leftarrow \texttt{mean}(\text{KNNSV}_{\text{in}})$
9: $\mu_{\text{out}} \leftarrow \texttt{mean}(\text{KNNSV}_{\text{out}})$
10: $\sigma_{\text{in}}^2 \leftarrow \texttt{var}(\text{KNNSV}_{\text{in}})$
11: $\sigma_{\text{out}}^2 \leftarrow \texttt{var}(\text{KNNSV}_{\text{out}})$
12: $\phi_{\text{obs}} \leftarrow \phi_z(D)$ // query target data value
13: **Return** $\Lambda = \dfrac{p(\phi_{\text{obs}} \mid \mathcal{N}(\mu_{\text{in}}, \sigma_{\text{in}}^2))}{p(\phi_{\text{obs}} \mid \mathcal{N}(\mu_{\text{out}}, \sigma_{\text{out}}^2))}$

---

the attacker can craft the data point it owns, send it to the server and obtain the data value score of its own data point.

The attack goes as follows: firstly, we create a shadow dataset $D_{\text{shadow}}$ by randomly sampling from the data distribution $\mathbb{D}$. $D_{\text{shadow}}$ serves as the function of $D_{-z}$ in the maintext. We repeat this sampling process $T$ times, where $T$ is a predefined number of iterations. For each iteration, we calculate the KNN-Shapley scores of a query example $z = (x, y)$ both when it is included in the shadow dataset (IN) and when it is excluded (OUT). We thus collect sets of IN and OUT scores for the query example. Upon collecting all these scores, we calculate their respective mean and variance. Finally, we query the server about KNN-Shapley score of the query example $z$ on the actual dataset $D_{-z}$, which we refer to as the *target data value* (i.e., the server takes a copy the target data point as the data it holds, and send to the central server). We perform a likelihood ratio test using the distributions of the IN and OUT scores. This test involves comparing the probability of the observed target data value given the Normal distribution of the IN scores with the probability of the same given the Normal distribution of the OUT scores.

**Intuition of the attack.** The intuition of the attack is as follows: if the target data point is in the dataset, then if the attacker makes a copy of the target data point and send it to the server, the server effectively has two exactly the same data point, which will result in a lower value of each data point. On the other hand, if the target data point is a non-member, then the data value queried by the attacker will be higher.

**Experiment Settings** For each dataset we experiment on, we select 200 data points (members) as the private dataset held by the central server. We pick another 200 data points which serve as non-members. Moreover, we leverage another 400 data points where we can subsample "shadow dataset" in Algorithm 1. We set the number of shadow datasets we sample as 32.

**Results on KNN-Shapley and TKNN-Shapley.** Table 1 in the main paper shows the AUROC score of the attack results on KNN-Shapley. For completeness, we additionally conducted the experiment of the proposed MIA against (non-private) TKNN-Shapley and the results are shown in Table 4. As we can see, our MIA attack can achieve a detection performance that is better than the random guess (0.5) for most of the settings. On some datasets, the attack performance can achieve $> 0.7$ AUROC. This demonstrates that privacy leakage in data value scores can indeed lead to non-trivial privacy attacks.

**Results on DP-TKNN-Shapley.** To directly demonstrate how DP-TKNN-Shapley can mitigate the privacy risk, we additionally conducted the experiment of evaluating the proposed MI attack on DP-TKNN-Shapley (see Table 5 and 6). Compared with the result on non-private TKNN-Shapley, we can see that the overall attack performance drops to around 0.5 (the performance of random guess).

The result shows that DP-TKNN-Shapley is indeed very effective against membership inference attacks.

We stress again that the point of this experiment is not claiming that the membership inference attack that we developed here is an optimal MI attack on data value scores; instead, this is a proof-of-concept for the claim that data value scores can leak private information about other data points in the dataset, and we instantiate a possible privacy attack that exploits this privacy risks. We believe it is interesting future work to improve the attack performance as well as explore other possible threat models.

| | $\tau = -0.9$ | $\tau = -0.8$ | $\tau = -0.7$ | $\tau = -0.6$ | $\tau = -0.5$ | $\tau = -0.4$ | $\tau = -0.3$ | $\tau = -0.2$ | $\tau = -0.1$ |
|---|---|---|---|---|---|---|---|---|---|
| **2DPlanes** | 0.549 | 0.799 | 0.738 | 0.688 | 0.53 | 0.665 | 0.6 | 0.558 | 0.628 |
| **Phoneme** | 0.777 | 0.679 | 0.736 | 0.673 | 0.704 | 0.692 | 0.588 | 0.522 | 0.5 |
| **CPU** | 0.75 | 0.672 | 0.638 | 0.635 | 0.512 | 0.518 | 0.58 | 0.545 | 0.508 |
| **Fraud** | 0.752 | 0.529 | 0.577 | 0.594 | 0.558 | 0.715 | 0.678 | 0.7 | 0.645 |
| **Creditcard** | 0.55 | 0.501 | 0.664 | 0.685 | 0.59 | 0.56 | 0.597 | 0.552 | 0.52 |
| **Apsfail** | 0.506 | 0.529 | 0.608 | 0.574 | 0.558 | 0.507 | 0.569 | 0.571 | 0.526 |
| **Click** | 0.718 | 0.56 | 0.545 | 0.6 | 0.568 | 0.735 | 0.535 | 0.56 | 0.52 |
| **Wind** | 0.528 | 0.65 | 0.7 | 0.585 | 0.585 | 0.58 | 0.568 | 0.562 | 0.632 |
| **Pol** | 0.772 | 0.62 | 0.748 | 0.715 | 0.6 | 0.592 | 0.59 | 0.532 | 0.672 |

Table 4: Results of the AUROC of our MI attack on TKNN-Shapley.

| | $\tau = -0.9$ | $\tau = -0.8$ | $\tau = -0.7$ | $\tau = -0.6$ | $\tau = -0.5$ | $\tau = -0.4$ | $\tau = -0.3$ | $\tau = -0.2$ | $\tau = -0.1$ |
|---|---|---|---|---|---|---|---|---|---|
| **2DPlanes** | 0.51 | 0.474 | 0.488 | 0.518 | 0.494 | 0.51 | 0.479 | 0.466 | 0.478 |
| **Phoneme** | 0.502 | 0.505 | 0.512 | 0.506 | 0.5 | 0.484 | 0.552 | 0.538 | 0.482 |
| **CPU** | 0.468 | 0.489 | 0.486 | 0.51 | 0.523 | 0.524 | 0.468 | 0.466 | 0.527 |
| **Fraud** | 0.479 | 0.474 | 0.492 | 0.513 | 0.48 | 0.484 | 0.485 | 0.484 | 0.578 |
| **Creditcard** | 0.518 | 0.495 | 0.511 | 0.522 | 0.501 | 0.483 | 0.448 | 0.521 | 0.535 |
| **Apsfail** | 0.484 | 0.476 | 0.471 | 0.492 | 0.486 | 0.5 | 0.493 | 0.434 | 0.454 |
| **Click** | 0.492 | 0.491 | 0.49 | 0.484 | 0.488 | 0.482 | 0.544 | 0.452 | 0.434 |
| **Wind** | 0.52 | 0.496 | 0.514 | 0.492 | 0.53 | 0.488 | 0.504 | 0.522 | 0.43 |
| **Pol** | 0.491 | 0.503 | 0.51 | 0.496 | 0.492 | 0.488 | 0.458 | 0.523 | 0.487 |

Table 5: Results of the AUROC of our MIA attack on DP-TKNN-Shapley ($\varepsilon = 0.5$).

| | $\tau = -0.9$ | $\tau = -0.8$ | $\tau = -0.7$ | $\tau = -0.6$ | $\tau = -0.5$ | $\tau = -0.4$ | $\tau = -0.3$ | $\tau = -0.2$ | $\tau = -0.1$ |
|---|---|---|---|---|---|---|---|---|---|
| **2DPlanes** | 0.5 | 0.523 | 0.513 | 0.534 | 0.518 | 0.492 | 0.484 | 0.445 | 0.556 |
| **Phoneme** | 0.507 | 0.529 | 0.524 | 0.472 | 0.476 | 0.488 | 0.543 | 0.512 | 0.47 |
| **CPU** | 0.492 | 0.487 | 0.504 | 0.485 | 0.473 | 0.489 | 0.453 | 0.49 | 0.508 |
| **Fraud** | 0.508 | 0.501 | 0.502 | 0.506 | 0.497 | 0.501 | 0.496 | 0.436 | 0.53 |
| **Creditcard** | 0.482 | 0.494 | 0.522 | 0.505 | 0.505 | 0.495 | 0.494 | 0.498 | 0.429 |
| **Apsfail** | 0.472 | 0.498 | 0.499 | 0.486 | 0.488 | 0.48 | 0.521 | 0.545 | 0.441 |
| **Click** | 0.502 | 0.529 | 0.501 | 0.508 | 0.518 | 0.496 | 0.545 | 0.47 | 0.456 |
| **Wind** | 0.533 | 0.539 | 0.478 | 0.504 | 0.493 | 0.498 | 0.529 | 0.499 | 0.419 |
| **Pol** | 0.498 | 0.513 | 0.509 | 0.51 | 0.482 | 0.492 | 0.489 | 0.52 | 0.528 |

Table 6: Results of the AUROC of our MIA attack on DP-TKNN-Shapley ($\varepsilon = 1.0$).

## B.3 Challenges in making KNN-Shapley being differentially private

In this section, we give more details of the inherent difficulties in making the KNN-Shapley $\phi_{z_i}^{\texttt{KNN}}(D_{-z_i})$ (i.e., Theorem 7) to be differentially private.

### B.3.1 Large global sensitivity

We find it very challenging to tightly bound the global sensitivity of $\phi_{z_i}^{\texttt{KNN}}(D_{-z_i})$. Moreover, we show that the global sensitivity of $\phi_{z_i}^{\texttt{KNN}}(D_{-z_i})$ can significantly exceed the magnitude of $\phi_{z_i}^{\texttt{KNN}}$ by showing a lower bound. Specifically, we prove that the global sensitivity bound is at least around $O(1)$ by constructing a specific pair of neighboring datasets.

**Theorem 9** (Lower Bound for the global sensitivity of $\phi_z^{\texttt{KNN}}(D_{-z})$). *For a data point $z = (x, y)$ and validation data point $z^{(\text{val})} = (x^{(\text{val})}, y^{(\text{val})})$, denote the global sensitivity of $\phi_z^{\texttt{KNN}}(D_{-z}; z^{(\text{val})})$ as*

$\Delta(\phi_z^{KNN}; z^{(\text{val})}) := \sup_{D_{-z} \sim D'_{-z}} \left| \phi_z^{KNN}(D_{-z}; z^{(\text{val})}) - \phi_z^{KNN}(D'_{-z}; z^{(\text{val})}) \right|$. *We have*

$$\Delta(\phi_z^{KNN}) \geq \frac{1}{2} \left( \mathbb{1}[y = y^{(\text{val})}] - 1/C \right)$$

*where $C$ is the number of classes for the corresponding classification task.*

*Proof.* The proof idea is to construct two neighboring datasets: $D = \{z_2\}, D' = \{z_1, z_2\}$ where $z_1 = (x_1, y_1), z_2 = (x_2, y_2)$ are two data points and we let $y_1 = y^{(\text{val})}$. Moreover, we let $d(z_1, z^{(\text{val})}) \leq d(z_2, z^{(\text{val})})$. From KNN-Shapley's formula in Theorem 7, we have

$$\phi_{z_2}^{KNN}(D) = \mathbb{1}[y_2 = y^{(\text{val})}] - 1/C$$

and if $K \geq 2$, we have

$$\phi_{z_2}^{KNN}(D') = \frac{1}{4} \left( \mathbb{1}[y_2 = y^{(\text{val})}] - \mathbb{1}[y_1 = y^{(\text{val})}] \right) + \frac{1}{2} \left( \mathbb{1}[y_2 = y^{(\text{val})}] - 1/C \right)$$

, and if $K = 1$, we have

$$\phi_{z_2}^{KNN}(D') = \frac{1}{2} \left( \mathbb{1}[y_2 = y^{(\text{val})}] - 1/C \right)$$

Since $\mathbb{1}[y_1 = y^{(\text{val})}] = 1$ (our condition), we have

$$\left| \phi_{z_2}^{KNN}(D) - \phi_{z_2}^{KNN}(D') \right| \geq \frac{1}{2} \left( \mathbb{1}[y_2 = y^{(\text{val})}] - 1/C \right)$$

$\square$

The above theorem tells us that the global sensitivity for KNN-Shapley is at the order of $O(1)$. On the other hand, we can see from the formula of KNN-Shapley in Theorem 7 that the magnitude $\phi_z^{KNN}$ for many of the data points $z$ is at the order of $O(1/N)$. Hence, if we apply the Gaussian mechanism and add random noise proportional to the global sensitivity bound, the resulting privatized data value score could substantially deviate from its non-private counterpart, thereby compromising the utility of the privatized data value scores.

**Tight Global Sensitivity for the older version of KNN-Shapley from [29].** As we said earlier, it is hard to bound the global sensitivity for $\phi_z^{KNN}$. Moreover, even if we are able to bound the global sensitivity, the bound will highly likely be large compared with the magnitude of $\phi_z^{KNN}$, as we can see from Theorem 9.

In order to find a reasonable baseline for comparing with DP-TKNN-Shapley, we consider the older version of the KNN-Shapley developed in [29], where we show that its global sensitivity can be *tightly* bounded (but still large).

**Theorem 10** (Global sensitivity of $\phi_z^{KNN-OLD}(D_{-z})$ from [29]). *For a data point $z = (x, y)$ and validation data point $z^{(\text{val})} = (x^{(\text{val})}, y^{(\text{val})})$, denote the global sensitivity of $\phi_z^{KNN-OLD}(D_{-z}; z^{(\text{val})})$ as $\Delta(\phi_z^{KNN-OLD}; z^{(\text{val})}) := \sup_{D_{-z} \sim D'_{-z}} \left| \phi_z^{KNN-OLD}(D_{-z}; z^{(\text{val})}) - \phi_z^{KNN-OLD}(D'_{-z}; z^{(\text{val})}) \right|$. We have*

$$\Delta(\phi_z^{KNN-OLD}; z^{(\text{val})}) \leq \frac{1}{K(K+1)}$$

*Proof.* For any dataset $D = D_{-z} \cup \{z\}$, we sort the data points according to the distance to $x^{(\text{val})}$, and we denote $z_j$ for the $j$th closest data point to $z^{(\text{val})}$. WLOG, suppose $z_i := z$. We first write out the non-recursive expression for the older KNN-Shapley $\phi_{z_i}^{KNN-OLD}$:

$$\phi_{z_i}^{KNN-OLD} = \frac{\mathbb{1}[y_N = y_{\text{val}}]}{\max(K, N)} + \sum_{j=i}^{N-1} \frac{\mathbb{1}[y_j = y_{\text{val}}] - \mathbb{1}[y_{j+1} = y_{\text{val}}]}{K} \frac{\min(K, j)}{j}$$

If $N \leq K$, then we have $\phi_{z_i}^{KNN-OLD} = \frac{\mathbb{1}[y_i = y^{(\text{val})}]}{K}$ for all $i$ (i.e., no privacy leakage in this case).

If $N > K$, there are two cases:

**Case 1:** $i \geq K$, then we have

$$\phi_{z_i}^{\text{KNN-OLD}} = \frac{\mathbb{1}[y_i = y^{(\text{val})}]}{i} - \sum_{j=i+1}^{N} \frac{1}{(j-1)j}\mathbb{1}[y_j = y^{(\text{val})}]$$

Hence, if we add/remove a data point $z_j = (x_j, y_j)$ s.t. $d(x_j, x^{(\text{val})}) \geq d(x_i, x^{(\text{val})})$, we have

$$\left|\phi_{z_i}^{\text{KNN-OLD}}(D_{-z_i}) - \phi_{z_i}^{\text{KNN-OLD}}(D'_{-z_i})\right| \leq \frac{1}{(j-1)j} \leq \frac{1}{K(K+1)}$$

If we add a data point $z_j = (x_j, y_j)$ s.t. $d(x_j, x^{(\text{val})}) < d(x_i, x^{(\text{val})})$, we have

$$\left|\phi_{z_i}^{\text{KNN-OLD}}(D_{-z_i}) - \phi_{z_i}^{\text{KNN-OLD}}(D'_{-z_i})\right|$$

$$= \left|\left(\frac{\mathbb{1}[y_i = y^{(\text{val})}]}{i} - \sum_{j=i+1}^{N} \frac{1}{(j-1)j}\mathbb{1}[y_j = y^{(\text{val})}]\right) - \left(\frac{\mathbb{1}[y_i = y^{(\text{val})}]}{i+1} - \sum_{j=i+2}^{N} \frac{1}{(j-1)j}\mathbb{1}[y_j = y^{(\text{val})}]\right)\right|$$

$$= \left|\left(\frac{1}{i} - \frac{1}{i+1}\right)\mathbb{1}[y_i = y^{(\text{val})}] - \sum_{j=i+1}^{N}\left(\frac{1}{(j-1)j} - \frac{1}{(j+1)j}\right)\mathbb{1}[y_j = y^{(\text{val})}]\right|$$

$$\leq \frac{1}{i(i+1)}$$

$$\leq \frac{1}{K(K+1)}$$

When $i \geq K+1$, the sensitivity analysis for remove a data point $z_j = (x_j, y_j)$ s.t. $d(x_j, x^{(\text{val})}) < d(x_i, x^{(\text{val})})$ is similar to the analysis above where we also have

$$\left|\phi_{z_i}^{\text{KNN-OLD}}(D_{-z_i}) - \phi_{z_i}^{\text{KNN-OLD}}(D'_{-z_i})\right| \leq \frac{1}{K(K+1)}$$

**Case 2:** $i < K$, then we have

$$\phi_{z_i}^{\text{KNN-OLD}} = \frac{\mathbb{1}[y_i = y^{(\text{val})}]}{K} - \sum_{j=K+1}^{N} \frac{1}{(j-1)j}\mathbb{1}[y_j = y^{(\text{val})}]$$

By a similar analysis, we can also show that

$$\left|\phi_{z_i}^{\text{KNN-OLD}}(D_{-z_i}) - \phi_{z_i}^{\text{KNN-OLD}}(D'_{-z_i})\right| \leq \frac{1}{K(K+1)}$$

The only remaining case that we haven't discussed yet is when $i = K$, and we remove a data point $z_j = (x_j, y_j)$ s.t. $d(x_j, x^{(\text{val})}) < d(x_i, x^{(\text{val})})$. In this case, we have

$$\left|\phi_{z_i}^{\text{KNN-OLD}}(D_{-z_i}) - \phi_{z_i}^{\text{KNN-OLD}}(D'_{-z_i})\right|$$

$$= \left|\left(\frac{\mathbb{1}[y_i = y^{(\text{val})}]}{K} - \sum_{j=K+1}^{N} \frac{1}{(j-1)j}\mathbb{1}[y_j = y^{(\text{val})}]\right) - \left(\frac{\mathbb{1}[y_i = y^{(\text{val})}]}{K} - \sum_{j=K+1}^{N} \frac{1}{(j-1)j}\mathbb{1}[y_j = y^{(\text{val})}]\right)\right|$$

$$= 0 \leq \frac{1}{K(K+1)}$$

$\square$

We stress that the bound $\frac{1}{K(K+1)}$ is tight. For example, for the case where $i = K$, then if we add another data point $z^* = (x^*, y^*)$ s.t.

$$d(x_i, x^{(\text{val})}) = d(x_K, x^{(\text{val})}) \leq d(x^*, x^{(\text{val})}) \leq d(x_{K+1}, x^{(\text{val})})$$

the change of the value $\phi_{z_i}^{\text{KNN-OLD}}$ will be $\frac{1}{K(K+1)}$.

### B.3.2 Difficulty (computational challenge) in incorporating subsampling technique

"Privacy amplification by subsampling" [4] is a technique where the subsampling of a dataset amplifies the privacy guarantees due to the reduced probability of an individual's data being included. Being able to incorporate such a technique is often important for achieving a decent privacy-utility tradeoff. However, the recursive nature of KNN-Shapley computation makes it hard to incorporate the subsampling techniques.

Specifically, recall that in practical data valuation scenarios, it is often desirable to compute the data value scores for *all of* $z_i \in D$. To apply the subsampling technique, we first need to create a subsampled dataset and compute the KNN-Shapley for the target data point $z_i$, i.e., we need to compute $\phi_z^{\text{KNN}}(\text{sample}(D_{-z}))$. The subsampled dataset is usually constructed by sampling each data point independently with a probability $q$ (this is usually referred to as Poisson subsampling [4]). If we view $q$ as a constant, then the computation of $\phi_z^{\text{KNN}}(\text{sample}(D_{-z}))$ requires a runtime of $\widetilde{O}(N)$ for the computation of *each* $\phi_{z_i}^{\text{KNN}}$. This results in a final runtime of $\widetilde{O}(N^2)$ for the computation of *all* $(\phi_{z_i}^{\text{KNN}})_{z_i \in D}$, which is a significant increase in computational demand compared to the non-private KNN-Shapley. The recursive nature of KNN-Shapley computation significantly complicates the attempt of improving the computational efficiency. That is, it is not clear how to reuse the subsampled dataset to compute the KNN-Shapley score for those that are not sampled. Therefore, it seems that an $\widetilde{O}(N^2)$ runtime is necessary if we want to incorporate the subsampling technique.

## B.4 Baseline for experiments in Section 6.2.1

**DP-KNN-Shapley.** Given such an upper bound for the global sensitivity of $\phi_z^{\text{KNN-OLD}}$, we can use Gaussian mechanism (Theorem 4) to privatize $\phi_z^{\text{KNN-OLD}}$, i.e., we compute $\widehat{\phi}_z^{\text{KNN-OLD}}(D_{-z}) := \phi_z^{\text{KNN-OLD}}(D_{-z}) + \mathcal{N}\left(0, \frac{1}{K(K+1)}\right)$. We note that such a bound is still not satisfactory as the magnitude $\phi_z^{\text{KNN-OLD}}$ for many of the data points $z$ is at the order of $O(1/N)$. Nevertheless, this is a reasonable baseline (if it is not the only one) that we can use for comparison in DP-related experiments.

**DP-KNN-Shapley with subsampling.** Despite the high computational cost associated with the incorporation of the subsampling technique, we still compare our approach with this computationally intensive baseline for completeness. More specifically, we compute $\widehat{\phi}_z^{\text{KNN-OLD}}(D_{-z}) := \phi_z^{\text{KNN-OLD}}(\text{sample}(D_{-z})) + \mathcal{N}\left(0, \left(\frac{1}{K(K+1)}\right)^2\right)$.

# C  Details for TKNN-Shapley

## C.1  Consistency Result for TKNN

For a data point $x$ and a threshold $\tau$, we denote

$$S_{x,\tau} := \{x' | d(x, x') \leq \tau\}$$

the ball of radius $\tau$ centered at $x$. Recall that the prediction rule of TKNN when given a training set $D$ is

$$m_D(x; \tau) := \begin{cases} 0 & |\text{NB}_{x,\tau}(D)| = 0 \\ \frac{1}{|\text{NB}_{x,\tau}(D)|} \sum_{x' \in \text{NB}_{x,\tau}(D)} m(x') & |\text{NB}_{x,\tau}(D)| > 0 \end{cases}$$

**Theorem 11.** *Suppose $m$ is the target function that is Lipschitz on $\text{supp}(\mu)$ where $\mu$ is the probability measure of data distribution. As $n \to \infty$, if $\tau_n \to 0$ and $n\mu(S_{x,\tau}) \to \infty$ for all $x \in \text{supp}(\mu)$, then*

$$\lim_{n \to \infty} \mathbb{E}_{x \sim \mu, D_n \sim \mu^n} \left[ (m_{D_n}(x) - m(x))^2 \right] = 0$$

*Proof.*

$$\mathbb{E}_{x, D_n}[(m_{D_n}(x) - m(x))^2]$$
$$= \mathbb{E}_{x, D_n}[(m_{D_n}(x) - m(x))^2 | |\text{NB}_{x,\tau_n}(D_n)| > 0] \Pr_{x, D_n}[|\text{NB}_{x,\tau_n}(D_n)| > 0]$$
$$+ \mathbb{E}_{x, D_n}[(0 - m(x))^2 | |\text{NB}_{x,\tau_n}(D_n)| = 0] \Pr_{x, D_n}[|\text{NB}_{x,\tau_n}(D_n)| = 0]$$

$$\lim_{n \to \infty} \Pr_{x, D_n}[|\text{NB}_{x,\tau_n}(D_n)| = 0] = \lim_{n \to \infty} \mathbb{E}_x \left[ \Pr_{D_n}[|\text{NB}_{x,\tau_n}(D_n)| = 0] \right]$$
$$= \lim_{n \to \infty} \mathbb{E}_x \left[ (1 - \mu(S_{x,\tau_n}))^n \right]$$
$$\leq \lim_{n \to \infty} \mathbb{E}_x \left[ \frac{1}{1 + n\mu(S_{x,\tau_n})} \right]$$
$$= 0$$

Suppose $m$ has Lipschitz constant $L$, i.e., for any pair of data points $x, x' \in \text{supp}(\mu)$, we have

$$|m(x) - m(x')| \leq L \|x - x'\|$$

$$\mathbb{E}_{x, D_n} \left[ (m_{D_n}(x) - m(x))^2 | |\text{NB}_{x,\tau_n}(D_n)| > 0 \right]$$

$$= \mathbb{E}_{x, D_n} \left[ \left( \frac{1}{|\text{NB}_{x,\tau_n}(D_n)|} \sum_{x' \in \text{NB}_{x,\tau_n}(D_n)} (m(x') - m(x)) \right)^2 | |\text{NB}_{x,\tau_n}(D_n)| > 0 \right]$$

$$\leq \mathbb{E}_{x, D_n} \left[ \frac{1}{|\text{NB}_{x,\tau_n}(D_n)|} \sum_{x' \in \text{NB}_{x,\tau_n}(D_n)} (m(x') - m(x))^2 | |\text{NB}_{x,\tau_n}(D_n)| > 0 \right]$$

$$\leq \mathbb{E}_{x, D_n} \left[ \frac{1}{|\text{NB}_{x,\tau_n}(D_n)|} \sum_{x' \in \text{NB}_{x,\tau_n}(D_n)} (m(x') - m(x))^2 | |\text{NB}_{x,\tau_n}(D_n)| > 0 \right]$$

$$\leq \mathbb{E}_{x, D_n} \left[ \frac{1}{|\text{NB}_{x,\tau_n}(D_n)|} \sum_{x' \in \text{NB}_{x,\tau_n}(D_n)} L^2 \|x - x'\|^2 | |\text{NB}_{x,\tau_n}(D_n)| > 0 \right]$$

$$\leq \mathbb{E}_{x, D_n} \left[ \frac{1}{|\text{NB}_{x,\tau_n}(D_n)|} \sum_{x' \in \text{NB}_{x,\tau_n}(D_n)} L^2 \tau_n^2 | |\text{NB}_{x,\tau_n}(D_n)| > 0 \right]$$

$$= L^2 \tau_n^2$$

which $\to 0$ as $n \to \infty$. $\qquad\square$

## C.2 Proofs for TKNN-Shapley

Given a dataset $S$ and a test data point $(x^{(\mathrm{val})}, y^{(\mathrm{val})})$, let $\mathtt{NB}_{x^{(\mathrm{val})}, \tau}(S) := \{(x,y)|(x,y) \in S, d(x, x^{(\mathrm{val})}) \leq \tau\}$ the nearest neighbor of $x^{(\mathrm{val})}$ that is within a pre-specified threshold $\tau$. Recall that in this case, the utility function of soft-label TKNN classifier becomes

$$v(S; (x^{(\mathrm{val})}, y^{(\mathrm{val})})) := \begin{cases} 1/C & |\mathtt{NB}_{x^{(\mathrm{val})}, \tau}(S)| = 0 \\ \frac{1}{|\mathtt{NB}_{x^{(\mathrm{val})}, \tau}(S)|} \sum_{(x,y) \in \mathtt{NB}_{x^{(\mathrm{val})}, \tau}(S)} \mathbb{1}[y = y^{(\mathrm{val})}] & |\mathtt{NB}_{x^{(\mathrm{val})}, \tau}(S)| > 0 \end{cases}$$

where $C$ is the number of classes for the classification task, and $1/C$ is the random guess accuracy.

**Semivalue.** The class of data values that satisfy all the Shapley axioms except efficiency is called *semivalues*. It was originally studied in the field of economics and recently proposed to tackle the data valuation problem [39]. Unlike the Shapley value, semivalues are *not* unique. The following theorem by the seminal work of [13] shows that every semivalue of a player $z$ (in our case the player is a data point) can be expressed as the weighted average of marginal contributions $v(S \cup \{z\}) - v(S)$ across different subsets $S \subseteq N \setminus \{z\}$.

**Theorem 12** (Representation of Semivalue [13]). *A value function $\phi_{\mathrm{semi}}$ is a semivalue, if and only if, there exists a* weight function $w : [N] \to \mathbb{R}$ *such that* $\sum_{k=1}^{N} \binom{N-1}{k-1} w(k) = N$ *and the value function $\phi$ can be expressed as follows:*

$$\phi_{z_i}(D_{-z_i}; v, w) := \frac{1}{N} \sum_{k=1}^{N} w(k) \sum_{\substack{S \subseteq D_{-z_i}, \\ |S| = k-1}} [v(S \cup \{z\}) - v(S)] \tag{6}$$

Semivalues subsume both the Shapley value and the Banzhaf value with $w_{\mathrm{shap}}(k) = \binom{N-1}{k-1}^{-1}$ and $w_{\mathrm{banz}}(k) = \frac{N}{2^{N-1}}$, respectively. In the following, we first discuss the general semivalue for soft-label TKNN classifier with a weight function $w$. We then plug in the specific weight function for the Shapley value.

Our goal is to derive $\phi_{z_i}(D_{-z_i}; v, w)$, the semivalue of $z_i = (x_i, y_i)$ with a weight function $w$ when using soft-label TKNN classifier, i.e.,

$$\phi_{z_i}(D_{-z_i}; v, w) := \frac{1}{N} \sum_{k=1}^{N} w(k) \sum_{\substack{S \subseteq D_{-z_i}, \\ |S| = k-1}} [v(S \cup \{z_i\}) - v(S)]$$

Consider a validation data point $(x^{(\mathrm{val})}, y^{(\mathrm{val})})$, and recall that $v(S) := v(S; (x^{(\mathrm{val})}, y^{(\mathrm{val})}))$. Denote

$$\mathbf{c}_{x^{(\mathrm{val})}, \tau} = \mathbf{c}_{x^{(\mathrm{val})}, \tau}(D_{-z_i}) := 1 + \left|\mathtt{NB}_{x^{(\mathrm{val})}, \tau}(D_{-z_i})\right| \qquad (1 + \# \text{ neighbors of } x^{(\mathrm{val})} \text{ in } D_{-z_i})$$

$$\mathbf{c}_{z^{(\mathrm{val})}, \tau}^{(+)} = \mathbf{c}_{z^{(\mathrm{val})}, \tau}^{(+)}(D_{-z_i}) := \sum_{(x,y) \in \mathtt{NB}_{x^{(\mathrm{val})}, \tau}(D_{-z_i})} \mathbb{1}[y = y^{(\mathrm{val})}] \qquad (\# \text{ same-label neighbors in } D_{-z_i})$$

**Case 1:** if $\left\|x_i - x^{(\mathrm{val})}\right\| > \tau$, we have $v(S \cup z_i) - v(S) = 0$ for any $S$, hence $\phi_{z_i}(D_{-z_i}; v, w) = 0$.

**Case 2:** if $\left\|x_i - x^{(\mathrm{val})}\right\| \leq \tau$, we have the following:

$$v(S \cup z_i) - v(S) =$$

$$\begin{cases} \frac{\mathbb{1}[y_i = y^{(\mathrm{val})}]}{1 + |\mathtt{NB}_{x^{(\mathrm{val})}, \tau}(S)|} + \left(\frac{1}{1 + |\mathtt{NB}_{x^{(\mathrm{val})}, \tau}(S)|} - \frac{1}{|\mathtt{NB}_{x^{(\mathrm{val})}, \tau}(S)|}\right) \sum_{(x,y) \in \mathtt{NB}_{x^{(\mathrm{val})}, \tau}(S)} \mathbb{1}[y = y_{\mathrm{val}}] & |\mathtt{NB}_{x^{(\mathrm{val})}, \tau}(S)| > 0 \\ \mathbb{1}[y_i = y^{(\mathrm{val})}] - 1/C & |\mathtt{NB}_{x^{(\mathrm{val})}, \tau}(S)| = 0 \end{cases}$$

This means that the marginal contribution on $S$ only depends on the subset $\text{NB}_{x^{(\text{val})},\tau}(S) \subseteq S$.
Plugging in the above equation for $v(S \cup z_i) - v(S)$, we have

$$\phi_{z_i}\left(D_{-z_i}; v, w\right)$$

$$= \frac{1}{N} \sum_{k=0}^{N-1} w(k+1) \sum_{\substack{S \subseteq D_{-z_i}, \\ |S|=k}} [v(S \cup \{z_i\}) - v(S)]$$

$$= \frac{1}{N} \sum_{k=0}^{N-1} w(k+1) \sum_{\substack{S_0 \subseteq \text{NB}_{x^{(\text{val})},\tau}(D_{-z_i}), \\ |S_0| \leq k}} \binom{N - \mathbf{c}_{x^{(\text{val})},\tau}}{k - |S_0|} [v(S_0 \cup \{z_i\}) - v(S_0)]$$

$$= \frac{1}{N} \sum_{k=0}^{N-1} w(k+1) \sum_{j=0}^{k} \sum_{\substack{S_0 \subseteq \text{NB}_{x^{(\text{val})},\tau}(D_{-z_i}), \\ |S_0|=j}} \binom{N - \mathbf{c}_{x^{(\text{val})},\tau}}{k - j} [v(S_0 \cup \{z_i\}) - v(S_0)]$$

$$= \frac{1}{N} \sum_{k=0}^{N-1} w(k+1) \left\{ \underbrace{\sum_{j=1}^{k} \sum_{\substack{S_0 \subseteq \text{NB}_{x^{(\text{val})},\tau}(D_{-z_i}), \\ |S_0|=j}} \binom{N - \mathbf{c}_{x^{(\text{val})},\tau}}{k - j} \left[ \frac{\mathbb{1}[y_i = y^{(\text{val})}]}{1+j} + \left(\frac{1}{1+j} - \frac{1}{j}\right) \sum_{(x,y) \in S_0} \mathbb{1}[y = y^{(\text{val})}] \right]}_{(*)} \right.$$

$$\left. + \binom{N - \mathbf{c}_{x^{(\text{val})},\tau}}{k} \left[ \mathbb{1}[y_i = y^{(\text{val})}] - 1/C \right] \right\}$$

and

$$(*) = \sum_{j=1}^{k} \binom{N - \mathbf{c}_{x^{(\text{val})},\tau}}{k - j} \left[ \binom{\mathbf{c}_{x^{(\text{val})},\tau} - 1}{j} \frac{\mathbb{1}[y_i = y^{(\text{val})}]}{1+j} + \left(\frac{1}{1+j} - \frac{1}{j}\right) \sum_{\substack{S_0 \subseteq \text{NB}_{x^{(\text{val})},\tau}(D_{-z_i}) \\ |S_0|=j}} \sum_{(x,y) \in S_0} \mathbb{1}[y = y^{(\text{val})}] \right]$$

$$\sum_{\substack{S_0 \subseteq \text{NB}_{x^{(\text{val})},\tau}(D_{-z_i}) \\ |S_0|=j}} \sum_{(x,y) \in S_0} \mathbb{1}[y = y^{(\text{val})}] = \sum_{\ell=0}^{j} \ell \binom{\mathbf{c}_{z^{(\text{val})},\tau}^{(+)}}{\ell} \binom{\mathbf{c}_{x^{(\text{val})},\tau} - 1 - \mathbf{c}_{z^{(\text{val})},\tau}^{(+)}}{j - \ell}$$

$$= \mathbf{c}_{z^{(\text{val})},\tau}^{(+)} \sum_{\ell=1}^{j} \binom{\mathbf{c}_{z^{(\text{val})},\tau}^{(+)} - 1}{\ell - 1} \binom{\mathbf{c}_{x^{(\text{val})},\tau} - 1 - \mathbf{c}_{z^{(\text{val})},\tau}^{(+)}}{j - \ell}$$

$$= \mathbf{c}_{z^{(\text{val})},\tau}^{(+)} \binom{\mathbf{c}_{x^{(\text{val})},\tau} - 2}{j - 1}$$

Hence

$(*)$

$$= \sum_{j=1}^{k} \binom{N - \mathbf{c}_{x^{(\mathrm{val})},\tau}}{k-j} \left[ \binom{\mathbf{c}_{x^{(\mathrm{val})},\tau} - 1}{j} \frac{\mathbb{1}[y_i = y^{(\mathrm{val})}]}{1+j} + \mathbf{c}^{(+)}_{z^{(\mathrm{val})},\tau} \left( \frac{1}{1+j} - \frac{1}{j} \right) \binom{\mathbf{c}_{x^{(\mathrm{val})},\tau} - 2}{j-1} \right]$$

$$= \mathbb{1}[\mathbf{c}_{x^{(\mathrm{val})},\tau} \geq 2] \mathbb{1}[y_i = y^{(\mathrm{val})}] \sum_{j=1}^{k} \binom{N - \mathbf{c}_{x^{(\mathrm{val})},\tau}}{k-j} \binom{\mathbf{c}_{x^{(\mathrm{val})},\tau} - 1}{j} \frac{1}{1+j}$$

$$- \mathbf{c}^{(+)}_{z^{(\mathrm{val})},\tau} \sum_{j=1}^{k} \binom{N - \mathbf{c}_{x^{(\mathrm{val})},\tau}}{k-j} \binom{\mathbf{c}_{x^{(\mathrm{val})},\tau} - 2}{j-1} \frac{1}{j(j+1)}$$

$$= \mathbb{1}[\mathbf{c}_{x^{(\mathrm{val})},\tau} \geq 2] \mathbb{1}[y_i = y^{(\mathrm{val})}] \sum_{j=1}^{k} \binom{N - \mathbf{c}_{x^{(\mathrm{val})},\tau}}{k-j} \binom{\mathbf{c}_{x^{(\mathrm{val})},\tau}}{j+1} \frac{1}{\mathbf{c}_{x^{(\mathrm{val})},\tau}}$$

$$- \mathbf{c}^{(+)}_{z^{(\mathrm{val})},\tau} \sum_{j=1}^{k} \binom{N - \mathbf{c}_{x^{(\mathrm{val})},\tau}}{k-j} \binom{\mathbf{c}_{x^{(\mathrm{val})},\tau}}{j+1} \frac{\mathbb{1}[\mathbf{c}_{x^{(\mathrm{val})},\tau} \geq 2]}{\mathbf{c}_{x^{(\mathrm{val})},\tau}(\mathbf{c}_{x^{(\mathrm{val})},\tau} - 1)}$$

$$= \mathbb{1}[\mathbf{c}_{x^{(\mathrm{val})},\tau} \geq 2] \left( \frac{\mathbb{1}[y_i = y^{(\mathrm{val})}]}{\mathbf{c}_{x^{(\mathrm{val})},\tau}} - \frac{\mathbf{c}^{(+)}_{z^{(\mathrm{val})},\tau}}{\mathbf{c}_{x^{(\mathrm{val})},\tau}(\mathbf{c}_{x^{(\mathrm{val})},\tau} - 1)} \right) \sum_{j=1}^{k} \binom{N - \mathbf{c}_{x^{(\mathrm{val})},\tau}}{k-j} \binom{\mathbf{c}_{x^{(\mathrm{val})},\tau}}{j+1}$$

$$= \mathbb{1}[\mathbf{c}_{x^{(\mathrm{val})},\tau} \geq 2] \left( \frac{\mathbb{1}[y_i = y^{(\mathrm{val})}]}{\mathbf{c}_{x^{(\mathrm{val})},\tau}} - \frac{\mathbf{c}^{(+)}_{z^{(\mathrm{val})},\tau}}{\mathbf{c}_{x^{(\mathrm{val})},\tau}(\mathbf{c}_{x^{(\mathrm{val})},\tau} - 1)} \right) \underbrace{\left[ \binom{N}{k+1} - \binom{N - \mathbf{c}_{x^{(\mathrm{val})},\tau}}{k+1} - \mathbf{c}_{x^{(\mathrm{val})},\tau} \binom{N - \mathbf{c}_{x^{(\mathrm{val})},\tau}}{k} \right]}_{B(k)}$$

Hence,

$$\phi_{z_i}(D_{-z_i}; v, w) = \frac{1}{N} \sum_{k=0}^{N-1} w(k+1) \left\{ \mathbb{1}[\mathbf{c}_{x^{(\mathrm{val})},\tau} \geq 2] \left( \frac{\mathbb{1}[y_i = y^{(\mathrm{val})}]}{\mathbf{c}_{x^{(\mathrm{val})},\tau}} - \frac{\mathbf{c}^{(+)}_{z^{(\mathrm{val})},\tau}}{\mathbf{c}_{x^{(\mathrm{val})},\tau}(\mathbf{c}_{x^{(\mathrm{val})},\tau} - 1)} \right) B(k) \right.$$

$$\left. + \binom{N - \mathbf{c}_{x^{(\mathrm{val})},\tau}}{k} \left[ \mathbb{1}[y_i = y^{(\mathrm{val})}] - 1/C \right] \right\}$$

$$= \frac{\mathbb{1}[\mathbf{c}_{x^{(\mathrm{val})},\tau} \geq 2]}{N} \left( \frac{\mathbb{1}[y_i = y^{(\mathrm{val})}]}{\mathbf{c}_{x^{(\mathrm{val})},\tau}} - \frac{\mathbf{c}^{(+)}_{z^{(\mathrm{val})},\tau}}{\mathbf{c}_{x^{(\mathrm{val})},\tau}(\mathbf{c}_{x^{(\mathrm{val})},\tau} - 1)} \right) \sum_{k=0}^{N-1} w(k+1) B(k)$$

$$+ \frac{1}{N} \left[ \mathbb{1}[y_i = y^{(\mathrm{val})}] - 1/C \right] \sum_{k=0}^{N-1} w(k+1) \binom{N - \mathbf{c}_{x^{(\mathrm{val})},\tau}}{k}$$

### C.2.1 The Shapley value for TKNN (TKNN-Shapley)

**Theorem 13** (Full version of TKNN-Shapley). *Consider the utility function $v^{TKNN}_{z^{(\mathrm{val})}}$ in (3). Given a validation data point $z^{(\mathrm{val})} = (x^{(\mathrm{val})}, y^{(\mathrm{val})})$, the Shapley value $\phi^{TKNN}_{z_i}(v^{TKNN}_{z^{(\mathrm{val})}})$ of each training point $z_i = (x_i, y_i) \in D$ can be calculated as follows:*

$$\phi^{TKNN}_{z_i} = \begin{cases} \mathbb{1}[\mathbf{c}_{x^{(\mathrm{val})},\tau} \geq 2] A_1 A_2 + \frac{\mathbb{1}[y_i = y^{(\mathrm{val})}] - 1/C}{\mathbf{c}_{x^{(\mathrm{val})},\tau}} & d(x_i, x^{(\mathrm{val})}) \leq \tau \\ 0 & d(x_i, x^{(\mathrm{val})}) > \tau \end{cases} \tag{7}$$

*where $A_1 = \frac{\mathbb{1}[y_i = y^{(\mathrm{val})}]}{\mathbf{c}_{x^{(\mathrm{val})},\tau}} - \frac{\mathbf{c}^{(+)}_{z^{(\mathrm{val})},\tau}}{\mathbf{c}_{x^{(\mathrm{val})},\tau}(\mathbf{c}_{x^{(\mathrm{val})},\tau} - 1)}$, $A_2 = \sum_{k=0}^{\mathbf{c}} \left[ \frac{1}{k+1} - \frac{1}{k+1} \cdot \frac{\binom{\mathbf{c}-k}{\mathbf{c}_{x^{(\mathrm{val})},\tau}}}{\binom{\mathbf{c}+1}{\mathbf{c}_{x^{(\mathrm{val})},\tau}}} \right] - 1$,*

*and*

$$\mathbf{c} = \mathbf{c}(D_{-z_i}) := |D_{-z_i}| \qquad\qquad\qquad\qquad\qquad (\textit{size of } D_{-z_i})$$

$$\mathbf{c}_{x^{(\mathrm{val})},\tau} = \mathbf{c}_{x^{(\mathrm{val})},\tau}(D_{-z_i}) := 1 + \left| \mathit{NB}_{x^{(\mathrm{val})},\tau}(D_{-z_i}) \right| \qquad (1 + \textit{\# neighbors of } x^{(\mathrm{val})} \textit{ in } D_{-z_i})$$

$$\mathbf{c}^{(+)}_{z^{(\mathrm{val})},\tau} = \mathbf{c}^{(+)}_{z^{(\mathrm{val})},\tau}(D_{-z_i}) := \sum_{(x,y)\in \mathit{NB}_{x^{(\mathrm{val})},\tau}(D_{-z_i})} \mathbb{1}[y = y^{(\mathrm{val})}] \qquad (\textit{\# same-label neighbors in } D_{-z_i})$$

*and $C$ is the number of classes for the classification task, and $1/C$ is the random guess accuracy.*

**Runtime:** *the computation of all $(\phi^{TKNN}_{z_1}, \ldots, \phi^{TKNN}_{z_N})$ can be achieved in $O(N)$ runtime in total.*

**The Shapley value when using full validation set:** *The Shapley value corresponding to the utility function $v^{TKNN}_{D^{(\mathrm{val})}}$ can be calculated as $\phi^{TKNN}_{z_i}\left(v^{TKNN}_{D^{(\mathrm{val})}}\right) = \sum_{z^{(\mathrm{val})}\in D^{(\mathrm{val})}} \phi^{TKNN}_{z_i}\left(v^{TKNN}_{z^{(\mathrm{val})}}\right).$*

*Proof.* For the Shapley value, we have $w(k) = \binom{N-1}{k-1}^{-1}$.

$$\sum_{k=0}^{N-1} w(k+1)\cdot \binom{N}{k+1} = \sum_{k=0}^{N-1} \binom{N-1}{k}^{-1}\cdot\binom{N}{k+1}$$
$$= \sum_{k=0}^{N-1} \frac{N}{k+1}$$

$$\sum_{k=0}^{N-1} w(k+1)\cdot \binom{N-\mathbf{c}_{x^{(\mathrm{val})},\tau}}{k+1} = \sum_{k=0}^{N-1} \binom{N-1}{k}^{-1}\cdot\binom{N-\mathbf{c}_{x^{(\mathrm{val})},\tau}}{k+1}$$
$$= \sum_{k=0}^{N-1} \frac{k!(N-1-k)!}{(N-1)!}\frac{(N-\mathbf{c}_{x^{(\mathrm{val})},\tau})!}{(k+1)!(N-\mathbf{c}_{x^{(\mathrm{val})},\tau}-k-1)!}$$
$$= \frac{(N-\mathbf{c}_{x^{(\mathrm{val})},\tau})!}{(N-1)!}\sum_{k=0}^{N-1} \frac{1}{k+1}\frac{(N-1-k)!}{(N-\mathbf{c}_{x^{(\mathrm{val})},\tau}-k-1)!}$$
$$= \frac{(N-\mathbf{c}_{x^{(\mathrm{val})},\tau})!\,\mathbf{c}_{x^{(\mathrm{val})},\tau}!}{(N-1)!}\sum_{k=0}^{N-1} \frac{1}{k+1}\binom{N-1-k}{\mathbf{c}_{x^{(\mathrm{val})},\tau}}$$
$$= N\binom{N}{\mathbf{c}_{x^{(\mathrm{val})},\tau}}^{-1}\sum_{k=0}^{N-1} \frac{1}{k+1}\binom{N-1-k}{\mathbf{c}_{x^{(\mathrm{val})},\tau}}$$

$$\sum_{k=0}^{N-1} w(k+1)\cdot \binom{N-\mathbf{c}_{x^{(\mathrm{val})},\tau}}{k} = \sum_{k=0}^{N-1} \binom{N-1}{k}^{-1}\cdot\binom{N-\mathbf{c}_{x^{(\mathrm{val})},\tau}}{k}$$
$$= \frac{(N-\mathbf{c}_{x^{(\mathrm{val})},\tau})!}{(N-1)!}\sum_{k=0}^{N-1} \frac{(N-1-k)!}{(N-\mathbf{c}_{x^{(\mathrm{val})},\tau}-k)!}$$
$$= \binom{N-1}{\mathbf{c}_{x^{(\mathrm{val})},\tau}-1}^{-1}\sum_{k=0}^{N-1} \binom{N-1-k}{\mathbf{c}_{x^{(\mathrm{val})},\tau}-1}$$
$$= \binom{N-1}{\mathbf{c}_{x^{(\mathrm{val})},\tau}-1}^{-1}\binom{N}{\mathbf{c}_{x^{(\mathrm{val})},\tau}}$$
$$= \frac{N}{\mathbf{c}_{x^{(\mathrm{val})},\tau}}$$

$$\phi_{z_i}(D_{-z_i})$$

$$= \frac{\mathbb{1}[\mathbf{c}_{x^{(\mathrm{val})},\tau} \geq 2]}{N} \left( \frac{\mathbb{1}[y_i = y^{(\mathrm{val})}]}{\mathbf{c}_{x^{(\mathrm{val})},\tau}} - \frac{\mathbf{c}_{z^{(\mathrm{val})},\tau}^{(+)}}{\mathbf{c}_{x^{(\mathrm{val})},\tau}(\mathbf{c}_{x^{(\mathrm{val})},\tau} - 1)} \right) \sum_{k=0}^{N-1} \binom{N-1}{k}^{-1} B(k)$$

$$+ \frac{1}{N} \left[ \mathbb{1}[y_i = y^{(\mathrm{val})}] - 1/C \right] \sum_{k=0}^{N-1} \binom{N-1}{k}^{-1} \binom{N - \mathbf{c}_{x^{(\mathrm{val})},\tau}}{k}$$

$$= \mathbb{1}[\mathbf{c}_{x^{(\mathrm{val})},\tau} \geq 2] \left( \frac{\mathbb{1}[y_i = y^{(\mathrm{val})}]}{\mathbf{c}_{x^{(\mathrm{val})},\tau}} - \frac{\mathbf{c}_{z^{(\mathrm{val})},\tau}^{(+)}}{\mathbf{c}_{x^{(\mathrm{val})},\tau}(\mathbf{c}_{x^{(\mathrm{val})},\tau} - 1)} \right) \left[ \sum_{k=0}^{N-1} \left( \frac{1}{k+1} - \frac{1}{k+1} \frac{\binom{N-1-k}{\mathbf{c}_{x^{(\mathrm{val})},\tau}}}{\binom{N}{\mathbf{c}_{x^{(\mathrm{val})},\tau}}} \right) - 1 \right]$$

$$+ \left[ \mathbb{1}[y_i = y^{(\mathrm{val})}] - 1/C \right] \frac{1}{\mathbf{c}_{x^{(\mathrm{val})},\tau}}$$

$$= \mathbb{1}[\mathbf{c}_{x^{(\mathrm{val})},\tau} \geq 2] \left( \frac{\mathbb{1}[y_i = y^{(\mathrm{val})}]}{\mathbf{c}_{x^{(\mathrm{val})},\tau}} - \frac{\mathbf{c}_{z^{(\mathrm{val})},\tau}^{(+)}}{\mathbf{c}_{x^{(\mathrm{val})},\tau}(\mathbf{c}_{x^{(\mathrm{val})},\tau} - 1)} \right) \left[ \sum_{k=0}^{\mathbf{c}} \left( \frac{1}{k+1} - \frac{1}{k+1} \frac{\binom{\mathbf{c}-k}{\mathbf{c}_{x^{(\mathrm{val})},\tau}}}{\binom{\mathbf{c}+1}{\mathbf{c}_{x^{(\mathrm{val})},\tau}}} \right) - 1 \right]$$

$$+ \left[ \mathbb{1}[y_i = y^{(\mathrm{val})}] - 1/C \right] \frac{1}{\mathbf{c}_{x^{(\mathrm{val})},\tau}}$$

$$\square$$

### C.2.2 Efficient Computation of TKNN-Shapley

As we can see, all of the quantities that are needed to compute TKNN-Shapley value $(\mathbf{c}, \mathbf{c}_{x^{(\mathrm{val})},\tau}, \mathbf{c}_{z^{(\mathrm{val})},\tau}^{(+)})$ are simply counting queries on $D_{-z_i}$, and hence $\mathbf{C}_{z^{(\mathrm{val})}}(D_{-z_i})$ can be computed in $O(N)$ runtime for any $z_i \in D$. Since our goal is to release $\phi_{z_i}^{\mathrm{TKNN}}$ for *all* $z_i \in D$, we need to compute $\mathbf{C}_{z^{(\mathrm{val})}}(D_{-z_i})$ for *all* $z_i \in D$. A more efficient way for this case is to first compute $\mathbf{C}_{z^{(\mathrm{val})}}(D)$ on the full dataset $D$, and then we can compute each of $\mathbf{C}_{z^{(\mathrm{val})}}(D_{-z_i})$ by

$$\mathbf{c}(D_{-z_i}) = \mathbf{c}(D) - 1$$
$$\mathbf{c}_{x^{(\mathrm{val})},\tau}(D_{-z_i}) = \mathbf{c}_{x^{(\mathrm{val})},\tau}(D) - \mathbb{1}[z_i \in \mathrm{NB}_{x^{(\mathrm{val})},\tau}]$$
$$\mathbf{c}_{z^{(\mathrm{val})},\tau}^{(+)}(D_{-z_i}) = \mathbf{c}_{z^{(\mathrm{val})},\tau}^{(+)}(D) - \mathbb{1}[z_i \in \mathrm{NB}_{x^{(\mathrm{val})},\tau}]\mathbb{1}[y_i = y^{(\mathrm{val})}]$$

which can be computed in $O(1)$ runtime. Hence, we can compute TKNN-Shapley $\phi_{z_i}^{\mathrm{TKNN}}(v_{z^{(\mathrm{val})}}^{\mathrm{TKNN}})$ for *all* $z_i \in D$ within an overall computational cost of $O(N)$.

# D  Extra Details for DP-TKNN-Shapley

## D.1  Privacy Guarantee for DP-TKNN-Shapley

**Theorem 14** (Restatement of Theorem 6). *For any $z^{(\mathrm{val})} \in D^{(\mathrm{val})}$, releasing $\widehat{\phi}_{z_i}^{\mathtt{TKNN}}(v_{z^{(\mathrm{val})}}^{\mathtt{TKNN}}) := \phi_{z_i}^{\mathtt{TKNN}}\left[\widehat{\boldsymbol{C}}_{z^{(\mathrm{val})}}(D_{-z_i})\right]$ to data owner $i$ is $(\varepsilon, \delta)$-DP with $\sigma = \sqrt{3} \cdot \sqrt{\log(1.25/\delta)}/\varepsilon$.*

*Proof.* The Gaussian mechanism is known to satisfy $(\varepsilon, \delta)$-DP with $\sigma = \Delta \cdot \sqrt{2\log(1.25/\delta)}/\varepsilon$ [16], where $\Delta := \sup_{D \sim D'} \|f(D) - f(D')\|$ represents the global sensitivity of the underlying function $f$ in $\ell_2$ norm. In our case, $f$ calculates the 3-dimensional vector $\left[\mathbf{c}, \mathbf{c}_{x^{(\mathrm{val})}, \tau}, \mathbf{c}_{z^{(\mathrm{val})}, \tau}^{(+)}\right]$. As each training data point may alter both $\mathbf{c}_{x^{(\mathrm{val})}, \tau}$ and $\mathbf{c}_{z^{(\mathrm{val})}, \tau}^{(+)}$ by 1, the sensitivity $\Delta$ is $\sqrt{3}$. Since both $\widehat{\mathbf{c}}_{x^{(\mathrm{val})}, \tau}$ and $\widehat{\mathbf{c}}_{z^{(\mathrm{val})}, \tau}^{(+)}$ are privatized using the Gaussian mechanism, the privacy guarantee of $\widehat{\phi}_{z_i}$ is ensured by the post-processing property of differential privacy. $\qquad\square$

## D.2  Advantages of DP-TKNN-Shapley

### D.2.1  Computational Efficiency via Reusing Privatized Statistics

Recall that in practical data valuation scenarios, it is often desirable to compute the data value scores for *all of* $z_i \in D$. As we detailed in Section 4.2 and Appendix C.2.2, for TKNN-Shapley, such a computation only requires $O(N)$ runtime in total if we first compute $\mathbf{C}_{z^{(\mathrm{val})}}(D)$ and subsequently $\mathbf{C}_{z^{(\mathrm{val})}}(D_{-z_i})$ for each $z_i \in D$. In a similar vein, we can efficiently compute the DP variant $\widehat{\phi}_{z_i}^{\mathtt{TKNN}}$ for *all* $z_i \in D$. To do so, we first calculate $\widehat{\mathbf{C}}_{z^{(\mathrm{val})}}(D)$ and then, for each $z_i \in D$, we compute $\widehat{\mathbf{C}}_{z^{(\mathrm{val})}}(D_{-z_i})$ as follows:

$$\widehat{\mathbf{c}}(D_{-z_i}) = \widehat{\mathbf{c}}(D) - 1$$
$$\widehat{\mathbf{c}}_{x^{(\mathrm{val})}, \tau}(D_{-z_i}) = \widehat{\mathbf{c}}_{x^{(\mathrm{val})}, \tau}(D) - \mathbb{1}[z_i \in \mathtt{NB}_{x^{(\mathrm{val})}, \tau}]$$
$$\widehat{\mathbf{c}}_{z^{(\mathrm{val})}, \tau}^{(+)}(D_{-z_i}) = \widehat{\mathbf{c}}_{z^{(\mathrm{val})}, \tau}^{(+)}(D) - \mathbb{1}[z_i \in \mathtt{NB}_{x^{(\mathrm{val})}, \tau}]\mathbb{1}[y_i = y^{(\mathrm{val})}]$$

which can be computed in $O(1)$ runtime. Hence, we can compute DP-TKNN-Shapley $\widehat{\phi}_{z_i}^{\mathtt{TKNN}}(v_{z^{(\mathrm{val})}}^{\mathtt{TKNN}})$ for *all* $z_i \in D$ within an overall computational cost of $O(N)$. It is important to note that when releasing $\widehat{\phi}_{z_i}^{\mathtt{TKNN}}$ to individual $i$, the data point they hold, $z_i$, is not private to the individual themselves. Therefore, as long as $\widehat{\mathbf{C}}_{z^{(\mathrm{val})}}(D)$ is privatized, $\widehat{\mathbf{C}}_{z^{(\mathrm{val})}}(D_{-z_i})$ and hence $\widehat{\phi}_{z_i}^{\mathtt{TKNN}}$ also satisfy the same privacy guarantee due to DP's post-processing property.

### D.2.2  Collusion Resistance via Reusing Privatized Statistics

In Section 5.1, we consider the single Shapley value $\phi_{z_i}^{\mathtt{TKNN}}(D_{-z_i})$ as the function to privatize. However, when we consider the release of all Shapley values $\mathcal{M}(D) := (\widehat{\phi}_{z_1}^{\mathtt{TKNN}}(D_{-z_1}), \ldots, \widehat{\phi}_{z_N}^{\mathtt{TKNN}}(D_{-z_N}))$ as a whole mechanism, one can show that such a mechanism satisfies *joint differential privacy* (JDP) [34] if we reuse the privatized statistic $\widehat{\mathbf{C}}_{z^{(\mathrm{val})}}(D)$ for the release of all $\widehat{\phi}_{z_i}^{\mathtt{TKNN}}$. The consequence of satisfying JDP is that our mechanism is resilient against collusion among groups of individuals without any privacy degradation. That is, even if an arbitrary group of individuals in $[N] \setminus i$ colludes (i.e., shares their respective $\widehat{\phi}_{z_j}^{\mathtt{TKNN}}$ values within the group), the privacy of individual $i$ remains uncompromised. Our method also stands resilient in scenarios where a powerful adversary sends multiple data points and receives multiple value scores.

**Definition 15** (Joint Differential Privacy [34]). *For $\varepsilon, \delta \geq 0$, a randomized algorithm $\mathcal{M} : \mathbb{N}^{\mathcal{X}} \to \mathcal{Y}^N$ is $(\varepsilon, \delta)$-joint differentially private if for every possible pair of $z, z' \in \mathcal{X}$, for every $i \in [N]$, and for every subset of possible outputs $E \subseteq \mathcal{Y}^{N-1}$, we have*

$$\Pr_{\mathcal{M}}[\mathcal{M}(z \cup D_{-z})_{-i} \in E] \leq e^{\varepsilon} \Pr_{\mathcal{M}}[\mathcal{M}(z' \cup D_{-z})_{-i} \in E] + \delta$$

*where $\mathcal{M}_{-i}$ denotes the output of $\mathcal{M}$ that excludes the $i$th dimension.*

**Theorem 16.** *The mechanism of releasing of all noisy KNN-Shapley values*

$$\mathcal{M}(D) := (\widehat{\phi}_{z_1}^{TKNN}(D_{-z_1}), \ldots, \widehat{\phi}_{z_N}^{TKNN}(D_{-z_N}))$$

*satisfy $(\varepsilon, \delta)$-JDP if $\sigma = \sqrt{3} \cdot \sqrt{\log(1.25/\delta)}/\varepsilon$ and each $\phi_{z_i}^{TKNN}$ is computed via reusing the privatized* $\mathbf{C}_{z^{(\mathrm{val})}}$.

*Proof.* The proof relies on the "post-processing" property of differential privacy.

First of all, we select $\sigma = \sqrt{3} \cdot \sqrt{\log(1.25/\delta)}/\varepsilon$, ensuring that $\widehat{\mathbf{C}}_{z^{(\mathrm{val})}}(D)$ satisfies $(\varepsilon, \delta)$-DP.

In the context of DP-TKNN-shapley, we calculate $\widehat{\mathbf{C}}_{z^{(\mathrm{val})}}(D_{-z_i})$ for each $z_i$ as part of a post-processing step on $\widehat{\mathbf{C}}_{z^{(\mathrm{val})}}(D)$.

It is important to note that the released data value $\widehat{\phi}_{z_i}^{\mathrm{TKNN}}(D_{-z_i})$ is a function of $\widehat{\mathbf{C}}_{z^{(\mathrm{val})}}(D_{-z_i})$, and $\widehat{\mathbf{C}}_{z^{(\mathrm{val})}}(D_{-z_i})$ can be obtained as post-processing of $\widehat{\mathbf{C}}_{z^{(\mathrm{val})}}(D)$ given only the knowledge of $z_i$. Hence, we can express the probability of $\mathcal{M}(z_i, D_{-z_i})_{-i}$ belonging to a certain set $E$ as follows:

$$\Pr_{\mathcal{M}}[\mathcal{M}(z_i, D_{-z_i})_{-i} \in E] = \Pr_{\widehat{\mathbf{C}}_{z^{(\mathrm{val})}}(D)}[f_{D_{-z_i}}(\widehat{\mathbf{C}}_{z^{(\mathrm{val})}}(D)) \in E]$$

where

$$f_{D_{-z_i}}(\widehat{\mathbf{C}}_{z^{(\mathrm{val})}}(D))$$
$$= \left( \widehat{\phi}_{z_1}^{\mathrm{TKNN}}[\widehat{\mathbf{C}}_{z^{(\mathrm{val})}}(D_{-z_1})], \ldots, \phi_{z_{i-1}}^{\mathrm{TKNN}}[\widehat{\mathbf{C}}_{z^{(\mathrm{val})}}(D_{-z_{i-1}})], \phi_{z_{i+1}}^{\mathrm{TKNN}}[\widehat{\mathbf{C}}_{z^{(\mathrm{val})}}(D_{-z_{i+1}})], \ldots, \phi_{z_N}^{\mathrm{TKNN}}[\widehat{\mathbf{C}}_{z^{(\mathrm{val})}}(D_{-z_N})] \right)$$

We note that $f_{D_{-z_i}}$ is a function that does *not* depend on $z_i$.

Then, for any $z_i, z_i'$ and for any tuple of $D_{-z_i}$, denote $D := z_i \cup D_{-z_i}$ and $D' := z_i' \cup D_{-z_i}$, we have

$$\Pr_{\mathcal{M}}[\mathcal{M}(z_i, D_{-z_i})_{-i} \in E] = \Pr_{\widehat{\mathbf{C}}_{z^{(\mathrm{val})}}(D)}[f_{D_{-z_i}}(\widehat{\mathbf{C}}_{z^{(\mathrm{val})}}(D)) \in E]$$
$$= \Pr_{\widehat{\mathbf{C}}_{z^{(\mathrm{val})}}(D)}[\widehat{\mathbf{C}}_{z^{(\mathrm{val})}}(D) \in f_{D_{-z_i}}^{-1}(E)]$$
$$\leq e^{\varepsilon} \Pr_{\widehat{\mathbf{C}}_{z^{(\mathrm{val})}}(D')}[\widehat{\mathbf{C}}_{z^{(\mathrm{val})}}(D') \in f_{D_{-z_i}}^{-1}(E)] + \delta$$
$$= e^{\varepsilon} \Pr_{\widehat{\mathbf{C}}_{z^{(\mathrm{val})}}(D')}[f_{D_{-z_i}}(\widehat{\mathbf{C}}_{z^{(\mathrm{val})}}(D')) \in E] + \delta$$
$$= e^{\varepsilon} \Pr_{\mathcal{M}}[\mathcal{M}(z_i', D_{-z_i}) \in E] + \delta$$

where the first inequality is due to the $(\varepsilon, \delta)$-DP guarantee of $\widehat{\mathbf{C}}_{z^{(\mathrm{val})}}(D)$. $\square$

Hence, the privacy guarantee for our DP-TKNN-Shapley remains the same even if multiple attackers collude and share the received noisy data value scores with each other. As a comparison, the naive DP-KNN-Shapley which independently adds noise for each released data value score does not enjoy such powerful collusion resistance property.

### D.2.3 Privacy Amplification by Subsampling

In order to further boost the privacy guarantee, we incorporate the privacy amplification by sub-sampling technique [4]. Specifically, before any computation, we first sample each data point independently with a probability $q$ (this is usually referred to as Poisson subsampling). The differentially private quantities based on the subsampled dataset, denoted as $\mathrm{sample}(D)$, are calculated easily as follows:

$$\widehat{\mathbf{c}}(D) := \mathrm{round}\left(\mathbf{c}(\mathrm{sample}(D)) + \mathcal{N}\left(0, \sigma^2\right)\right)$$
$$\widehat{\mathbf{c}}_{x^{(\mathrm{val})}, \tau}(D) := \mathrm{round}\left(\mathbf{c}_{x^{(\mathrm{val})}, \tau}(\mathrm{sample}(D)) + \mathcal{N}\left(0, \sigma^2\right)\right)$$
$$\widehat{\mathbf{c}}_{z^{(\mathrm{val})}, \tau}^{(+)}(D) := \mathrm{round}\left(\mathbf{c}_{z^{(\mathrm{val})}, \tau}^{(+)}(\mathrm{sample}(D)) + \mathcal{N}\left(0, \sigma^2\right)\right)$$

The function `round` rounds a noisy value to the nearest integer. If the noisy value is out of range (e.g., $< 0$), then it is rounded to the nearest possible value within the valid range. By subsampling the data points, the privacy guarantee is amplified. This is because the subsampling process itself is a randomized operation that provides a level of privacy, making it harder to infer information about individual data points. When combined with the Gaussian mechanism, the overall privacy guarantee is improved.

The subsampling technique can be incorporated easily for DP-TKNN-Shapley as above, where the subsampling technique only improves the computational efficiency. As a comparison, incorporating subsampling technique for DP-KNN-Shapley will introduce a significantly higher computational cost, as we discussed in Appendix B.3.2.

### D.3 Differentially private release of $\phi_{z_i}^{\text{TKNN}}(v_{D^{(\text{val})}}^{\text{TKNN}})$ (Privacy Accounting)

Recall that the TKNN-Shapley corresponding to the full validation set $D^{(\text{val})}$ is $\phi_{z_i}^{\text{TKNN}}\left(v_{D^{(\text{val})}}^{\text{TKNN}}\right) = \sum_{z^{(\text{val})} \in D^{(\text{val})}} \phi_{z_i}^{\text{TKNN}}(D_{-z_i}; z^{(\text{val})})$. We can compute privatized $\phi_{z_i}^{\text{TKNN}}\left(v_{D^{(\text{val})}}^{\text{TKNN}}\right)$ by simply releasing $\phi_{z_i}^{\text{TKNN}}(D_{-z_i}; z^{(\text{val})})$ for all $z^{(\text{val})} \in D^{(\text{val})3}$. To better keep track of the overall privacy cost, we use the current state-of-the-art privacy accounting technique based on the notion of the Privacy Loss Random Variable (PRV) [17]. Here, we provide more details about the background of the composition of differential privacy and the privacy accounting techniques we use.

**Background: Composition of Differential Privacy ("Privacy Accounting").** In practice, multiple differentially private mechanisms may be applied to the same dataset, denoted as $\mathcal{M}(D) = \mathcal{M}_1 \circ \mathcal{M}_2(D) := (\mathcal{M}_1(D), \mathcal{M}_2(D))$.[4] Differential privacy offers strong composition guarantees, and the guarantees are derived by various composition theorems or privacy accounting techniques, including the basic composition theorem [14], advanced composition theorem [18], and Moments Accountant [1]. For example, the basic composition theorem states that if $\mathcal{M}_1$ is $(\varepsilon_1, \delta_1)$-DP and $\mathcal{M}_2$ is $(\varepsilon_2, \delta_2)$-DP, then the composition of $\mathcal{M}_1$ and $\mathcal{M}_2$ is $(\varepsilon_1 + \varepsilon_2, \delta_1 + \delta_2)$-DP. In our scenario, each individual mechanism is the release of $\widehat{\phi}_{z_i}^{\text{TKNN}}(D_{-z_i}; z^{(\text{val})})$ for each of $z^{(\text{val})} \in D^{(\text{val})}$, and we would like to keep track of the privacy loss of releasing all of them.

**Privacy Accounting Techniques based on Privacy Loss Random Variable (PRV).** To better keep track of the privacy cost, we use the current state-of-the-art privacy accounting based on the notion of the Privacy Loss Random Variable (PRV) [17]. The PRV accountant was introduced in [37] and later refined in [36, 25]. For any DP algorithm, one can easily compute its $(\varepsilon, \delta)$ privacy guarantee based on the distribution of its PRV. The key property of PRVs is that, under (adaptive) composition, they simply add up; the PRV $Y$ of the composition $\mathcal{M} = \mathcal{M}_1 \circ M_2 \circ \cdots \circ M_k$ is given by $Y = \sum_{i=1}^{k} Y_i$, where $Y_i$ is the PRV of $\mathcal{M}_i$. Therefore, one can then find the distribution of $Y$ by convolving the distributions of $Y_1, Y_2, \ldots, Y_k$. Prior works [36, 25] approximate the distribution of PRVs by truncating and discretizing them, then using the Fast Fourier Transform (FFT) to efficiently convolve the distributions. In our experiment, we use the FFT-based accountant from [25], the current state-of-the-art privacy accounting technique.

---

[3]Directly privatizing $\phi_{z_i}^{\text{TKNN}}\left(v_{D^{(\text{val})}}^{\text{TKNN}}\right)$ is very difficult. Releasing more privatized statistics for the ease of privacy analysis is common in DP, e.g., DP-SGD [1] releases all privatized gradients.

[4]Multiple DP mechanisms can be *adaptively* composed in the sense that the output of one mechanism can be used as an input to another mechanism, i.e., $\mathcal{M}(D) = \mathcal{M}_1 \circ \mathcal{M}_2(D) := (\mathcal{M}_1(D), \mathcal{M}_2(D, \mathcal{M}_1(D)))$.

# E   Additional Experiment Settings & Results

## E.1   Datasets

A comprehensive list of datasets and sources is summarized in Table 7. Similar to the existing data valuation literature [23, 39, 30, 60], we preprocess datasets for ease of training. For Fraud, Creditcard, and all datasets from OpenML, we subsample the dataset to balance positive and negative labels. For these datasets, if they have multi-class, we binarize the label by considering $\mathbb{1}[y = 1]$. For the image dataset MNIST, CIFAR10, we apply a ResNet50 [27] that is pre-trained on the ImageNet dataset as the feature extractor. This feature extractor produces a 1024-dimensional vector for each image. We employ sentence embedding models [54] to extract features for the text classification dataset AGNews and DBPedia, and the extracted features are 1024-dimensional vectors for each text instance. We perform L2 normalization on the extracted features as part of the pre-processing step.

The size of each dataset we use is shown in Table 7. For some of the datasets, we use a subset of the full set. We stress that we are using a *much larger* size of the dataset compared with prior data valuation literature [30, 39, 60] (these works usually only pick a very small subset of the full dataset, e.g., 2000 data points for binarized CIFAR10). The validation data size we use is 10% of the training data size.

| Dataset | Number of classes | Size of dataset | Source |
|---|---|---|---|
| MNIST | 10 | 50000 | [41] |
| CIFAR10 | 10 | 50000 | [38] |
| AGnews | 4 | 10000 | [65] |
| DBPedia | 14 | 10000 | [2] |
| Click | 2 | 2000 | https://www.openml.org/d/1218 |
| Fraud | 2 | 2000 | [11] |
| Creditcard | 2 | 2000 | [71] |
| Apsfail | 2 | 2000 | https://www.openml.org/d/41138 |
| Phoneme | 2 | 2000 | https://www.openml.org/d/1489 |
| Wind | 2 | 2000 | https://www.openml.org/d/847 |
| Pol | 2 | 2000 | https://www.openml.org/d/722 |
| CPU | 2 | 2000 | https://www.openml.org/d/761 |
| 2DPlanes | 2 | 2000 | https://www.openml.org/d/727 |

Table 7: A summary of datasets used in Section 6's experiments.

## E.2   Settings & Additional Experiments for Runtime Experiments

For the runtime comparison experiment in Section 6.1, we follow similar experiment settings from prior study [40] and use a synthetic binary classification dataset. To generate the synthetic dataset, we sample data points from a $d$-dimensional standard Gaussian distribution. The labels are assigned based on the sign of the sum of the two features.

We choose a range of training data sizes $N$, and compare the runtime of both TKNN/KNN-Shapley at each $N$. We use 100 validation data points. For Figure 2 in the maintext, the data dimension $d = 10$. Here, we show additional experiment results when $d = 100$. As we can see from Figure 5, again TKNN-Shapley achieves better computational efficiency than KNN-Shapley across all training data sizes, and is around 30% faster than KNN-Shapley for large $N$.

## E.3   Settings & Additional Experiments for Distinguishing Data Quality Experiments

### E.3.1   Details for Mislabel/Noisy Data Detection Experiment

In the experiment of mislabeled (or noisy) data detection, we randomly choose 10% of the data points and flip their labels (or add strong noise to their features). For mislabel data detection, we flip 10% of the labels by picking an alternative label from the rest of the classes uniformly at random. For noisy data detection, we add zero-mean Gaussian noise to data features, where the standard deviation of the

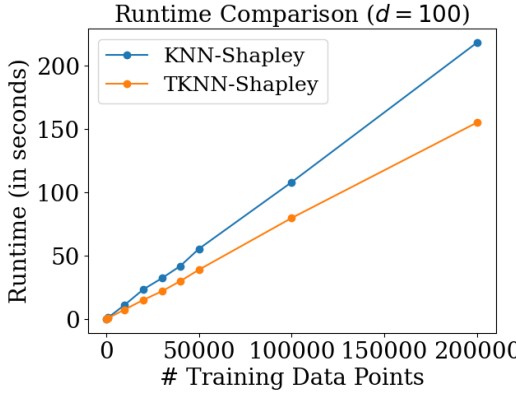

Figure 5: Runtime comparison of TKNN-Shapley and KNN-Shapley across various training data sizes $N$ when data dimension $d = 100$. The plot shows the average runtime based on 5 independent runs. Experiments were conducted with an AMD 64-Core CPU Processor.

Gaussian noise added to each feature dimension is equal to the average absolute value of the feature dimension across the full dataset.

### E.3.2   Additional Experiment Results for Private Setting

Table 8 provides a comprehensive comparison of DP-TKNN-Shapley and DP-KNN-Shapley. We conducted the experiments on DP-KNN-Shapley with subsampling on only half of the datasets due to the considerable computational cost (at least $30\times$ longer to execute compared to its non-subsampled counterpart). Additionally, because DP-KNN-Shapley injects Gaussian noise independently into the value score of each data holder, it does not have the property of being robust against collusion.

As evidenced by the results, DP-TKNN-Shapley consistently demonstrates a markedly superior privacy-utility tradeoff compared to DP-KNN-Shapley without subsampling across all datasets. Even when comparing with DP-KNN-Shapley with subsampling, DP-TKNN-Shapley still shows superior performance, and often significantly outshining this time-intensive and collusion-susceptible baseline. Notably, DP-TKNN-Shapley manages to maintain high AUROC values even when $\varepsilon \approx 0.1$. The low performance of DP-KNN-Shapley can be attributed to its relatively high global sensitivity, as we discussed in Appendix B.3.1.

### E.3.3   Additional Experiment Results for Non-private Setting

Here, we show full experimental results for various data valuation techniques in mislabel/noisy data detection scenarios, summarized in Tables 9 and 10. In line with the main text findings, TKNN-Shapley exhibits performance largely on par with KNN-Shapley across the majority of datasets, demonstrating that TKNN-Shapley effectively mirrors KNN-Shapley's ability to discern data quality. In addition, both KNN-Shapley and TKNN-Shapley significantly outperform traditional retrain-based data valuation techniques (LOO, Data Shapley, and Data Banzhaf), corroborating observations made in existing studies [31, 52]. The inferior performance of Data Shapley can be attributed to sample inefficiency and the inherent randomness during the retraining process [60].

**Settings for LOO, Data Shapley and Data Banzhaf.** Following the experiment settings in [23, 39] we use logistic regression as the ML model for these techniques. For Data Shapley, we use permutation sampling [48] for estimation. For Data Banzhaf, we use the maximum-sample-reuse (MSR) algorithm [60] for estimation. For each dataset, we train models on 10,000 subsets (which takes more than $1000\times$ longer time compared with both TKNN/KNN-Shapley).

### E.3.4   Ablation Study on the Choice of Threshold for TKNN-Shapley

KNN-Shapley is known for being effective on a wide range of choices of $K$ [29]. In this experiment, we study the impact of different choices of $\tau$ on the performance of TKNN-Shapley on the tasks of mislabeled/noisy data detection. The results are shown in Table 11 and 12. As we can see, the

| Dataset | $\varepsilon$ | Mislabeled Data Detection | | | Noisy Data Detection | | |
|---|---|---|---|---|---|---|---|
| | | DP-TKNN-Shapley | DP-KNN-Shapley (no subsampling) | DP-KNN-Shapley (with subsampling) | DP-TKNN-Shapley | DP-KNN-Shapley (no subsampling) | DP-KNN-Shapley (with subsampling) |
| 2dplanes | 0.1 | 0.883 (0.017) | 0.49 (0.024) | 0.733 (0.011) | 0.692 (0.014) | 0.494 (0.023) | 0.615 (0.01) |
| | 0.5 | 0.912 (0.009) | 0.488 (0.022) | 0.815 (0.006) | 0.706 (0.004) | 0.494 (0.012) | 0.66 (0.004) |
| | 1 | 0.913 (0.009) | 0.504 (0.019) | 0.821 (0.005) | 0.705 (0.007) | 0.495 (0.011) | 0.665 (0.004) |
| phoneme | 0.1 | 0.816 (0.011) | 0.5 (0.014) | 0.692 (0.011) | 0.648 (0.028) | 0.475 (0.042) | 0.566 (0.01) |
| | 0.5 | 0.826 (0.007) | 0.497 (0.011) | 0.738 (0.003) | 0.683 (0.014) | 0.536 (0.033) | 0.588 (0.004) |
| | 1 | 0.826 (0.005) | 0.486 (0.01) | 0.741 (0.002) | 0.685 (0.016) | 0.494 (0.071) | 0.59 (0.005) |
| CPU | 0.1 | 0.932 (0.007) | 0.49 (0.028) | 0.881 (0.005) | 0.805 (0.037) | 0.42 (0.074) | 0.709 (0.011) |
| | 0.5 | 0.946 (0.004) | 0.507 (0.029) | 0.928 (0.002) | 0.838 (0.007) | 0.472 (0.092) | 0.746 (0.003) |
| | 1 | 0.948 (0.002) | 0.512 (0.008) | 0.931 (0.002) | 0.839 (0.003) | 0.455 (0.079) | 0.748 (0.002) |
| Creditcard | 0.1 | 0.661 (0.008) | 0.511 (0.026) | 0.584 (0.011) | 0.558 (0.017) | 0.479 (0.115) | 0.519 (0.014) |
| | 0.5 | 0.663 (0.008) | 0.503 (0.021) | 0.624 (0.011) | 0.571 (0.058) | 0.513 (0.089) | 0.541 (0.008) |
| | 1 | 0.668 (0.005) | 0.491 (0.026) | 0.627 (0.009) | 0.585 (0.048) | 0.525 (0.107) | 0.545 (0.006) |
| Click | 0.1 | 0.566 (0.017) | 0.512 (0.013) | 0.511 (0.024) | 0.55 (0.012) | 0.439 (0.094) | 0.497 (0.017) |
| | 0.5 | 0.567 (0.013) | 0.502 (0.018) | 0.529 (0.012) | 0.555 (0.01) | 0.506 (0.071) | 0.503 (0.018) |
| | 1 | 0.567 (0.007) | 0.502 (0.021) | 0.532 (0.01) | 0.559 (0.009) | 0.488 (0.076) | 0.504 (0.017) |
| Wind | 0.1 | 0.861 (0.021) | 0.489 (0.022) | 0.836 (0.01) | 0.622 (0.167) | 0.464 (0.282) | 0.557 (0.017) |
| | 0.5 | 0.883 (0.002) | 0.511 (0.021) | 0.873 (0.004) | 0.668 (0.055) | 0.554 (0.282) | 0.601 (0.007) |
| | 1 | 0.882 (0.002) | 0.486 (0.022) | 0.875 (0.004) | 0.681 (0.032) | 0.595 (0.257) | 0.605 (0.008) |
| Fraud | 0.1 | 0.947 (0.004) | 0.497 (0.02) | - | 0.813 (0.033) | 0.468 (0.127) | - |
| | 0.5 | 0.952 (0.002) | 0.502 (0.026) | - | 0.85 (0.017) | 0.548 (0.192) | - |
| | 1 | 0.953 (0.003) | 0.502 (0.008) | - | 0.859 (0.007) | 0.497 (0.162) | - |
| Apsfail | 0.1 | 0.903 (0.078) | 0.501 (0.018) | - | 0.693 (0.2) | 0.456 (0.357) | - |
| | 0.5 | 0.952 (0.001) | 0.521 (0.023) | - | 0.873 (0.014) | 0.489 (0.139) | - |
| | 1 | 0.952 (0.0) | 0.49 (0.031) | - | 0.882 (0.003) | 0.494 (0.179) | - |
| Pol | 0.1 | 0.844 (0.006) | 0.508 (0.012) | - | 0.604 (0.082) | 0.53 (0.212) | - |
| | 0.5 | 0.858 (0.006) | 0.5 (0.029) | - | 0.673 (0.039) | 0.626 (0.153) | - |
| | 1 | 0.857 (0.001) | 0.516 (0.02) | - | 0.695 (0.02) | 0.622 (0.195) | - |
| MNIST | 0.1 | 0.801 (0.002) | 0.492 (0.007) | - | 0.841 (0.002) | 0.384 (0.129) | - |
| | 0.5 | 0.948 (0.002) | 0495 (0.005) | - | 0.873 (0.001) | 0.389 (0.125) | - |
| | 1 | 0.958 (0.002) | 0.495 (0.005) | - | 0.881 (0.002) | 0.4 (0.002) | - |
| CIFAR10 | 0.1 | 0.924 (0.002) | 0.496 (0.014) | - | 0.821 (0.002) | 0.558 (0.013) | - |
| | 0.5 | 0.936 (0.002) | 0.497 (0.012) | - | 0.847 (0.004) | 0.560 (0.009) | - |
| | 1 | 0.949 (0.003) | 0.497 (0.012) | - | 0.852 (0.001) | 0.560 (0.006) | - |
| AGNews | 0.1 | 0.725 (0.002) | 0.491 (0.002) | - | 0.611 (0.003) | 0.421 (0.065) | - |
| | 0.5 | 0.733 (0.005) | 0.492 (0.002) | - | 0.621 (0.001) | 0.426 (0.075) | - |
| | 1 | 0.738 (0.002) | 0.494 (0.002) | - | 0.627 (0.001) | 0.426 (0.075) | - |
| DBPedia | 0.1 | 0.652 (0.005) | 0.499 (0.002) | - | 0.579 (0.002) | 0.360 (0.07) | - |
| | 0.5 | 0.654 (0.002) | 0.499 (0.002) | - | 0.586 (0.002) | 0.360 (0.09) | - |
| | 1 | 0.657 (0.001) | 0.5 (0.002) | - | 0.588 (0.001) | 0.361 (0.07) | - |

Table 8: Privacy-utility tradeoff of DP-TKNN-Shapley and DP-KNN-Shapley for mislabeled/noisy data detection task on 13 datasets. We show the AUROC at different privacy budgets $\varepsilon$ (We set $\delta = 10^{-5}$ for MNIST, CIFAR10, AGNews, and PBPedia, and we set $\delta = 10^{-4}$ for the rest of dataset, following the common rule of $\delta \approx 1/N$). The higher the AUROC is, the better the method is. We show the standard deviation of AUROC across 5 independent runs in (). The results for DP-KNN-Shapley with subsampling are omitted for half of the datasets as it requires a significant amount of runtime.

performance of TKNN-Shapley is quite stable on a wide range of choices of $\tau$. For example, for 2dPlanes dataset, the performance of TKNN-Shapley only varies $< 0.032$ between $\tau \in [-0.1, -0.6]$. Moreover, $\tau \approx -0.5$ is a relatively robust choice of the threshold value. Hence, hyperparameter tuning is not difficult for TKNN-Shapley. This is not too surprising; the formula of TKNN-Shapley (in Equation (7)) shows that regardless of the choice of $\tau$, if a training data point is within the threshold but has a different label from the validation data, then its value is necessarily negative. At the same time, the points that share same label as validation data are guaranteed to have a positive value. The choice of $\tau$ only decides the magnitude but not the sign. As data valuation methods only rely on the rank of data values to perform data selection, the performance of TKNN-Shapley is relatively robust to the choice of $\tau$ in identifying mislabeled data points (who get negative values). A similar argument also applies to the task of noisy data detection: noisy data is typically far from the population; hence, as long as $\tau$ is not too large, the noisy data will receive a value score close to 0. On the other hand, clean data will receive a positive value score, and the choice of $\tau$ only affects the magnitude of the data value but not the sign.

Ideally, we would like to set a threshold such that for most of the queries, there are a certain amount of neighbors whose distance to the query is below the threshold. One possible way to do this is simply picking a $\tau$ that optimizes the final validation accuracy of the TKNN classifier (note that TKNN is very efficient in hyperparameter tuning).

| Dataset | TKNN-Shapley | KNN-Shapley | Data Shapley | LOO | Data Banzhaf |
|---|---|---|---|---|---|
| **2DPlanes** | 0.919 | 0.913 | 0.552 | 0.458 | 0.548 |
| **Phoneme** | 0.826 | 0.873 | 0.525 | 0.484 | 0.604 |
| **CPU** | 0.946 | 0.932 | 0.489 | 0.496 | 0.513 |
| **Fraud** | 0.96 | 0.967 | 0.488 | 0.495 | 0.534 |
| **Creditcard** | 0.662 | 0.646 | 0.517 | 0.487 | 0.536 |
| **Apsfail** | 0.958 | 0.948 | 0.496 | 0.485 | 0.506 |
| **Click** | 0.572 | 0.568 | 0.474 | 0.504 | 0.528 |
| **Wind** | 0.889 | 0.896 | 0.469 | 0.456 | 0.512 |
| **Pol** | 0.871 | 0.928 | 0.512 | 0.538 | 0.473 |
| **MNIST** | 0.962 | 0.974 | - | - | - |
| **CIFAR10** | 0.957 | 0.991 | - | - | - |
| **AG News** | 0.956 | 0.971 | - | - | - |
| **DBPedia** | 0.981 | 0.991 | - | - | - |

Table 9: Full Table for the AUROC of Mislabel Data Detection (non-private setting). The results for LOO, Data Shapley, Data Banzhaf are omitted for MNIST, CIFAR10, AGNews and DBPedia as they require a significant amount of runtime (prior works typically use a subset for MNIST and CIFAR10 [23, 39, 60]).

| Dataset | TKNN-Shapley | KNN-Shapley | Data Shapley | LOO | Data Banzhaf |
|---|---|---|---|---|---|
| **2DPlanes** | 0.705 | 0.718 | 0.545 | 0.504 | 0.656 |
| **Phoneme** | 0.71 | 0.706 | 0.49 | 0.531 | 0.631 |
| **CPU** | 0.751 | 0.726 | 0.527 | 0.459 | 0.531 |
| **Fraud** | 0.818 | 0.794 | 0.502 | 0.526 | 0.582 |
| **Creditcard** | 0.611 | 0.64 | 0.535 | 0.513 | 0.506 |
| **Apsfail** | 0.841 | 0.666 | 0.533 | 0.465 | 0.566 |
| **Click** | 0.558 | 0.547 | 0.461 | 0.503 | 0.572 |
| **Wind** | 0.691 | 0.794 | 0.495 | 0.488 | 0.54 |
| **Pol** | 0.68 | 0.79 | 0.526 | 0.503 | 0.588 |
| **MNIST** | 0.613 | 0.815 | - | - | - |
| **CIFAR10** | 0.85 | 0.935 | - | - | - |
| **AG News** | 0.804 | 0.746 | - | - | - |
| **DBPedia** | 0.719 | 0.88 | - | - | - |

Table 10: Full Table for the AUROC of Noisy Data Detection (non-private setting). The results for LOO, Data Shapley, Data Banzhaf are omitted for MNIST, CIFAR10, AGNews and DBPedia as they require a significant amount of runtime (prior works typically use a subset for MNIST and CIFAR10 [23, 39, 60]).

For completeness, we also show the impact of different choices of $K$ on the performance of KNN-Shapley in Table 13 and 14. As we can see, KNN-Shapley's performance is quite stable against the different choices of $K$.

| Dataset | $\tau = -0.1$ | $\tau = -0.2$ | $\tau = -0.3$ | $\tau = -0.4$ | $\tau = -0.5$ | $\tau = -0.6$ | $\tau = -0.7$ | $\tau = -0.8$ | $\tau = -0.9$ |
|---|---|---|---|---|---|---|---|---|---|
| **2dPlanes** | 0.889 | 0.902 | 0.911 | 0.918 | 0.92 | 0.887 | 0.862 | 0.725 | 0.553 |
| **CPU** | 0.952 | 0.956 | 0.954 | 0.952 | 0.95 | 0.943 | 0.917 | 0.895 | 0.834 |
| **Phoneme** | 0.79 | 0.799 | 0.805 | 0.811 | 0.825 | 0.834 | 0.844 | 0.859 | 0.861 |
| **Fraud** | 0.961 | 0.956 | 0.951 | 0.954 | 0.959 | 0.957 | 0.931 | 0.884 | 0.75 |
| **Creditcard** | 0.653 | 0.671 | 0.668 | 0.669 | 0.662 | 0.652 | 0.648 | 0.606 | 0.539 |
| **Apsfail** | 0.959 | 0.952 | 0.962 | 0.967 | 0.962 | 0.952 | 0.924 | 0.899 | 0.851 |
| **Click** | 0.576 | 0.552 | 0.557 | 0.56 | 0.568 | 0.556 | 0.547 | 0.538 | 0.522 |
| **Wind** | 0.875 | 0.879 | 0.882 | 0.885 | 0.888 | 0.889 | 0.879 | 0.888 | 0.775 |
| **Pol** | 0.847 | 0.854 | 0.864 | 0.871 | 0.875 | 0.878 | 0.89 | 0.878 | 0.86 |

Table 11: Impact of different choices of $\tau$ for TKNN-Shapley on the performance of mislabel data detection.

| Dataset | $\tau = -0.1$ | $\tau = -0.2$ | $\tau = -0.3$ | $\tau = -0.4$ | $\tau = -0.5$ | $\tau = -0.6$ | $\tau = -0.7$ | $\tau = -0.8$ | $\tau = -0.9$ |
|---|---|---|---|---|---|---|---|---|---|
| **2dPlanes** | 0.671 | 0.691 | 0.704 | 0.696 | 0.705 | 0.694 | 0.678 | 0.626 | 0.528 |
| **CPU** | 0.691 | 0.687 | 0.696 | 0.709 | 0.751 | 0.777 | 0.789 | 0.786 | 0.75 |
| **Phoneme** | 0.681 | 0.688 | 0.691 | 0.698 | 0.71 | 0.718 | 0.716 | 0.725 | 0.72 |
| **Fraud** | 0.758 | 0.781 | 0.804 | 0.809 | 0.818 | 0.837 | 0.847 | 0.814 | 0.664 |
| **Creditcard** | 0.579 | 0.576 | 0.585 | 0.597 | 0.611 | 0.624 | 0.642 | 0.626 | 0.574 |
| **Apsfail** | 0.771 | 0.822 | 0.834 | 0.855 | 0.841 | 0.842 | 0.828 | 0.792 | 0.749 |
| **Click** | 0.53 | 0.544 | 0.537 | 0.539 | 0.558 | 0.563 | 0.521 | 0.521 | 0.539 |
| **Wind** | 0.592 | 0.612 | 0.637 | 0.663 | 0.691 | 0.727 | 0.747 | 0.754 | 0.719 |
| **Pol** | 0.594 | 0.638 | 0.643 | 0.661 | 0.68 | 0.682 | 0.712 | 0.721 | 0.699 |

Table 12: Impact of different choices of $\tau$ for TKNN-Shapley on the performance of noisy data detection.

| Dataset | $K=1$ | $K=2$ | $K=3$ | $K=4$ | $K=5$ | $K=6$ | $K=7$ | $K=8$ | $K=9$ | $K=10$ | $K=11$ | $K=12$ | $K=13$ | $K=14$ | $K=15$ |
|---|---|---|---|---|---|---|---|---|---|---|---|---|---|---|---|
| **2dPlanes** | 0.914 | 0.914 | 0.915 | 0.915 | 0.916 | 0.915 | 0.915 | 0.914 | 0.915 | 0.915 | 0.914 | 0.91 | 0.905 | 0.898 | 0.896 |
| **CPU** | 0.936 | 0.936 | 0.937 | 0.937 | 0.937 | 0.937 | 0.938 | 0.937 | 0.936 | 0.934 | 0.932 | 0.929 | 0.928 | 0.927 | 0.921 |
| **Phoneme** | 0.859 | 0.861 | 0.862 | 0.862 | 0.864 | 0.865 | 0.868 | 0.868 | 0.869 | 0.871 | 0.871 | 0.871 | 0.871 | 0.874 | 0.86 |
| **Fraud** | 0.972 | 0.972 | 0.971 | 0.971 | 0.972 | 0.972 | 0.972 | 0.971 | 0.971 | 0.971 | 0.97 | 0.97 | 0.967 | 0.968 | 0.966 |
| **Creditcard** | 0.652 | 0.652 | 0.653 | 0.653 | 0.653 | 0.651 | 0.65 | 0.651 | 0.651 | 0.649 | 0.646 | 0.642 | 0.638 | 0.637 | 0.634 |
| **Apsfail** | 0.952 | 0.952 | 0.952 | 0.952 | 0.951 | 0.951 | 0.949 | 0.948 | 0.947 | 0.947 | 0.948 | 0.949 | 0.949 | 0.95 | 0.95 |
| **Click** | 0.56 | 0.562 | 0.564 | 0.563 | 0.566 | 0.566 | 0.563 | 0.562 | 0.563 | 0.567 | 0.566 | 0.557 | 0.547 | 0.553 |
| **Wind** | 0.9 | 0.9 | 0.9 | 0.9 | 0.9 | 0.899 | 0.899 | 0.899 | 0.899 | 0.898 | 0.896 | 0.897 | 0.896 | 0.891 | 0.89 |
| **Pol** | 0.931 | 0.931 | 0.931 | 0.931 | 0.93 | 0.929 | 0.929 | 0.93 | 0.929 | 0.927 | 0.925 | 0.922 | 0.919 | 0.914 | 0.917 |

Table 13: Impact of different choices of $K$ for KNN-Shapley on the performance of mislabel data detection.

| Dataset | $K=1$ | $K=2$ | $K=3$ | $K=4$ | $K=5$ | $K=6$ | $K=7$ | $K=8$ | $K=9$ | $K=10$ | $K=11$ | $K=12$ | $K=13$ | $K=14$ | $K=15$ |
|---|---|---|---|---|---|---|---|---|---|---|---|---|---|---|---|
| **2dPlanes** | 0.719 | 0.719 | 0.718 | 0.718 | 0.718 | 0.717 | 0.716 | 0.715 | 0.714 | 0.712 | 0.712 | 0.709 | 0.704 | 0.703 | 0.703 |
| **CPU** | 0.7 | 0.702 | 0.703 | 0.705 | 0.706 | 0.706 | 0.709 | 0.711 | 0.717 | 0.723 | 0.73 | 0.736 | 0.744 | 0.751 | 0.766 |
| **Phoneme** | 0.718 | 0.721 | 0.722 | 0.724 | 0.726 | 0.729 | 0.732 | 0.735 | 0.738 | 0.741 | 0.739 | 0.74 | 0.739 | 0.74 | 0.742 |
| **Fraud** | 0.786 | 0.788 | 0.789 | 0.792 | 0.794 | 0.794 | 0.795 | 0.797 | 0.797 | 0.799 | 0.803 | 0.805 | 0.811 | 0.815 | 0.822 |
| **Creditcard** | 0.637 | 0.638 | 0.636 | 0.638 | 0.64 | 0.642 | 0.643 | 0.644 | 0.646 | 0.646 | 0.645 | 0.642 | 0.636 | 0.636 | 0.64 |
| **Apsfail** | 0.667 | 0.667 | 0.667 | 0.666 | 0.666 | 0.666 | 0.666 | 0.668 | 0.671 | 0.674 | 0.678 | 0.683 | 0.69 | 0.711 | 0.763 |
| **Click** | 0.544 | 0.547 | 0.552 | 0.549 | 0.547 | 0.549 | 0.549 | 0.547 | 0.548 | 0.544 | 0.544 | 0.543 | 0.543 | 0.544 | 0.545 |
| **Wind** | 0.796 | 0.796 | 0.795 | 0.794 | 0.794 | 0.793 | 0.792 | 0.791 | 0.791 | 0.79 | 0.788 | 0.788 | 0.788 | 0.789 | 0.791 |
| **Pol** | 0.789 | 0.788 | 0.788 | 0.788 | 0.79 | 0.79 | 0.788 | 0.787 | 0.786 | 0.786 | 0.787 | 0.79 | 0.786 | 0.786 | 0.785 |

Table 14: Impact of different choices of $K$ for KNN-Shapley on the performance of noisy data detection.

