# OpenReview forum: "A Privacy-Friendly Approach to Data Valuation"
_NeurIPS.cc/2023/Conference — NeurIPS 2023 spotlight_

### Official Review · Reviewer_yczn · 2023-07-10

**Soundness:** 4 excellent
**Presentation:** 3 good
**Contribution:** 3 good
**Rating:** 7
**Confidence:** 4

**Summary:**

This paper proposes an approach for data valuation that tries to be privacy-preserving by design. More precisely, the proposed method aims at computing the data valuation of a point through its Shapley value for the k-nearest neighbours classifier (knn) in a differentially-private manner.

**Strengths:**

The paper is well-written and the authors have clearly summarized the main contributions of the paper. The background notions on Shapley values as well as knn Shapley values are also clearly explained.

An analysis of the privacy risks associated to the release of knn Shapley values has been conducted, which demonstrates the associated potential threats such as the possibility to perform a membership inference attack. Several arguments are also provided to explain why making a differentially-private version of the knn Shapley values is not a trivial task.

A variant of knn Shapley, called threshold-based knn (tknn), is introduced that focuses on the neighbourhood around a point according to a specified threshold to compute the Shapley value of this point. The benefits of this approach is terms of robustness and computation efficiency are clearly highlighted and supported by the results of the experiments conducted.

Finally another contribution is the design of a differentially-private variant of tknn. This approach is elegant in the sense that it exploits the structure of tknn to bound the sensitivity and thus the noise that needs to be added to achieve (\epsilon,\delta)-privacy.

A thorough set of experiments has been performed on a variety of different datasets against various baselines corresponding to variants of knn Shapley. The results demonstrate clearly that both the non-private and differentially-private versions of tknn-Shapley outperform the knn-shapley method or offer similar utility but at a lower cost.

**Weaknesses:**

The paper currently lacks a discussion on the limitations of the knn Shapley values compared to other methods for computing Shapley values in a more generic context. For instance, is there situations in which knn Shapley values are known not to be informative such as high-dimensional data?

The notions of collusion resistance and joint differential privacy should be defined in more depth. In particular, it would important to discuss how they differ from traditional differential privacy and whether or not this is a property that a lot of differential-privacy methods based on perturbations of counts could have.

With respect to analyzing the privacy risks associated to Shapley values, the paper is currently missing an important work from CCS last year that had conducted an in-depth study on the privacy risks related to the disclosure of Shapley values :
Luo, X., Jiang, Y., & Xiao, X. (2022, November). Feature inference attack on shapley values. In Proceedings of the 2022 ACM SIGSAC Conference on Computer and Communications Security (pp. 2233-2247).
It is important that the authors position their works with respect to this work. In particular, it would be great if the authors could comment on whether or not the attacks described in this paper could applied even against Shapley values computed with DP-tknn.

**Questions:**

See weakness section for the main points raised.

**Limitations:**

OK.

---

> ### Author Rebuttal · Authors · 2023-08-09
>
> We thank the reviewer for the very positive feedback!
>
> **Q1. [Limitation of KNN-Shapley?]** *“The paper currently lacks a discussion on the limitations of the knn Shapley values compared to other methods for computing Shapley values in a more generic context. For instance, is there situations in which knn Shapley values are known not to be informative such as high-dimensional data?”*
>
> **A:** We appreciate the insightful question. Indeed, in comparison to the line of works of Data Shapley/Banzhaf, KNN-Shapley may have the following limitations:
>
> **(1)** KNN-Shapley focuses on KNN classifiers. As a result, the applicability of KNN-Shapley scores as a proxy of data points' value with respect to other ML models may not be straightforward. However, it is noteworthy that KNN is asymptotically Bayes optimal, implying that KNN-Shapley scores can be justified as a proxy for the data's value relative to the best possible model, i.e., the Bayes classifier, under certain asymptotic conditions.
>
> **(2)** For high-dimensional data, such as images, KNN-Shapley requires a public model to first map the original data into data embeddings, and evaluates the value of these embeddings rather than the original data. While this is indeed a constraint in certain scenario, it is important to recognize that utilizing a publicly available foundation model to convert original data into embeddings, followed by the fine-tuning of the model's last layer, has become a common practice. Therefore, in many situations, it might be more desirable to evaluate the value of data embeddings instead of the original data.
>
>
> **Q2. [More discussion for collusion resistance and Joint differential privacy?]** *“The notions of collusion resistance and joint differential privacy should be defined in more depth. In particular, it would important to discuss how they differ from traditional differential privacy and whether or not this is a property that a lot of differential-privacy methods based on perturbations of counts could have.”*
>
> **A:**
> **(1) How Joint DP differ from traditional DP?** Traditional differential privacy (DP) is designed to safeguard the identities of **all** private individuals within a dataset. In contrast, joint differential privacy considers a mechanism that releases one piece of its output to each individuals in the dataset (aligns well with our data valuation setting where each data value score is released to the corresponding data owner). Joint DP requires the privacy guarantees of the i-th piece of output to other private individuals in the dataset except for the i-th individual’s data. In our data valuation scenario, one can think of it as “the data value score for $i$th data wonder does not need to protect the privacy of $i$th data owner itself”. In particular, joint DP preserves the privacy guarantee when multiple data owners collude with each other and try to infer the membership of the rest of the data owners.
>
> **(2) whether or not this is a property that a lot of differential-privacy methods based on perturbations of counts could have?** No. Our DP-TKNN-Shapley mechanism $M(D) := (\hat \phi_{z_1}(D_{-z_1}), \ldots, \hat \phi_{z_N}(D_{-z_N}))$ satisfies JDP guarantee is **because of the special, post-processing-based computation algorithm we use**. Recall from Appendix E.2.1, to compute DP-TKNN-Shapley, we first calculate $\hat C(D)$ and then, for each $z_i \in D$, we compute $\hat C(D_{-z_i})$ through the deterministic post-processing step outlined in Appendix E.2.1, where the post-processing for $\hat C(D_{-z_i})$ only depends on $z_i$ itself. Hence, due to DP’s post-processing property, in Theorem 16 we show that the JDP guarantee for the overall mechanism $M(D)$ has the same privacy parameter as the one for “sub-mechanism” $M_i(D_{-z_i}) := \hat \phi_{z_i}(D_{-z_i})$. An intuitive way of thinking why $M(D)$ is collusion resistant is that because random noise is only applied once on $\hat C(D)$, each post-processed $\hat C(D_{-z_i})$ are privatized with the same random noise. Hence, even if multiple attackers collude and share the received noisy data value scores with each other, they are effectively only sharing the same random sample of $\hat C(D)$, and cannot gain more information about $z_i$ other than that. As a comparison, if we naively privatize each “sub-mechanism” $M_i(D_{-z_i}) = \hat \phi_{z_i}(D_{-z_i})$ with independent random noise, then the attackers can gather more information by sharing its own random sample that contains information about $z_i$, and thus such an approach is not only less efficient, but also does not enjoy such powerful collusion resistance property.
>
>
> **Q3. [Related literature on feature inference attack on the Shapley value]**
>
> **A:** We thank the reviewer for referring relevant literature! We have added the discussion for this CCS work in Appendix A. The privacy risks studied in their paper and our work are different. Specifically, the attack proposed in this CCS work is against the Shapley value for feature attribution, while we study the privacy risks when using the Shapley value for data valuation. Regarding the question of *“whether or not the attacks described in this paper could applied even against Shapley values computed with DP-TKNN.”*, we would like to stress that it is not clear how to extend TKNN-Shapley for feature attribution, hence currently it’s not possible to evaluate the attack on DP-TKNN-Shapley.

---

> > ### Comment · Reviewer_yczn · 2023-08-19
> > **Thanks for your response**
> >
> > Thanks for your answer, you have clarified the main points that I have raised in my review. In addition, I have seen the answer to other reviewers and I definitively believe that this is a solid work, which was already reflected in my original score.

---

> > > ### Author Response · Authors · 2023-08-19
> > > **Thanks!**
> > >
> > > We sincerely thank the reviewer for the very encouraging comments!

---

### Official Review · Reviewer_i1sa · 2023-07-11

**Soundness:** 3 good
**Presentation:** 3 good
**Contribution:** 3 good
**Rating:** 6
**Confidence:** 4

**Summary:**

The authors propose a relaxation of the KNN based Shapley valuation framework based on thresholding. By doing so, they are able to reduce the sensitivity associated with shapley valuation, and consequently apply DP to the mechanism.

**Strengths:**

1. First formalization of TKNN based shapley valuation.

**Weaknesses:**

1. Unclear what the primary contribution is. I conjecture it is the TKNN relaxation, and its reduced run time + closed form solution.

**Questions:**

1. Could the authors kindly clarify how they validate the “goodness” of their TKNN method? Since the scores returned are “real numbers”, what does the AUROC measure actually capture? Similarly, what is the dependence of the threshold to determine the closeness of TKNN and KNN based shapley valuation?
2. The primary contribution of this work is demonstrating a relaxation from the KNN to the TKNN setting. The application of gaussian noise to ensure DP is a standard technique. However, the authors motivate the need for DP with an experiment that motivates how privacy is leaked through membership inference. Could the authors clarify the setting once more? The authors claim that there are multiple colluding entities that aim to identify if a particular point is present or absent in the data valuation pipeline. To this end, they need to know the data point under consideration, which seems impractical given the setup described for data valuation (e.g., where data is contributed by individual entities for medical purposes).
3. In algorithm 1, what is 	\phi_z (Dattack \cup z) formally defined as?

**Limitations:**

None discussed

---

> ### Author Rebuttal · Authors · 2023-08-09
>
> We thank the reviewer for the positive feedback!
>
> **Q1. [Primary contribution]**
>
> **A:** Our contributions in this work are three-fold:
>
> 1. **Uncovering the privacy risks of data valuation technique:** To the best of our knowledge, this work is the first formal study on the privacy risks associated with releasing data value scores. Another contribution is that we show KNN-Shapley cannot be easily modified to incorporate differential privacy, which calls for a more privacy-friendly approach for data valuation.
>
> 2. **Introduction of TKNN-Shapley:** We found that Threshold KNN (TKNN), a simplified variant of KNN also possesses a closed-form Data Shapley formula and offers superior computational efficiency, reducing computation time to exact $O(N)$ from $O(N \log N)$ compared to KNN-Shapley.
>
> 3. **Incorporating TKNN-Shapley with differential privacy (DP-TKNN-Shapley):** Moreover, we demonstrate how TKNN-Shapley can be easily adapted to integrate differential privacy, offering a solution to the earlier limitations we identified with KNN-Shapley. Further to this, we show that DP-TKNN-Shapley brings several benefits such as collusion resistance.
>
> **Q2. [How did we evaluate the “goodness” of data valuation method?]**
>
> **A:** In our work, we follow the same evaluation method for data valuation techniques as the one used in the prior works in this field [1-2]: the effectiveness of data valuation techniques is quantified based on their performance in filtering out bad (mislabeled/noisy) data points. The underlying rationale is that a reasonable data valuation should assign lower scores to poor-quality data points, and higher scores to high-quality data points. Hence, we can measure the “goodness” of data value notions based on whether those poor-quality data points are actually receive lower value scores. In the task of filtering out bad (mislabeled/noisy) data points, we can use data valuation techniques to assign a value score for each data point, and exclude data points with the lowest scores. In other words, we are using data valuation scores as the evidence for classifying each data point as either “good” or “bad”. AUROC is a classic metric for evaluating the performance of such binary classification, and thus can be used as the measure for assessing the ability of a data value notion in *“distinguish between good and bad data”*.
>
> For **“what is the dependence of the threshold to determine the closeness of TKNN and KNN based shapley valuation?”**, it is important to emphasize that **there isn't a single "ground truth" for data value scores**. The main criteria for evaluating the efficacy of a data valuation method are its ability to identify high-quality data points with high scores and low-quality data points with lower scores. The proximity of TKNN-Shapley and KNN-Shapley scores is not particularly meaningful in itself. Instead, these data valuation methods are compared based on their capacity to detect bad data.
>
> [1] Kwon, Yongchan, and James Zou. "Beta shapley: a unified and noise-reduced data valuation framework for machine learning." AISTATS 2022.
> [2] Wang, Jiachen T., and Ruoxi Jia. "Data banzhaf: A robust data valuation framework for machine learning." AISTATS 2023.
>
>
> **Q3. [Setting of Membership Inference Attack (Algorithm 1)?]** *“the authors motivate ... for medical purposes).”*
>
> **A:** **Setting of Algorithm 1**: In Appendix C.2.2, we consider the scenario where the server owns a dataset $D$ (collected from other users), and the attacker would like to know whether a target example $z$ is included in $D$. **Threat model:** in Appendix C.2.2, we consider that an attacker with the ability of (1) computing the value of any data point among any datasets (analogue to the setting of MIA against ML models where the attacker can train models on any datasets he/she constructs), and (2) crafting the data point it owns, send it to the server and obtain the data value score of its own data point.
>
> We have further enriched the details in Appendix C.2.2 and we hope the attack setting is clear now :)
>
> **For “they need to know the data point under consideration, which seems impractical given the setup described for data valuation”**: we note that membership inference attack, i.e., confirming one's membership in a database, could pose significant privacy risks. For example, if a medical database is known to be a dataset for patients with specific conditions, confirming one’s membership in this database could disclose members' health status. Indeed, to conduct membership inference attack, the attacker needs to first know about the features of the target data point. This is also common in many scenarios, e.g., the height and weight of an individual sometimes may not be considered as sensitive, and what is considered as sensitive is the “membership” of an individual in a special database. In this work, we mainly use membership inference attacks to illustrate the privacy risks of data valuation techniques, as membership inference attacks have long been recognized as a fundamental privacy challenge across various domains, and such attacks have been extend well into the field of machine learning.
>
> [1] Carlini, Nicholas, et al. "Membership inference attacks from first principles." S&P 2022.
>
> **Q4.** *“In algorithm 1, what is \phi_z (Dattack \cup z) formally defined as?”*
>
> **A:** $\phi_z(D_{attack} \cup {z})$ means the data value score of $z$ when the rest of the dataset is $D_{attack} \cup {z}$, i.e., the full dataset has two identical copies of $z$. In other words, we are collecting and comparing the KNN-Shapley of $z$ between the cases of **(1)** the rest of the dataset contains its identical copy and **(2)** the rest of the dataset does **not** contain its identical copy. The intuition of the attack is as follows: if the target point $z \in D$, then $\phi_z(D)$ will be closer to the values in case 1, and if the target point $z \notin D$, then $\phi_z(D)$ will be closer to the values in case 2.

---

> > ### Comment · Reviewer_i1sa · 2023-08-13
> > **Thank you!**
> >
> > Thanks for your response. I will reflect upon the score.

---

> > > ### Author Response · Authors · 2023-08-13
> > > **Thanks for your response!**
> > >
> > > We sincerely thank the reviewer for the positive comments! Your questions are highly valuable and we have made several edits to our paper accordingly.

---

### Official Review · Reviewer_z2hx · 2023-07-12

**Soundness:** 3 good
**Presentation:** 3 good
**Contribution:** 3 good
**Rating:** 6
**Confidence:** 3

**Summary:**

In this paper, the authors suggest a modified version of KNN-Shapley that aims to enhance the privacy of the data valuation procedure. They introduce TKNN-Shapley, which focuses only on neighbors within a $\tau$ vicinity of the utilized data point instead of a fixed number of neighbors (k), and DP-TKNN-Shapley to ensure differential privacy by incorporating the Gaussian mechanism. The paper also includes numerical experiments comparing the efficiency of TKNN-Shapley to KNN-Shapley and demonstrates the competitive performance of TKNN-Shapley in both private and non-private settings for detecting mislabeled and noisy data. The proposed approach consistently performs well across all presented results.

**Strengths:**

The paper is generally clear with good coverage of related work (although I humbly listed some improvement points later in my review).

I think that the main strength of the paper is the performance of the proposed method in particular in terms of data-valuation efficiency and detecting mislabeled/noisy data in the private setting. It also seems to be competitive in the non-private setting, as such thresholding operator itself for finding neighbors appears to be a safe choice. The theoretical analysis and claims also seem sound as far as my knowledge extends.



**Weaknesses:**

I have concerns regarding  a. the originality of the proposed method and b. how it concretely addresses the privacy risks.

a. The method is a combination of well-known techniques such as KNN-Shapley, threshold-KNN, and differential privacy, and most of the analyses are adapted from previous derivations (eg KNN-Shapley) for this very specific algorithm. As such, the theoretical contributions in the paper are mainly to be attributed to the existing result. I will gladly stand corrected by the authors if my understanding is limited. All said, as I indicated in my previous comment, its performance seems significant in the demonstrated tasks and it is valuable work.

b. Despite having read the privacy risk section, I do not immediately see how the privacy risks are addressed, which is the main claim of the work. I am fairly less confident in this argument though, as I am not familiar with the privacy field. I wish to understand this aspect better.

Other than that, in terms of clarity, I believe there is room for improvement:
- I think the background is too long and the actual contribution starts on page 6. This also seems to be because the proposed method combines existing strategies, but I think the background can still rather be cut short and more room can be created for experimental results or even the related work in the appendix section.

Further notes are presented in the form of questions below.



**Questions:**

It would be helpful if the authors could clarify the following confusions:
- line 265: where does the computational complexity $\mathcal{O}(N)$ of the TKNN-Shapley comes from?
- Concretely what privacy risks can the proposed methods address? I was not able to understand the concrete risks (apart from line 156 which is outside the scope of the paper).
- How does the noisy/mislabeled data detection in the numerical experiments tie to addressing privacy risks? I am not knowledgeable in the privacy field so I would appreciate the clarification even if it is an obvious explanation.
- line 251: I am a bit confused about the raw summation. I thought it is rather averaged over the validation set
- I think KNN-Shapley uses K in the denominator of Equation (2). Otherwise, I don't think their equation (69)* holds. It doesn't seem to affect anything with the proposed method though. Happy to be corrected.


*https://arxiv.org/pdf/1908.08619.pdf







**Limitations:**

The authors addressed several limitations in the submission. Additionally, I might add the sensitivity of DP-TKNN-Shapley's performance towards the choice of threshold $\tau$ (though, probably not much further than what $k$ brings in the KNN-Shapley).

---

> ### Author Rebuttal · Authors · 2023-08-08
>
> We thank the reviewer for the positive feedback about our experiment!
>
> **Q1. [Theoretical Contributions in this work]** *“The method ... is limited.”*
>
> **A:** We thank the reviewer for the comment. There are in fact at least two highly non-trivial theoretical contributions (in our opinion) in this work.
>
> **(1) The derivation of TKNN-Shapley is a unique contribution distinct from existing derivations for KNN-Shapley.** The proof of Theorem 5 is different from the one for KNN-Shapley, and the primary technical challenge is observing that $C_{\tau}$ are sufficient for calculating $\phi^{tknn}_{z_i}$. This is the key reason why TKNN-Shapley can be so effectively paired with differential privacy. Furthermore, only a few games have yet been discovered to have a polynomial-time algorithm for computing the exact Shapley value. The discovery of TKNN-Shapley is a contribution we believe will be of independent interest to the game theory community.
>
> **(2) Privatization of TKNN-Shapley with collusion resistance.** While Gaussian mechanism is a standard privatization method, this is already the reward of recognizing TKNN-Shapley’s special property mentioned above. **A more non-trivial theoretical contribution is that we show DP-TKNN-Shapley also satisfies *joint differential privacy* (elaborated in Appendix E.2.2 and Theorem 16).** This ensures the collusion resistance property of DP-TKNN-Shapley, which is important for the practical deployment.**
>
> **Q2.1. [*What* are the privacy risks are being addressed?]**
>
> **A:** **Membership inference (MI) attack:** In Section 3, we mainly discussed the privacy risks of revealing data value scores with respect to MI attack, i.e., confirming one's membership in a database. MI attack could pose significant privacy risks. For example, if a medical database is known to be a dataset for patients with specific conditions, confirming one’s membership in this database could disclose members' health status. We choose MI attack as our main example of the privacy risk as it has long been recognized as a fundamental privacy challenge across various domains, and such attacks have been extended well into the field of machine learning.
>
> **DP is not limited to mitigating MI attack:** It's crucial to highlight that differential privacy (DP) goes beyond just countering membership inference. At a high level, any changes to individual data that do not change the privatized output, and hence the privatized output cannot be used to infer much about any individual.
>
> There's a rich history of studying privacy risks in aggregated dataset statistics, with KNN-Shapley being an example. For this topic, please refer to **Q4 in the global response** :)
>
> **Q2.2 [*How* the privacy risks are being addressed?]**
>
> **A:** Our primary tool for mitigating privacy risks is Differential Privacy (DP). DP has become the de-facto standard in the realm of privacy protection as differentially private algorithms offer robust, quantifiable, and provable privacy guarantees. Importantly, DP provides provable privacy guarantee against any potential privacy attacks, including those not yet developed. Thus, **if we have proved our DP-TKNN-Shapley algorithm is differentially private, we have the provable privacy guarantee** against both current and emergent privacy attacks. For a more direct illustration of how DP can protect against the privacy threat, we additionally conducted the experiment of evaluating the MI attack proposed in Appendix C.2.2 on DP-TKNN-Shapley (**see Q1 & Table 5 and 6 in Global Response**). Compared with the attack result on non-private TKNN-Shapley, the overall attack performance drops to around 0.5 (the performance of random guess). The result shows that DP-TKNN-Shapley is indeed very effective against membership inference attack.
>
> **Q3. [Background too long & the actual contribution starts on page 6?]**
>
> **A:** Thanks for the suggestion! We will cut some background and move more related works & experiment results to the main paper.
>
> We would like to clarify that our contribution starts in Section 3 (page 4). To the best of our knowledge, the privacy risks of data valuation techniques have not been extensively studied before, and our work is the first to show that data valuation techniques indeed serve as another channel for privacy leakage, and we explicitly constructed a membership inference attack which achieves non-trivial attack performance.
>
> **Q4. [Runtime of TKNN-Shapley]**
>
> **A:** The algorithm for computing TKNN-Shapley in $O(N)$ runtime is detailed in Appendix D.2.2. We have highlighted it in Theorem 5’s statement to make sure the reader can find it easily. At a high-level, the efficient TKNN-Shapley computation leverages the property that the required quantities for its determination are fundamentally counting queries performed on the dataset $D_{-z_i}$, which can be easily computed from the same counting queries performed on the full dataset $D$.
>
> **Q5.** *“How does the noisy/mislabeled data detection in the numerical experiments tie to addressing privacy risks?”*
>
> **A:** The experiments related to noisy/mislabeled data detection were not designed with the intent to evaluate the privacy risk of DP-TKNN-Shapley. We have already established, through rigorous proof, that DP-TKNN-Shapley adheres to differential privacy guarantees (and we also have additional empirical experiments for the privacy guarantee as we mentioned in **Q2.2**). The primary objective behind these experiments is to investigate the *utility* of DP-TKNN-Shapley scores. It is important that the introduction of privacy-preserving technique does not come at a big cost of utility. After all, it is trivial to attain perfect privacy by simply assigning random value scores to each data point. Our experiments demonstrate that DP-TKNN-Shapley remain useful and informative even when privacy guarantees are enforced.
>
> **We will provide additional clarifications for Q4 and more questions in "official comment".**

---

> > ### Author Response · Authors · 2023-08-10
> > **Additional clarifications for Q4 and more**
> >
> > **Q4 (enriched discussion) [Runtime of TKNN-Shapley]**
> >
> > **A:** The algorithm for computing TKNN-Shapley in $O(N)$ runtime is detailed in Appendix D.2.2. We have highlighted in Theorem 5’s statement to make sure the reader can find it easily. At a high-level, the efficient TKNN-Shapley computation leverages the property that the required quantities for its determination, namely the three quantities in **C**$(D_{-z_i})$, are fundamentally counting queries performed on the dataset $D_{-z_i}$, which can be easily computed from the same counting queries performed on the full dataset $D$.
> >
> > **Base Computation on the Full Dataset:** Instead of computing **C**$(D_{-z_i})$ for each individual data point omission (which would be computationally intensive), we first determine **C**$(D)$ on the entire dataset $D$. This step incurs computational cost of $O(N)$.
> >
> > **Efficient Computation for Individual Data Points:** For each specific data point $z_i$, we observe that all three quantities in **C**$(D_{-z_i})$ can be derived from **C**$(D)$ in $O(1)$ runtime.
> >
> > **Overall Efficiency:** Each of the computations for individual data points, owing to the preparation in step 1, can be achieved in constant time ($O(1)$). Therefore, for the entirety of the dataset, the computation scales linearly with the number of data points, resulting in an overall computational cost of $O(N)$.
> >
> > **Q6** *“line 251: I am a bit confused about the raw summation. I thought it is rather averaged over the validation set.”*
> >
> > **A:** Thanks for the question! Average or Summation does not matter in this case. It only scales the magnitude of the value scores. In the downstream task of mislabeled/noisy data detection, it is the rank of data value scores that matters, which does not change with the scaling of data value scores.
> >
> > **Q7** *“I think KNN-Shapley uses K in the denominator of Equation (2). Otherwise, I don't think their equation (69)* holds. It doesn't seem to affect anything with the proposed method though. Happy to be corrected.”*
> >
> > **A:** A recent technical note for KNN-Shapley [1] has pointed out that by using ‘K’ as the denominator of Equation (2), the utility function of KNN becomes less interpretable, and the authors of the technical note proposed a corrected version of KNN-Shapley that uses $\min(K, |S|)$ as the denominator. The two versions of KNN-Shapley are very close in terms of their performance, and in this work we use the more advanced version in the technical note for the same reason.
> >
> > [1] Wang, Jiachen T., and Ruoxi Jia. "A Note on" Efficient Task-Specific Data Valuation for Nearest Neighbor Algorithms"." arXiv preprint arXiv:2304.04258 (2023).

---

> > > ### Comment · Reviewer_z2hx · 2023-08-13
> > > **Response to the authors**
> > >
> > > Thank you for the rebuttal.
> > >
> > > You sufficiently addressed my concerns about the work, and I will reflect this on my score accordingly.

---

> > > > ### Author Response · Authors · 2023-08-13
> > > > **Thanks for raising score!**
> > > >
> > > > We sincerely thank the reviewer for the positive comments and for raising the score! We will incorporate all of your suggestions for revision.

---

### Official Review · Reviewer_SBGb · 2023-07-22

**Soundness:** 3 good
**Presentation:** 3 good
**Contribution:** 2 fair
**Rating:** 5
**Confidence:** 2

**Summary:**

The paper studies the privacy concerns surrounding KNN-Shapley, a prevalent method for data valuation. Authors introduce TKNN-Shapley, which employs the concept of differential privacy to protect privacy. Empirical tests demonstrate that TKNN-Shapley performs on par with KNN-Shapley in assessing data quality, while also providing a favorable balance between privacy.

**Strengths:**

1. The presentation is generally good.
2. The methodology is solid. The proposed TKNN-Shapley method provides a guarantee of differential privacy, which enhances the privacy protection of the data valuation process.


**Weaknesses:**

1. The research problem is quite narrow. The paper focuses on a specific metric (KNN-Shapley) for a narrow ask (data valuation), which may not arise wide interest in the ML community.
2. The significance of the privacy leakage of KNN-Shapley is not convincing.

**Questions:**

Please see limitations

**Limitations:**

1. The privacy risk of KNN-Shapley is not convincingly demonstrated. The evidence presented in Figure 2 and the corresponding figures in Appendix C only suggests that modifying certain data points can lead to changes in the Shapley score, but it's unclear if these changes are unique and can lead to a membership inference attack (MIA). Additionally, KNN is not easily susceptible to overfitting, which makes MIA less likely. Furthermore, in Figure 1, the assumption of a "trusted" center for data upload is not representative of real-world scenarios, where untrusted centers are more common and raise additional privacy concerns.

2. Data valuation is not the sole and universally accepted method for evaluating the importance of training data. Alternative dataset scoring schemes, such as core-sets or general dataset distillation approaches may be more prevalent in the field. As a recommendation, further research on these alternative methods could be beneficial.
[1] Feldman, Dan. "Introduction to core-sets: an updated survey." arXiv preprint arXiv:2011.09384 (2020).
[2] Wang, Tongzhou, et al. "Dataset distillation." arXiv preprint arXiv:1811.10959 (2018).

---

> ### Author Rebuttal · Authors · 2023-08-09
>
> We thank the reviewer for the positive feedback about our presentation and methodology!
>
> **Q1. [Privacy risk of KNN-Shapley]** *“The privacy risk of KNN-Shapley ... can lead to a membership inference attack (MIA). Additionally, KNN is not easily ... less likely. Furthermore, in Figure 1, the assumption of a "trusted" center ... concerns.”*
>
> **A:** For ***“... it's unclear if these changes are unique and can lead to a membership inference attack (MIA)”***, we presented an instantiation of a membership inference attack on KNN-Shapley score (**summarized in the paragraph above Section 3.1 and detailed in Appendix C.2.2**). The attack is designed analogue to the *likelihood ratio test* attack on ML models [1]. The attack results in Appendix C.2.2 demonstrates that privacy leakage in data value scores can indeed lead to non-trivial membership inference attacks. We have highlighted the actual instantiation of MI attack as well as moving part of the attack results to Section 3 (in Figure 2) so that the reader will not miss this contribution.
>
> To further improve the readability of Section 3, we also added a remark on the long history of the study of privacy risks underlying the aggregated statistics of a dataset (KNN-Shapey is one such example). The remark's content (*"Brief background for the privacy risks in releasing aggregated statistics"*) is outlined in **global response's Q4** for the reviewer’s reference :)
>
> [1] Carlini, Nicholas, et al. "Membership inference attacks from first principles." S&P 2022.
>
> For ***“KNN is not easily susceptible to overfitting, which makes MIA less likely”***, we’d like to clarify that we are performing MI attack against **KNN-Shapley instead of KNN classifier**. We have shown that MI attack on KNN-Shapley is possible and effective with a specific instantiation of MI attack.
>
> For ***“the assumption of a "trusted" center for data upload is not representative of real-world scenarios, where untrusted centers are more common and raise additional privacy concerns”*** We appreciate the reviewer's valid point on the assumption of a "trusted" center for data upload. We would like to clarify the following three points: **(1) Existence of real-world examples:** Many AI data marketplaces (e.g., [1, 2]) we have seen are central servers that are implicitly trusted by the users, and the users can directly upload data to these servers for selling their data. The trustworthiness of these central servers is enforced through existing legal and regulatory frameworks (e.g., [3]). **(2) Common practice in many literatures for simplifying theoretical analysis:** Many studies in the domain of data marketplace [4, 5] assume the presence of a trusted central party to facilitate specific operations. This assumption isn't unique to our work, and this allows us to understand the privacy risks incurred by releasing data valuation results without the extra complexities of an untrusted central party. **(3) Our work establishes the foundation for future research on end-to-end privacy protection** Our work primarily focuses on the privacy risks associated with revealing data value scores, which need to be protected by incorporating differential privacy (DP). As we mentioned in Remark 2 as well as Conclusion section, the privacy risks of revealing individuals' raw data to a central server are indeed crucial but they represent a different category of privacy risks that need to be addressed via secure multi-party computation (MPC). In practice, MPC and DP should be used in conjunction to provide end-to-end privacy protection. An interesting follow-up of our work is to further incorporate DP-TKNN-Shapley with an MPC framework. We emphasize that this is a substantial endeavor on its own and worth a separate, dedicated research paper. Our work establishes the foundation for this interesting future research direction.
>
> In other words, incorporating TKNN-Shapley with MPC is beyond the scope of our current work but stands as an important direction for future research.
>
> [1] Datarade https://datarade.ai/
>
> [2] Innodata https://innodata.com/data-privacy-framework/
>
> [3] https://innodata.com/data-privacy-framework/
>
> [4] Fernandez, Raul Castro, Pranav Subramaniam, and Michael J. Franklin. "Data Market Platforms: Trading Data Assets to Solve Data Problems." VLDB
>
> [5] Agarwal, Anish, Munther Dahleh, and Tuhin Sarkar. "A marketplace for data: An algorithmic solution." EC 2019.
>
>
> **Q2.** *“Data valuation is not the sole and universally accepted method for evaluating the importance of training data. Alternative dataset scoring schemes, such as coresets or general dataset distillation approaches may be more prevalent in the field.”*
>
> **A:** We acknowledge the relevance of coresets and dataset distillation approaches in the broader landscape of assessing training data importance. However, coresets and dataset distillation primarily focus on selecting/synthesizing a condensed but representative subset of a larger dataset. Although these techniques might assign some form of importance scores to individual data points, the interpretations and implications of these scores significantly differ from those of data valuation. Specifically, many data valuation techniques such as KNN-Shapley score possess a game-theoretic interpretation and carry essential fairness properties. These properties facilitate their application in data marketplace scenarios for equitable data pricing. In contrast, the importance scores derived from coresets and dataset distillation approaches may not provide an equitable valuation of data and thus might not be readily applicable for data pricing in a data marketplace. This divergence underscores the necessity and significance of the line of data valuation research.
>
> On the other hand, we completely agree with the reviewer that coresets and dataset distillation are also important techniques, and studying them from the privacy perspective is definitely interesting future work.

---

> > ### Comment · Reviewer_SBGb · 2023-08-16
> >
> > Dear authors,
> > Thanks for your response. I will update my score accordingly.

---

> > > ### Author Response · Authors · 2023-08-17
> > > **Thanks for raising the score!**
> > >
> > > We sincerely thank the reviewer for raising the score! Your review comments are very useful for improving our work!

---

### Official Review · Reviewer_QHAC · 2023-07-26

**Soundness:** 3 good
**Presentation:** 4 excellent
**Contribution:** 3 good
**Rating:** 6
**Confidence:** 3

**Summary:**

The paper addresses the problem of evaluating and assigning a value to the quality of training data for KNN in a privacy preserving way. Therefore, it first introduces an adaptation of the KNN-Shapley algorithm, namely TKNN-Shapley which due to its non-recursive way of implementation facilitates integrating differential privacy. The paper then finally proposes how to do that and provides empirical evaluation.

**Strengths:**

I did really enjoy reading the paper due to its highly accessible flow, good motivation (in terms of demonstration of the privacy risks), the gentle introduction into the KNN-Shapley, the intuitive explanation why an integration of privacy is difficult, and then the solution in terms of an adapted algorithm and the integration of differential privacy.

The experimental evaluation seems sound and thorough, so does the supplementary material.

The figures (especially Figure 1) serve well the purpose of illustrating the approach described in the work.

**Weaknesses:**

My main concern is the selection of the threshold in the TKNN. It seems crucial for the success of the algorithm, yet I could not find a good intuition and evaluation of the choice of this parameter. It would be helpful if the rebuttal could provide information about it.
Especially, I am curious about cases where there remain only few neighbors in the area below the threshold.

Additionally, I would enjoy reading more in the main paper around lines 310-312 about how subsampling can be efficiently integrated into the algorithm. Maybe, some intuition from Appendix E.2 could be migrated there?

Even after consulting Appendix E.3, it remains unclear to me what the final privacy costs of releasing all Shapley-scores for the dataset would be and how they depend on the size of the test dataset.

The choice of the privacy parameter delta (10e-4) for, e.g. the 2DPlanes dataset seems relatively high. The rule of thumb would be to choose it 1/N with N being the number of private data points. From https://www.openml.org/search?type=data&sort=runs&id=215&status=active, I understand that the dataset has north of 40k data points, so choosing delta=10e-5 might be more appropriate.



**Questions:**

Q1: Can you specify the selection of the threshold in TKNN?

Q2: Could you provide an explanation on the runtime of TKNN (O(N)) instead of the O(N log N)?

Q3: Could you extend the privacy analysis of the release of all valuation scores of the full dataset - and show how it grows with the number of data points?

Q4: Could you specify what the "naively privatized KNN-Shapley" baseline from line 324 refers to exactly? Without that understanding, it seems hard to follow on what are the improved privacy-utility trade-offs mentioned in the abstract.




**Limitations:**

There seems to be no proper limitation section, if I haven't overlooked any. Adding a discussion on the strength of privacy protection, the ethical aspects of valuating private data, and the danger of wrong incentives for individuals should be discussed.

---

> ### Author Rebuttal · Authors · 2023-08-08
>
> We thank the reviewer for the positive comments!
>
> **Q1 [Selection of the threshold in TKNN]**
>
> **A:** In our experiment, we found that the performance of TKNN-Shapley is quite robust to the selection of the threshold $\tau$. In Section 6, we set the threshold $\tau = -0.5$ (note that cosine distance ranges from $[-1, +1]$) across all experiments. **In Appendix F.3.4, we conduct the ablation study** and we find that the performance is relatively stable across a wide range of $\tau$. This is not too surprising; the formula of TKNN-Shapley (in Equation (7) in Appendix D.2.1) shows that regardless of the choice of $\tau$, if a training data point is within the threshold but has a different label from the validation data, then its value is necessarily negative. At the same time, the points that share same label as validation data are guaranteed to have a positive value. The choice of $\tau$ only decides the magnitude but not the sign. As data valuation methods only rely on the rank of data values to perform data selection, the performance of TKNN-Shapley is relatively robust to the choice of $\tau$ in identifying mislabeled data points (who get negative values). A similar argument also applies to the task of noisy data detection: noisy data is typically far from the population; hence, as long as $\tau$ is not too large, the noisy data will receive a value score close to 0. On the other hand, clean data will receive a positive value score, and the choice of $\tau$ only affects the magnitude of the data value but not the sign.
>
> Ideally, we would like to set a threshold such that for most of the queries, there are a certain amount of neighbors whose distance to the query is below the threshold. One possible way to do this is simply picking a $\tau$ that optimizes the final validation accuracy of the TKNN classifier (note that TKNN is very efficient in hyperparameter tuning).
>
> Additionally, we stress that the case where there are only a few neighbors whose distance to the query is below the threshold may not necessarily be bad. This may happen when the query is an outlier that is very far from the center of population. Hence, a reasonable classifier is supposed to assign low confidence for such a query for all classes. TKNN, in this case, is guaranteed to have such reasonable behavior. However, for KNN, the behavior is less predictable as it depends on the query example’s K nearest neighbors (which may still be far from the query in absolute distance).
>
> We have enriched our Appendix F.3.4 with the above discussion.
>
> **Q2 [Intuition for DP-TKNN-Shapley with subsampling?]** *“Additionally, I would enjoy ... there?”*
>
> **A:** Thanks for the suggestion! We have enriched the intuition for the corresponding texts! At a high level, because of the non-recursive nature of TKNN-Shapley’s computation, incorporating it with subsampling becomes straightforward.
>
> **Q3 [How to compute the final privacy cost?]**
>
> **A:** **Privacy loss & training (private) set size:** As stated in Sec 5.2, if we view the release of all Shapley scores as a whole mechanism, then it satisfies Joint DP with the same privacy parameter as in Theorem 6. At a high level, this is because we only spend privacy budget to privatize the entire private set’s statistics *once*, and reuse the privatized statistics to calculate individual DP-TKNN-Shapley value, which does not increase the privacy cost (see Theorem 16).
>
> **Privacy loss & validation set size:** Modern PRV-based privacy accountant computes the privacy loss numerically (and hence the final privacy loss has no closed-form expressions). PRV accountant [1] is a standard practice in DP literature nowadays due to the tight privacy bound calculation.
>
> [1] "Numerical composition of differential privacy." NeurIPS 2021
>
> **Q4 [Choice of $\delta$]** *“The choice of ... be more appropriate..”*
>
> **A:** For 2DPlanes dataset, we follow the prior works [1-2] and use a size-2000 subset of the training set for experiment (see Table 5 in Appendix F.1 for the summary of datasets). Hence $\delta=1e-4$ is reasonable. We note that size-2000 is already much larger compared with [1-2] (which only use a size-200 subset). We fix size-2000 to adapt the implementation from [1] easily. For completeness, we also conducted experiments on the full 2DPlanes and other datasets and use $\delta = 10^{-5}$. **See Q1 in global response**.
>
> [1] "Beta shapley: a unified and noise-reduced data valuation framework for machine learning." AISTATS 2022.
>
> [2] "Data banzhaf: A robust data valuation framework for machine learning." AISTATS 2023.
>
> **We will provide enriched discussion for Q5-7 in "official comment".**
>
> **Q5 [Runtime of TKNN-Shapley]**
>
> **A:** The algorithm for computing TKNN-Shapley in $O(N)$ runtime is detailed in Appendix D.2.2. We have highlighted in Theorem 5’s statement to make sure the reader can find it easily. At a high-level, the efficient TKNN-Shapley computation leverages the property that the required quantities for its determination are fundamentally counting queries performed on the dataset $D_{-z_i}$, which can be easily computed from the same counting queries performed on the full dataset $D$.
>
> **Q6 [What is "naively privatized KNN-Shapley" baseline?]**
>
> **A:** For "naively privatized KNN-Shapley", we are referring to the two baselines we use in the experiments of Section 6.2.1. We have highlighted this point in the paper to avoid any confusion. These two baselines are detailed in the last paragraph of Appendix C.3.1 and Appendix C.3.2, respectively.
>
> **Q7 [Limitation Section]** *“There seems ... be discussed.”*
>
> **A:** The limitation of this work is discussed in the Conclusion section. We appreciate the reviewer’s suggestion for additional discussion on those interesting topics! Some discussion on the scope of privacy protection and the ethical aspects can be found in our Conclusion. See our enriched response in "official comment" for additional discussion on the danger of wrong incentives.

---

> > ### Author Response · Authors · 2023-08-10
> > **Enriched discussion for Q5-7**
> >
> > **Q5 (enriched discussion) [Runtime of TKNN-Shapley]**
> >
> > **A:** The algorithm for computing TKNN-Shapley in $O(N)$ runtime is detailed in Appendix D.2.2. We have highlighted in Theorem 5’s statement to make sure the reader can find it easily. At a high-level, the efficient TKNN-Shapley computation leverages the property that the required quantities for its determination, namely the three quantities in **C**$(D_{-z_i})$, are fundamentally counting queries performed on the dataset $D_{-z_i}$, which can be easily computed from the same counting queries performed on the full dataset $D$.
> >
> > **Base Computation on the Full Dataset:** Instead of computing **C**$(D_{-z_i})$ for each individual data point omission (which would be computationally intensive), we first determine **C**$(D)$ on the entire dataset $D$. This step incurs computational cost of $O(N)$.
> >
> > **Efficient Computation for Individual Data Points:** For each specific data point $z_i$, we observe that all three quantities in **C**$(D_{-z_i})$ can be derived from **C**$(D)$ in $O(1)$ runtime.
> >
> > **Overall Efficiency:** Each of the computations for individual data points, owing to the preparation in step 1, can be achieved in constant time ($O(1)$). Therefore, for the entirety of the dataset, the computation scales linearly with the number of data points, resulting in an overall computational cost of $O(N)$.
> >
> >
> > **Q6 (enriched discussion) [What is "naively privatized KNN-Shapley" baseline?]** *“Could you specify what the "naively privatized KNN-Shapley" baseline from line 324 refers to exactly? Without that understanding, it seems hard to follow on what are the improved privacy-utility trade-offs mentioned in the abstract.”*
> >
> > **A:** For "naively privatized KNN-Shapley", we are referring to the two baselines we use in the experiments of Section 6.2.1. We have highlighted this point in the paper to avoid any confusion.
> >
> > **(1) DP-KNN-Shapley** Recall from Section 3.1, the original KNN-Shapley has a large global sensitivity. Nevertheless, we can still use Gaussian mechanism to privatize it based on its global sensitivity bound. We call this approach as DP-KNN-Shapley and is detailed in the last paragraph of Appendix C.3.1.
> >
> > **(2) DP-KNN-Shapley with subsampling.** Recall from Section 3.1, it is computationally expensive to incorporate subsampling techniques for DP-KNN-Shapley (detailed in the last paragraph of Appendix C.3.2). For instance, subsampled DP-KNN-Shapley with subsampling rate $q=0.01$ generally takes $30 \times$ longer time compared with non-subsampled counterpart. Nevertheless, we still compare with subsampled DP-KNN-Shapley for completeness.
> >
> > **Q7 (enriched discussion) [Limitation Section]** *“There seems to be no proper limitation section, if I haven't overlooked any. Adding a discussion on the strength of privacy protection, the ethical aspects of valuating private data, and the danger of wrong incentives for individuals should be discussed.”*
> >
> > **A:** The limitation of this work is discussed in the Conclusion section. We appreciate the reviewer’s suggestion for additional discussion on those interesting topics.
> >
> > For the strength of privacy protection, we mentioned in Conclusion section that the privacy risks addressed in this work is regarding releasing the data value scores, and we do not consider the privacy risks associated with revealing individuals' data to the central server. Future work should consider integrating secure multi-party computation (MPC) techniques to mitigate this risk, and the MPC and DP techniques need to be used together for end-to-end privacy protection.
> >
> > For the ethical aspects of valuating private data, we mentioned in Conclusion section that the incorporation of differential privacy necessarily adds a degree of randomness to the data value scores. This randomization could potentially impact the fairness of payments to data providers. The influence of this randomness, and its potential implications for payment fairness, remain areas for further investigation.
> >
> > For the danger of wrong incentives for individuals, we think it’s quite interesting and practical question. Because of the incorporation of DP, the privatized data value scores may deviate from the original data value score, and it is important to study how this deviation will impact the incentive of data contributors, and how it affects the quality of data collection.

---

> > ### Comment · Reviewer_QHAC · 2023-08-13
> > **Thank you for the clarifications**
> >
> > I would like to thank the authors for their response that clarifies my main concerns and I stay with my positive assessment of the paper.
> >
> > Yet, our of curiosity, I would still like to ask further about my Q3. The authors‘ response about this part was a bit short to help me improve my understanding. I am not very familiar with the concept of Joint DP (maybe some of the future readers of the paper might as well not be). Therefore, I would like to ask if the authors, if they have the opportunity to do so, could elaborate more on the guarantee over the whole mechanism and the privacy costs.

---

> > > ### Author Response · Authors · 2023-08-13
> > > **Thank you and here's an enriched discussion for Q3!**
> > >
> > > **Q3 (enriched discussion) [How to compute the final privacy cost?]**
> > >
> > > **A:** We apologize for the relatively short answer in the Rebuttal post due to space constraints. We provide an enriched discussion for Q3 here.
> > >
> > > **Privacy loss & training (private) set size:** Theorem 6 states the privacy loss of the mechanism $M_i(D_{-z_i}) := \widehat \phi_{z_i}(D_{-z_i})$, i.e., releasing DP-TKNN-Shapley for a *single* data point $i$. However, if we view the release of *all* Shapley scores as a whole mechanism, i.e., $M(D) := (\widehat \phi_{z_1}(D_{-z_1}), \ldots, \widehat \phi_{z_N}(D_{-z_N}))$, then this “full mechanism” satisfies Joint DP with the same privacy parameter as in Theorem 6. Joint DP considers a special kind of mechanism that releases its $i$th output (e.g., individual $i$’s data value) to individual $i$, and it requires its $i$th output to protect the privacy of the full dataset **except for $i$**. In other words, individual $i$ already knows its data, and its data value does not need to protect its own privacy.
> > >
> > > **(1) Joint DP vs standard DP.** Joint DP (JDP) considers a special kind of mechanism that releases one piece of its output to each individual in the dataset. It aligns well with the setting of data valuation where each data value score $\widehat \phi_{z_i}(D_{-z_i})$ is released to the corresponding data owner $i$. While the mechanism’s output is the full data score vector $(\widehat \phi_{z_1}, \ldots, \widehat \phi_{z_N})$, JDP only requires the data score vector that *excludes* the $i$th data value score $M_{-i}(D) := (\widehat \phi_{z_1}, \ldots, \widehat \phi_{z_{i-1}}, \widehat \phi_{z_{i+1}}, \ldots, \widehat \phi_{z_N})$ to be private with respect to $z_i$. After all, $i$th data owner already knows about its own data, and its data value score does not need to protect the privacy for itself. More formally, JDP requires $\Pr[M_{-i}(D) \in S] \le e^\epsilon \Pr[M_{-i}(D’) \in S] + \delta$ where $D, D’$ are differed by $i$th data owner’s data. Under JDP’s guarantee, even if multiple data owners collude with each other, share their data value scores, and try to infer the private information about $i$th data owner, the privacy guarantee for the $i$th data owner is still preserved since the collection of the rest of the data value scores $M_{-i}(D)$ are probabilistically indistinguishable with $M_{-i}(D’)$.
> > >
> > > **(2) Intuition of why our DP-TKNN-Shapley satisfies joint DP, and why the privacy guarantee is the same as Theorem 6.** Our DP-TKNN-Shapley mechanism $M(D) := (\widehat \phi_{z_1}(D_{-z_1}), \ldots, \widehat \phi_{z_N}(D_{-z_N}))$ satisfies JDP guarantee is **because of the special, post-processing-based computation algorithm we use**. Recall from Appendix E.2.1, to compute DP-TKNN-Shapley, we first calculate $C(D)$ and then, for each $z_i \in D$, we compute $C(D_{-z_i})$ through the deterministic post-processing step outlined in Appendix E.2.1, where the post-processing for $C(D_{-z_i})$ only depends on $z_i$ itself. Hence, due to DP’s post-processing property, in Theorem 16 we show that the JDP guarantee for the overall mechanism $M(D)$ has the same privacy parameter as the one for “sub-mechanism” $M_i(D_{-z_i}) = \widehat \phi_{z_i}(D_{-z_i})$ (stated in Theorem 6). Moreover, here’s an intuitive way to understand why $M(D)$ is collusion resistant: because random noise is only applied once on $C(D)$, each post-processed $C(D_{-z_i})$ is privatized with the same random noise. Hence, even if multiple attackers collude and share the received noisy data value scores with each other, they are effectively only sharing the same noisy sample of $C(D)$, and cannot gain more information about $z_i$ other than that. As a comparison, if we naively privatize each “sub-mechanism” $M_i(D_{-z_i}) = \widehat \phi_{z_i}(D_{-z_i})$ with independent random noise, then the attackers can gather more information by sharing its own noisy sample that contains information about $z_i$. Hence, such an approach is not only less efficient, but also does not enjoy such powerful collusion resistance property.
> > >
> > > **Privacy loss & the size of validation set:** Since we compute a privatized Shapley value score with respect to each validation point, we use privacy accounting technique to calculate the overall privacy cost for the composed mechanism. We apply the SoTA privacy accountant [1] to track the privacy loss over composition. The modern PRV-based privacy accountant computes the privacy loss numerically (and hence the final privacy loss has no closed-form expressions). It is worth noting that PRV accountant techniques [1] are a standard practice in differential privacy literature nowadays due to the tight privacy bound it can compute.
> > >
> > > [1] Gopi, et al. "Numerical composition of differential privacy." NeurIPS 2021
> > >
> > > We thank the reviewer for the valuable question and will incorporate these discussions into the paper. If there’s anything that is not clear, we are more than to provide more detailed elaboration!

---

> > > > ### Comment · Reviewer_QHAC · 2023-08-16
> > > > **Thank you for the clarification.**
> > > >
> > > > I would like to thank the authors for this helpful clarification and think that this detailed answer will make a valuable addition to the paper.

---

### Official Review · Reviewer_qAKo · 2023-07-29

**Soundness:** 3 good
**Presentation:** 2 fair
**Contribution:** 3 good
**Rating:** 6
**Confidence:** 4

**Summary:**

The paper proposes a new variant of KNN-Shapley named TKNN-Shapley from which they derive a privacy-preserving data value approach DP-TKNN-Shapley. They conduct experiments to showcase that their approaches obtain better privacy-utility tradeoff and comparable performance to baselines.

**Strengths:**

- Data valuation has limited work on privacy-preserving variants and incorporating privacy in these methods (especially KNN-Shapley) is a timely problem.
- The empirical results demonstrate the usefulness of the TKNN-Shapley and DP-TKNN-Shapley approaches.
- As a standalone variant, TKNN-Shapley has linear time complexity which is a logarithmic factor improvement over KNN-Shapley while incurring minimal drop in performance. The non-recursive nature of the proposed solution is beneficial for privacy amplification via subsampling as well.

**Weaknesses:**

- My main issue with the work is regarding motivation and presentation. Currently, the need for a DP variant to standard data valuation approaches is motivated in Section 3. However, a number of key experiments (such as with regards to the membership inference attack) are deferred to the appendix and are even then, fairly limited. In my opinion, the need for a DP variant only exists if there is a risk for membership inference, that then leads to the research question-- _is it possible that data valuation approaches leak membership information?_ If the answer is yes, then the paper is very well-motivated and DP-TKNN-Shapley is a significant contribution _if it is resilient to such adversarial scenarios_.
- However, even though the membership inference attack proposed provides an affirmative answer to the first research question (as the results in the appendix indicate) it is not covered in the main paper in sufficient detail. I believe the authors need to make the following changes: 1) provide a simple version of the membership inference attack (MIA) algorithm in the main paper (Section 3) and present basic attack results (Table 3 in appendix) to motivate the need for a privacy-preserving data valuation algorithm; 2) evaluate DP-TKNN-Shapley for the MIA paradigm to demonstrate its effectiveness under such adversarial influence. Here the latter is highly significant. Otherwise I feel that the problem is not well-motivated and some necessary results are missing from the work. To reiterate, without an extant threat to inferring membership, there should be no need for a private version of KNN-Shapley.
- While the approach itself is quite simplistic, I am inclined to agree with the authors' statement that simplicity is better when private solutions need to be implemented.
- Is there a reason why the authors have not considered the "learning with weighted samples" or "point addition/removal" application experiments for TKNN-Shapley evaluation as in previous work (refer to [1-2]) on data valuation?
- The authors should also go through the manuscript and fix any minor formatting errors and typos-- for example, Table 10 and 11 captions in the appendix should read $K$ instead of $\tau$.

References:
- [1] Kwon, Yongchan, and James Zou. "Beta shapley: a unified and noise-reduced data valuation framework for machine learning." arXiv preprint arXiv:2110.14049 (2021).
- [2] Wang, Tianhao, and Ruoxi Jia. "Data banzhaf: A data valuation framework with maximal robustness to learning stochasticity." arXiv preprint arXiv:2205.15466 (2022).

**Questions:**

These are provided in the Weaknesses section above.

**Limitations:**

Yes.

---

> ### Author Rebuttal · Authors · 2023-08-09
>
> We thank the reviewer for the positive comments!
>
> **Q1. [Improving the motivation section]** *“I believe the authors need to make the following changes: 1) provide a simple version of the membership inference attack (MIA) algorithm in the main paper (Section 3) and present basic attack results (Table 3 in appendix) to motivate the need for a privacy-preserving data valuation algorithm; 2) evaluate DP-TKNN-Shapley for the MIA paradigm to demonstrate its effectiveness under such adversarial influence.”*
>
> **A:** We thank the reviewer for the very concrete suggestions! Here are our changes to the paper according to your two suggestions:
> - In Section 3, we have enriched the text with a high-level summary of the MI attack proposed in Appendix C.2.2, and we have moved part of the MI attack results to the main text. For completeness, we additionally conducted the experiment of the proposed MIA against (non-private) TKNN-Shapley (**see Q1 in global response and Table 4 in the Rebuttal’s pdf**). The result shows that both KNN-Shapley and TKNN-Shapley serve as an additional channel for leaking membership information of private data.
> - We additionally conducted the experiment of the proposed MIA against DP-TKNN-Shapley (**see Q1 in global response and Table 5 and 6 in the Rebuttal’s pdf**). Compared with the result on non-private TKNN-Shapley, the overall attack performance drops to around 0.5 (the performance of random guess). The result shows that DP-TKNN-Shapley is indeed very effective against membership inference attacks. Moreover, we stress that differential privacy is a robust solution against any attacks in an information-theoretic sense. Hence, the privacy protection is not only for the MI attack we design here, but also for any future attacks.
>
> We have added these additional results to Appendix C.2.2.
>
> **Q2. [Why not consider "learning with weighted samples" or "point addition/removal" application experiments?]**
>
> **A:** Thanks for the great question! We would first like to point out some limitations of using these two tasks to evaluate a data value notion.
>
> For "learning with weighted samples" experiment, the performance is highly dependent on the *distribution* of data value scores. Imagine that there are a very small group of data points that receive extremely high value, weighted training based on the value scores will result in poor performance as the learning algorithm will be “lazy” to fit the rest of data points that receive lower value scores.
>
> For “point addition/removal” experiment, the performance is highly dependent on the diversity of selected data points. However, a reasonable data value notion typically needs to satisfy “symmetry axiom”, which is being interpreted as fairness. Hence, if one selects data points with high values, it is likely that the selected data points lack diversity, as similar data points are required to receive similar value scores. In fact, preliminary theoretical and empirical results [3] show that the Shapley value as well as other cooperative-game-theory-based data valuation can result in arbitrarily low model performance in the worst case.
>
> Having said that, we additionally conducted the “point removal” experiment (**see Q2 in global response**) where we remove the data points that receive the lowest data value scores, then train the model on the remaining of the dataset and check whether the test accuracy is actually improved. Here, we remove at most 10% of the full dataset. In this case, the dataset diversity is not being affected a lot by removing bad training data points. The specific settings for the model architecture used in the experiment follow from [2]'s Appendix D.3.2, and the specific settings for the datasets follow from the mislabeled data detection experiment (Appendix F.3.1). **Figure 7 in the Rebuttal’s pdf** shows results on the private setting (where we use $\varepsilon=1$), and **Figure 8 in the Rebuttal’s pdf** shows results on the non-private setting. Similar to the results for mislabel/noisy data detection experiment, we observe that **(1)** DP-TKNN-Shapley significantly outperforms the naively privatized KNN-Shapley in terms of privacy-utility tradeoff; **(2)** non-private TKNN-Shapley achieves comparable performance as the original KNN-Shapley in discerning data quality.
>
> [1] Kwon, Yongchan, and James Zou. "Beta shapley: a unified and noise-reduced data valuation framework for machine learning." AISTATS 2022.
>
> [2] Wang, Jiachen T., and Ruoxi Jia. "Data banzhaf: A robust data valuation framework for machine learning." AISTATS 2023.
>
> [3] https://openreview.net/pdf?id=kz6rsFehYjd (Section 4).
>
>
> **Q3. [Typos]**
>
> **A:** Thanks for the catch! We have taken another pass of the paper to address the typos.

---

> > ### Comment · Reviewer_qAKo · 2023-08-13
> > **Response to Rebuttal**
> >
> > I would like to thank the authors for performing the experiments I suggested and for addressing concerns regarding motivation. Given this, I would like to maintain my current score on the work-- as a technically solid, moderate-to-high impact paper, and one with no major concerns w.r.t. evaluation, etc.

---

> > > ### Author Response · Authors · 2023-08-14
> > > **Thanks for your positive feedback!**
> > >
> > > We thank the reviewer for the positive feedback about our work! We appreciate the concrete suggestions from your review comments, and we have incorporated these suggestions into our paper.

---

### Author Rebuttal · Authors · 2023-08-09

We thank all of the reviewers for their detailed and valuable comments. We are glad that our work receives a majority of positive reviews. We considered the reviews carefully and modified our paper accordingly. We have answered other questions in the individual responses. Here’s our global response, which summarizes the additional experiments and background materials we have/will be added to the paper:

**Q1 (for Reviewer qAKo and Reviewer z2hx) [Additional experiments for the membership inference (MI) attack proposed in Appendix C.2.2]**

**A:** **Evaluate the proposed MI attack on (non-private) TKNN-Shapley:** We additionally conducted the experiment of the proposed MIA against (non-private) TKNN-Shapley (**see Table 4 in the Rebuttal’s pdf**). We follow the exact experiment setup as one described in Appendix C.2.2. Similar to the attack results on KNN-Shapley, our MI attack also achieves non-trivial performance (>0.6) for most of the datasets and choice of parameters. This shows that both KNN-Shapley and TKNN-Shapley serve as additional channels for leaking membership information of private data.

**Evaluate the proposed MI attack on DP-TKNN-Shapley:** To directly demonstrate how DP-TKNN-Shapley can mitigate the privacy risk, we additionally conducted the experiment of evaluating the proposed MI attack on DP-TKNN-Shapley (**see Table 5 and 6 in the Rebuttal’s pdf**). We follow the exact experiment setup as the one described in Appendix C.2.2, and we evaluate the attack performance when $\varepsilon = 0.5$ (Table 5) and $\varepsilon = 1$ (Table 6) for DP-TKNN-Shapley. Compared with the result on non-private TKNN-Shapley, we can see that the overall attack performance drops to around 0.5 (the performance of random guess). The result shows that DP-TKNN-Shapley is indeed very effective against membership inference attacks.

We have incorporated these results into the paper.

**Q2 (for Reviewer qAKo) [Additional experiments for the task of data removal]**

**A:** We additionally conducted the “point removal” experiment where we remove the data points that receive the lowest data value scores, then train the model on the remaining of the dataset and check whether the test accuracy is actually improved. Here, we remove at most 10% of the full dataset. In this case, the dataset diversity is not being affected a lot by removing bad training data points. The specific settings for the model architecture used in the experiment follow from [1]'s Appendix D.3.2, and the specific settings for the datasets follow from the mislabeled data detection experiment (Appendix F.3.1). **Figure 7 in the Rebuttal’s pdf** shows results on the private setting (where we use $\varepsilon=1$), and **Figure 8 in the Rebuttal’s pdf** shows results on the non-private setting. Similar to the results for mislabel/noisy data detection experiment, we observe that **(1)** DP-TKNN-Shapley significantly outperforms the naively privatized KNN-Shapley in terms of privacy-utility tradeoff; **(2)** even non-private TKNN-Shapley achieves comparable performance as the original KNN-Shapley in discerning data quality.

[1] Wang, Jiachen T., and Ruoxi Jia. "Data banzhaf: A robust data valuation framework for machine learning." AISTATS 2023.

**Q3 (for Reviewer QHAC) [Additional experiments for setting a smaller $\delta$]**

**A:** We thank the reviewer for the concrete suggestion, and we have additionally conducted experiments on the full datasets of 2DPlanes, Phonome, and CPU datasets (of sizes around 40k, 5k, and 8k respectively) and use $\delta = 10^{-5}$. The experiment settings follow exactly the one described in Appendix F.3.1, and the results are shown in **Table 15 in Rebuttal’s PDF**. Similar to the result for the case of $\delta = 10^{-4}$, DP-TKNN-Shapley still achieves significantly better performance compared with the two baseline approaches. We have incorporated this experiment result into Appendix F.3.

**Q4 (for Reviewer SBGb and QHAC) [Brief background for the privacy risks in releasing aggregated statistics]**

**A:** Since 1998, researchers have observed that a lot of seemingly benevolent aggregate statistics of a dataset can be used to reveal sensitive information about individuals [1]. A classic example is Netflix Prize fiasco, where the researchers show that an anonymized dataset can leak many sensitive information about individuals [2]. Irit and Nissim [3] proved that “revealing too many statistics too accurately leads to data privacy breach”. A great amount of discussion and practical realization of these privacy attacks on aggregated statistics can be found in [4]. In 2020, the US Census Bureau used these privacy attacks to justify their use of differential privacy.

KNN-Shapley score for an individual is one kind of aggregated statistics that depends on the rest of the dataset. Hence, KNN-Shapley score intrinsically reveals private information about the rest of the dataset (where we use membership inference attack as a concrete example in our paper). In addition, when users collude, their KNN-shapley values can be combined to make joint inferences about the rest of the dataset.

We will integrate this discussion into the paper to improve the readability of the privacy risks associated with data valuation techniques.

[1] Samarati, Pierangela, and Latanya Sweeney. "Protecting privacy when disclosing information: k-anonymity and its enforcement through generalization and suppression." (1998).

[2] Narayanan, Arvind, and Vitaly Shmatikov. "How to break anonymity of the netflix prize dataset." (2006).

[3] Dinur, Irit, and Kobbi Nissim. "Revealing information while preserving privacy." Proceedings of the twenty-second ACM SIGMOD-SIGACT-SIGART symposium on Principles of database systems. 2003.

[4] Dwork, Cynthia, et al. "Exposed! a survey of attacks on private data." Annual Review of Statistics and Its Application (2017)

---

### Comment · Area_Chair_k3SD · 2023-08-18
**Wait for Reviewers' Feedback**

Dear Reviewer yczn,

The authors and I are eager to know whether the author responses successfully address your concern. The Author-Reviewer discussion ends on **August 21**. You are strongly encouraged to directly reply to the authors.

Thank you for your hard work.

PS: (1) This is a public thread. (2) I am expecting to know your thoughts as well. If you want to individually reply to me, please use the thread of internal discussion.

Kind Regards,

AC

---

### Decision · Program_Chairs · 2023-09-21

**Decision:**

Accept (spotlight)

**Comment:**

I have read all the materials of this paper including the manuscript, appendix, comments, and response. Based on collected information from all reviewers and my personal judgment, I can make the recommendation on this paper, acceptance. All reviewers and I championed this paper.

**Research Question**

This paper considers the data valuation problem from the privacy perspective, which is a novel and overlooked angle in this area.

**Presentation**

The presentation is very clear with structured organizations and summarized roadmaps in each section.

**Challenges**

Before illustrating the research challenges, the authors provided two illustrating examples to demonstrate the risks of the current framework of Shaley values. Personally, the example in Figure 2 is not interesting to me. It is quite normal that the Shapley value will change with the nearby data point removed. Instead, the example of membership inference attack attracts me a lot, where I traced Appendix C for more details. I suggest the authors to put more content of this example in the main paper.

The authors clearly illustrated the challenges of extending the current framework of Shapley value for privacy protection in Line 200-215. I omitted my rephrase here.

**Philosophy**

The philosophy to tackle two challenges in Line 200-215 is missing. The authors directly provided their solution TKNN-Shapley in Section 4. I am not sure whether the authors can feel there is a gap between Section 3 and Section 4. In another word, the authors need to first demonstrate how they aimed to solve the challenges, then provide their solutions later.

**Technique**

The techniques and corresponding analyses make sense to me.

**Experiments**

The current results with the extra ones in the author response demonstrate the effectiveness of the proposed method in terms of both effectiveness and efficiency.